JCB Journal of Cell Biology

## TOOLS

# Genome-scale requirements for dynein-based transport revealed by a high-content arrayed CRISPR screen

Chun Hao Wong[1,2], Steven W. Wingett[1], Chen Qian[3], Morag Rose Hunter[2], J. Matthew Taliaferro[4], Douglas Ross-Thriepland[2], and Simon L. Bullock[1]

**The microtubule motor dynein plays a key role in cellular organization. However, little is known about how dynein's biosynthesis, assembly, and functional diversity are orchestrated. To address this issue, we have conducted an arrayed CRISPR loss-of-function screen in human cells using the distribution of dynein-tethered peroxisomes and early endosomes as readouts. From a genome-wide gRNA library, 195 validated hits were recovered and parsed into those impacting multiple dynein cargoes and those whose effects are restricted to a subset of cargoes. Clustering of high-dimensional phenotypic fingerprints revealed co-functional proteins involved in many cellular processes, including several candidate novel regulators of core dynein functions. Further analysis of one of these factors, the RNA-binding protein SUGP1, indicates that it promotes cargo trafficking by sustaining functional expression of the dynein activator LIS1. Our data represent a rich source of new hypotheses for investigating microtubule-based transport, as well as several other aspects of cellular organization captured by our high-content imaging.**

## Introduction

Cytoskeletal motors play a central role in organizing the intracellular space. The cytoplasmic dynein-1 motor (dynein) is responsible for almost all motility toward the minus ends of microtubules and consequently carries a large variety of cargoes—including organelles, macromolecules, and pathogens—toward the cell's interior (Reck-Peterson et al., 2018).

Dynein is a 1.3-MDa multisubunit complex whose motor and microtubule-binding activities are housed in the C-terminal region of the heavy chain subunit—DYNC1H1 (Carter et al., 2016). The N-terminal region of DYNC1H1 mediates self-dimerization and provides a scaffold for the accessory chains—two copies each of an intermediate chain (DYNC1I1 or DYNC1I2) and a light intermediate chain (DYNC1LI1 or DYNC1LI2), and six copies of a light chain (DYNLL1, DYNRB1, or DYNLT1).

In vitro reconstitutions have shown that long-range motility of dynein is dependent on another large, multisubunit complex—dynactin—and one of a number of coiled-coil-containing cargo adaptors (termed "activating adaptors") (McKenney et al., 2014; Schlager et al., 2014; Reck-Peterson et al., 2018). The activating adaptors stabilize the interaction of dynein with dynactin, which switches on processive movement. The Lissencephaly-1 (LIS1/

PAFAH1B1) protein also plays a critical role in cargo transport by binding to the DYNC1H1 motor domain and promoting formation of the dynein–dynactin-activating adaptor assembly (Baumbach et al., 2017; Qiu et al., 2019; Elshenawy et al., 2020; Htet et al., 2020). The importance of LIS1 is underlined by the finding that even modest reductions in its abundance impair dynein function and cause neurodevelopmental disease (Reiner et al., 1993; Cardoso et al., 2002; Gambello et al., 2003; Hebbar et al., 2008).

While in vitro studies have greatly advanced our understanding of dynein activation, many questions remain about how cargo trafficking by the motor is orchestrated in cells. For example, how is biosynthesis of the individual components of the transport machinery, as well as their assembly into larger complexes, controlled? And how is the functional diversity of dynein achieved: are there mechanisms that regulate the behavior of dynein complexes bound to specific cargoes?

To gain a foothold into these and other aspects of dynein biology, we conducted a genome-wide loss-of-function CRISPR screen in human cells for factors that influence localization of the motor's cargoes, followed by high-dimensional phenotypic analysis of the hits. Our results represent a valuable resource for

[1]Cell Biology Division, Medical Research Council Laboratory of Molecular Biology, Cambridge, UK; [2]Centre for Genomic Research, Discovery Sciences, AstraZeneca, Cambridge, UK; [3]Quantitative Biology, Discovery Sciences, AstraZeneca, Cambridge, UK; [4]Department of Biochemistry and Molecular Genetics, University of Colorado Anschutz Medical Campus, Aurora, CO, USA.

Correspondence to Simon L. Bullock: sbullock@mrc-lmb.cam.ac.uk; Douglas Ross-Thriepland: douglas.ross-thriepland@astrazeneca.com.



mechanistic dissection of microtubule-based trafficking, as well as several other aspects of cellular organization captured in our images.

## Results

### Optimized procedures for arrayed gene disruption with CRISPR/Cas9

We first sought to establish scalable procedures for CRISPR/Cas9-mediated gene editing in an arrayed format, i.e., in which one gene is targeted per well. Screening in this manner, as opposed to the more conventional pooled format, greatly facilitates the establishment of phenotype–genotype relationships and is compatible with multivariate imaging readouts (Przybyla and Gilbert, 2022).

We developed a protocol in which a large pool of cells is transfected with Cas9 mRNA and seeded into 384-well plates predispensed with four synthetic two-part guide RNAs (hereafter crRNAs) (Basila et al., 2017) that target a different gene in each well (Fig. 1 A). Delivering Cas9 by mRNA transfection circumvents the need to make stable cell lines expressing the enzyme, while using a pool of mRNA-Cas9–expressing cells as starting material for crRNA delivery removes well-to-well differences in Cas9 transfection as a variable.

The mRNA-Cas9 transfection protocol was optimized in a panel of five commonly used human cell lines derived from different organs (U-2 OS, ARPE-19, HEK-293, IMR-90, and SH-SY5Y; Fig. S1). These experiments defined mRNA concentrations, transfection reagents, and transfection conditions that gave a very high proportion of Cas9-expressing cells (90–100%), yet had minimal toxicity. Editing efficiency with the optimized conditions was evaluated in ARPE-19 and U-2 OS cells using crRNA pools targeting six genes, including LIS1, DYNC1H1, and DCTN1, which encodes a dynactin subunit. 70–80% of ARPE-19 cells (Fig. S2 A) and 85–100% of U-2 OS cells (Fig. 1, B and C; and Fig. S2 A) had strongly reduced expression of the targeted proteins 72 h after crRNA transfection. Thus, our RNA-based delivery methods disrupt a range of target genes with high efficiency. These experiments also demonstrate that a 72-h window allows retention of sufficient edited cells when targeting essential genes that disrupt dynein-based transport.

### Imaging-based assays for dynein activity

Because of the particularly high rates of editing observed in U-2 OS cells, we sought to develop a readout of dynein activity in this cell type that is suitable for an arrayed screen. We took advantage of a previously characterized U-2 OS line (hereafter U-2 OS PEX) (Vincent et al., 2020) that has an inducible system for dynein-mediated relocalization of fluorescent peroxisomes (Kapitein et al., 2010; Vincent et al., 2020). This line stably expresses the constitutively active N-terminal region of the activating adaptor BICD2 (BICD2N) fused to GFP and the FK506-rapamycin binding (FRB) domain (GFP-BICD2N-FRB), as well as a peroxisome targeting sequence (PTS) fused to RFP and FK506-binding protein 12 (FKBP) (PTS-RFP-FKBP) (Fig. 1 D). Addition of rapamycin triggers binding of BICD2N to peroxisomes via FRB–FKBP heterodimerization, which in turn recruits dynein–

dynactin (Fig. 1 D). This leads to tight clustering of peroxisomes—which otherwise are dispersed in the perinuclear region—at the juxtanuclear microtubule-organizing center (MTOC), where microtubule minus ends are enriched (Fig. 1 E).

The assay was optimized by quantifying the number of GFP and RFP spots, which acts as a proxy for clustering of peroxisomes at the MTOC, in response to rapamycin concentration and incubation time, as well as the number of seeded cells (Fig. S2, B–D). We also confirmed that rapamycin-induced relocalization of peroxisomes is impaired by microtubule depolymerization with nocodazole (Vincent et al., 2020) and demonstrated this is also the case when LIS1 or DYNC1H1 are targeted with crRNA pools (crLIS1 and crDYNC1H1) using our optimized CRISPR protocol (Fig. 1 E).

The assay was scaled and validated by seeding cells on multiple plates predispensed with rows of crRNA pools targeting LIS1, DYNC1H1, or PLK1 (disruption of which blocks cell proliferation and thus serves as a label-free control for editing efficiency [Strezoska et al., 2017; Ross-Thriepland et al., 2020]). As controls, wells were dispensed with crRNAs, lacking targets in the human genome (non-targeting control; NTC), or nocodazole. Highly efficient gene disruption was observed across the plates for all three gene targets (Fig. S3). Furthermore, there was a consistent change in the number of RFP and GFP spots in crLIS1, crDYNC1H1, and nocodozole-treated wells compared with NTC (Fig. 1 F), demonstrating robust dispersal phenotypes. crLIS1 caused a stronger peroxisome dispersal phenotype than crDYNC1H1 (Fig. 1 F), which may be related to greater reduction in the level of its target protein (Fig. 1, B and C). The assay window measured by the robust Z-prime (rZ′) score between NTC and crLIS1 was 0.34 and 0.55 for the RFP and GFP data, respectively, indicating suitability for imaging-based screening (Bray and Carpenter, 2004).

To maximize the information gained from a genome-wide CRISPR screen, we sought to additionally monitor localization of early endosomes, which rely on dynein–dynactin for enrichment in the perinuclear cytoplasm (Driskell et al., 2007). Staining with an antibody to early endosome antigen 1 (EEA1) confirmed that nocodazole, crDYNC1H1, and crLIS1 disperse early endosomes in U-2 OS cells (Fig. 1 G). Unlike the fluorescent peroxisomes in the U-2 OS PEX line, dynein–dynactin is linked to early endosomes by activating adaptors of the HOOK family (Christensen et al., 2021). Thus, simultaneously screening for defects in peroxisome and early endosome localization can potentially reveal factors involved in trafficking by discrete dynein–dynactin-activating adaptor complexes.

### Genome-wide screening and performance assessment

For the genome-wide screen, we adapted the assays for peroxisome and early endosome distribution for end-to-end execution via automated liquid handlers. Screening was performed across 61 unique 384-well plates arrayed with a commercial crRNA library that targets 18,253 genes with four guides per gene and has been designed to minimize off-target cutting (see Materials and methods). Neutral (NTC) and positive (crLIS1 and crPLK1) controls were included in each plate for downstream quality assessment and normalization (Fig. 2 A).

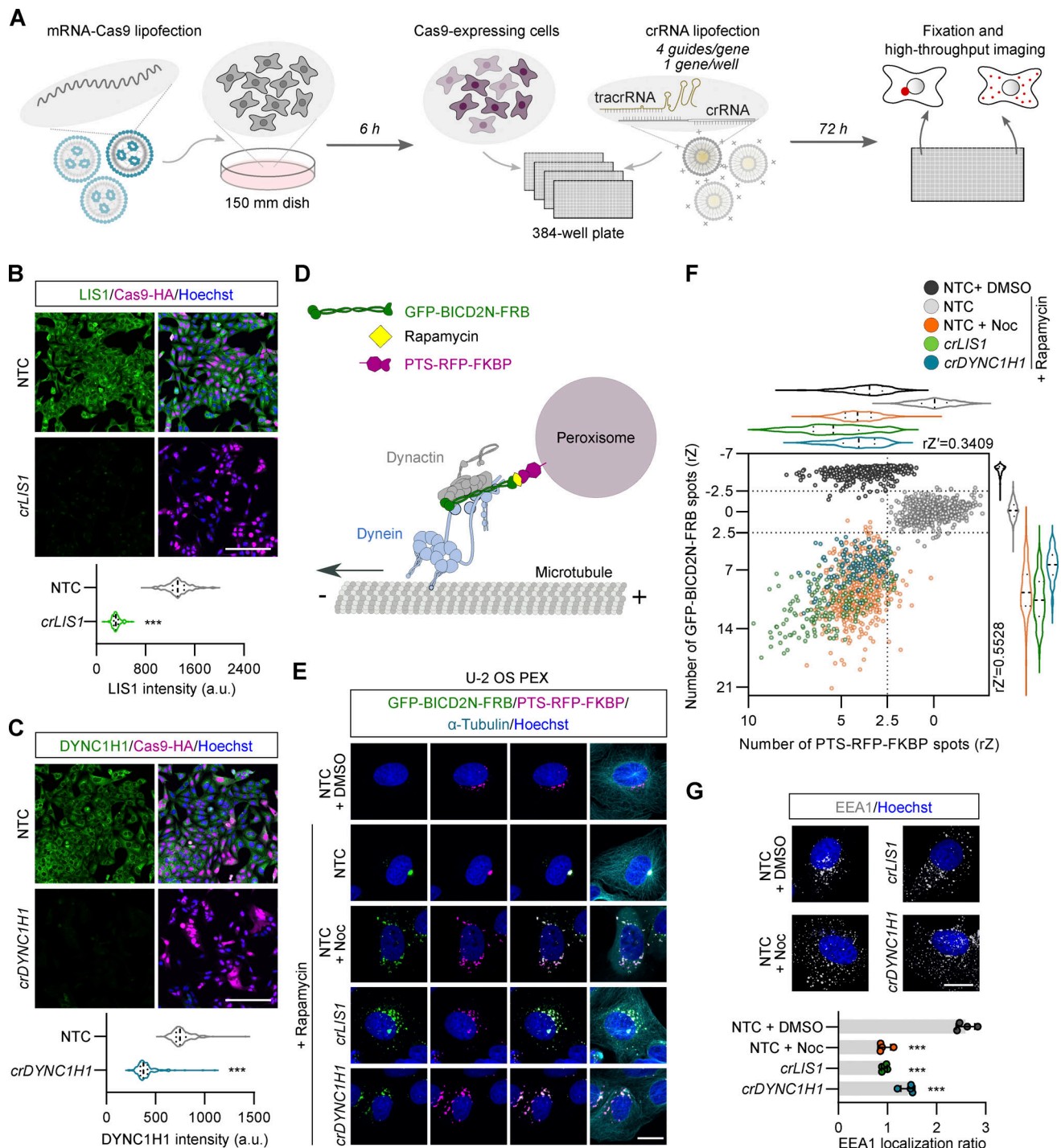

Figure 1. **Assay development for arrayed CRISPR/Cas9 screening. (A)** Workflow for image-based screening of dynein cargo localization. **(B and C)** Representative images and quantification of immunostained, unmodified U-2 OS cells following CRISPR/Cas9-mediated editing of *LIS1* (B) or *DYNC1H1* (C). Hoechst, DNA stain; *cr*, *crRNA*. Violin plots show fluorescence intensity values at the single-cell level (minimum of 100 cells from at least four wells for each group; median, bold line; first/third quartile, dashed lines). ***P < 0.001 (two-tailed Mann–Whitney-test). Scale bar, 200 μm. **(D)** Illustration of inducible peroxisome relocalization assay. **(E)** Representative images of U-2 OS PEX cells stained for microtubules (α-Tubulin) and DNA (Hoechst) after the indicated treatments. Cells were treated with either DMSO (vehicle), rapamycin alone, or rapamycin with nocodozole (Noc) for 2.5 h before fixation. Scale bar, 20 μm. **(F)** Validation of inducible peroxisome relocalization assay in high-throughput format. Scatter plot and corresponding violin plots (median, bold line; first/third quartile, dashed lines) of the number of GFP-BICD2N-FRB and PTS-RFP-FKBP spots. Data points represent rZ normalization (central reference = NTC treated with rapamycin; value increases with cargo dispersion) with mean per cell values aggregated at well level (minimum of 100 wells analyzed from 3 × 384-well plates). rZ' values show an assay window between NTC with rapamycin and *crLIS1* with rapamycin. Shift on plot for NTC + DMSO versus NTC + Rapamycin conditions is due to the combination of concentration of GFP-BICD2N-FRB on peroxisomes and perinuclear clustering of these structures. **(G)** Representative images and quantification of early endosome (EEA1) dispersion in unmodified U-2 OS cells after indicated treatments. The bar graph shows the ratio between EEA1 spot number in the perinuclear region versus the peripheral region (lower values indicate increased dispersion). Data points represent mean per cell values aggregated at well level (minimum of 100 cells analyzed per well; four wells analyzed per condition). Error bars, SD. ***P < 0.001 (one-way ANOVA with Dunnett's multiple comparison versus NTC + DMSO). Scale bar, 20 μm.

The assay entailed fixing mRNA-Cas9–transfected U-2 OS PEX cells 72 h after crRNA transfection and 2.5 h after rapamycin addition. Cells were then stained with antibodies to EEA1, along with Hoechst (to highlight DNA), and antibodies to α-Tubulin (for cell segmentation). The resulting signals, together with those from the GFP and RFP channels, were captured with a high-content imaging platform (Fig. 2 B). In total, 8,150,247 viable cells from 23,424 wells (four fields of view per well; median of 345 cells analyzed per well) were segmented for multiparametric analyses. The complete set of quantitative data from the screen is contained in Table S1.

To gauge editing efficiency, we first evaluated the effects of crPLK1 on cell survival. As cell number is variable in the context of microplate-based assays, we performed population gating for viable cells based on Hoechst staining (i.e., removing cells with apoptotic or mitotic features, or abnormal nuclear morphology: rZ′ = 0.7 [NTC versus crPLK1]). We observed a large decrease in cell viability in crPLK1 wells across the plates, as well as with the single copy of crPLK1 in the crRNA library (Fig. 2 C). Although PLK1 was recently implicated in regulation of BICD2 function (Gallisà-Suñé et al., 2023), the library copy of crBICD2 did not affect viability (Fig. S4 A) even though it was active in other assays (see below). This observation suggests that PLK1's role in cell proliferation does not involve BICD2. In keeping with a role of LIS1 in promoting dynein function in mitosis (Faulkner et al., 2000; Moon et al., 2014), crLIS1 controls across the plates, as well as the single library copy of crLIS1, reduced cell viability but to a lesser extent than crPLK1 (Fig. 2 C). Together, these results indicate highly consistent Cas9/crRNA activity across the screen.

In addition to crPLK1, crRNAs targeting 62 genes reduced cell viability to a significantly greater extent than crLIS1. Approximately 75% of these genes have been classed as essential in multiple cancer cell lines (Fig. S4 A) (Tsherniak et al., 2017). Evaluating nuclear area and roundness across our assay plates identified many genes previously implicated in the control of nuclear morphology (Fig. 2 D) (Tan and Martin, 2016; Strezoska et al., 2017; Yan et al., 2021; Funk et al., 2022). We also analyzed induction of micronuclei (Fig. S4 B), a phenotype that to our knowledge was not specifically assessed in earlier genome-wide screens for nuclear morphology defects (Tan and Martin, 2016; Strezoska et al., 2017; Yan et al., 2021; Funk et al., 2022). Many of the hits from this analysis encode components of the mitotic machinery, consistent with the contribution of chromosome segregation defects to micronuclei formation (Crasta et al., 2012). In addition to recovering genes that were expected to influence cell survival, nuclear morphology, and micronuclei formation, these analyses also implicated many other genes in these processes (Tables S1 and S2). These observations show that our procedures effectively identify known, as well as novel, genotype–phenotype associations.

## Recovery of known and candidate novel players in dynein biology

To identify genes that are candidates to contribute to dynein-based trafficking, we performed multi-parametric analysis on the PTS-RFP-FKBP, GFP-BICD2N-FRB, and EEA1 signals across the screening plates. crRNAs with strong effects on cell viability

and nuclear morphology were excluded from further analysis, as they may affect cargo localization indirectly. Fig. 2 E plots the total number of GFP-BICD2N-FRB and PTS-RFP-FKBP spots per cell, which was among the metrics that gave a robust readout of peroxisome dispersion (rZ′ = 0.41–0.45 [NTC versus crLIS1]). Genes encoding several components of the dynein complex, as well as LIS1, were among 217 factors whose targeting with library crRNAs caused a significant change in the number of GFP and/or RFP spots (≥ ±2*SD of NTC). BICD2 was also recovered as a hit in this analysis (Fig. 2 E), which may reflect the ability of two of the crRNAs in this pool to target the GFP-BICD2N-FRB construct. We used several other metrics to quantify subcellular localization of GFP or RFP spots by segmenting the cytoplasm into perinuclear, intermediate, and outer regions. This analysis recovered genes encoding additional dynein and dynactin constituents (Table S3), as well as 114 other genes not identified in the analysis of total spot number (Table S2). The gene encoding the BICD2 paralogue, BICD1, was not recovered in these analyses, despite being expressed in U-2 OS cells (Beck et al., 2011). This result is expected as peroxisome coupling in our assay is mediated by BICD2.

crRNAs targeting dynein–dynactin components and LIS1 were also part of 35 library pools that significantly reduced perinuclear enrichment of early endosomes (Fig. 2 F). These hits also included crRNAs for HOOK3 and AKTIP (also known as FTS), which encode two of the proteins that form an "FHF" complex linking dynein to early endosomes (Christensen et al., 2021). The gene encoding the third FHF component, FAM160A2 (also known as FHIP1B), did not meet the threshold for inclusion as a hit for endosome dispersion but was very close to doing so. crRNAs for HOOK3, AKTIP, and FAM160A2 did not affect peroxisome distribution (Fig. 2 E), consistent with their selective function in early endosome trafficking (Christensen et al., 2021). We also found that several crRNA pools were associated with excessive perinuclear clustering of early endosomes (Fig. 2 F), raising the possibility that their target genes inhibit endosomal transport by dynein. These genes included PAFAH1B2, which encodes a catalytic subunit of the platelet-activating factor acetylhydrolase Ib complex that binds LIS1 (Hattori et al., 1994). Our finding that PAFAH1B2 disruption promotes early endosome clustering supports the hypothesis that competition for LIS1 between PAFAH1B2 and dynein modulates motor activity (Ding et al., 2009). More generally, the identification of multiple known players in dynein-based transport demonstrates that our screening and analysis pipeline effectively identifies genes important for this process.

Our analysis additionally revealed crRNA pools that affected the morphology of peroxisomes (Fig. S4 C) and endosomes (Fig. 2 F) without changing their distribution in the cell (Table S2). Some of these genes have an established link with these structures, notably the PEX genes and DNML1, which function in peroxisome biogenesis and fission, respectively (Koch et al., 2003; Waterham and Ebberink, 2012), and VPS11, PIK3R4, and LYST, which have roles in endosome biogenesis and/or endocytosis (Balderhaar and Ungermann, 2013; Sepulveda et al., 2015). However, several other genes in this category have not previously been linked to peroxisome or endosome biology.

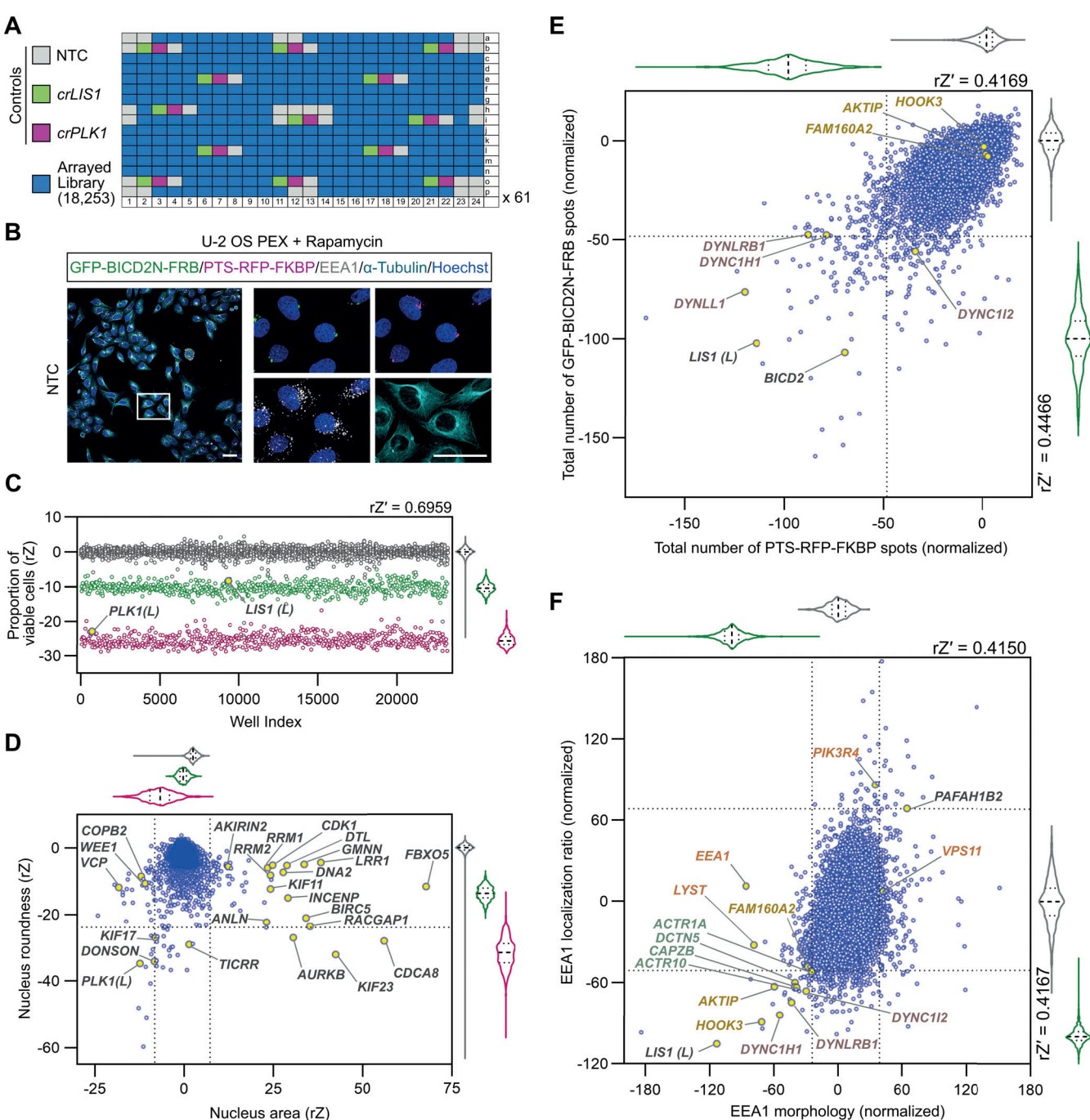

Figure 2. **The genome-wide screen recovers known components of the dynein–dynactin machinery, as well as novel hits. (A)** Plate layout for genome-wide screen. **(B)** Example of imaging data from the screen (maximum intensity projections of Z-stacks captured with a 20×/1.0 NA water objective). Scale bar, 50 µm. **(C)** Evaluation of editing efficiency across the screen using cell viability as a readout. Scatter plot and corresponding violin plot for controls (median, bold dashed line; first/third quartile, dashed lines) of the proportion of viable cells (gated based on nuclear morphology of NTC cells) per well after treatment with NTC, *crLIS1*, or *crPLK1*. Data points (color-coded as in A) are rZ normalized values (central reference = NTC). rZ' value shows assay window between NTC and *crPLK1*. Library copies of *crPLK1* and *crLIS1* are labeled with "(*L*)." **(D)** Effects of arrayed library crRNAs on area and roundness of nuclei. Scatter plot and corresponding violin plots (median, bold dashed line; first/third quartile, dashed lines) of rZ normalized values (color coded as in A; central reference = NTC). Dashed lines represent ± 4*SD of NTC and *crLIS1* (x-axis) or –3.5*SD of NTC and *crLIS1* (y-axis), which were thresholds for hit calling. Genes previously shown to influence nuclear morphology are highlighted. **(E and F)** Example of endpoints used for hit selection from peroxisome (E) and early endosome (F) data. Scatter plot and corresponding violin plots (median, bold dashed line; first/third quartile, dashed lines) of normalized values (color-coded as in A) based on the neutral control (NTC, 0) and positive control (*crLIS1*, –100); rZ' values show assay window between NTC and *crLIS1* control wells. Dashed lines on the x- and y-axes represent, respectively, thresholds of –2*SD (E) or –3*SD (F) used for hit calling. In F, "EEA1 morphology" values were generated from a linear discriminant analysis weighted average of three endpoints quantifying the symmetry, intensity profile, and texture of EEA1 signal. "EEA1 localization ratio" is the ratio between the number of EEA1 spots in the perinuclear region versus the peripheral region. Core components of dynein and dynactin that met the threshold for hit calling are labeled with purple and teal text, respectively. Other categories of genes highlighted in the text are labeled in gold or orange.

## Hit validation and coarse-grain phenotypic analysis

As our main objective was to identify factors important for dynein-based trafficking, we focused our subsequent efforts on hits that affected subcellular distribution of peroxisomes and/or early endosomes. We took forward a total of 376 hits in this category for validation in a secondary screen; 322 of these met our criteria for one or more metrics of peroxisome dispersion, 45 increased or decreased early endosome clustering, and nine affected localization of both cargoes (Table S4). We also selected *FAM160A2* for the secondary screen because, as described above, this gene was very close to the hit threshold for EEA1 and has an established role in early endosome transport. Analysis of the genes targeted by the shortlisted crRNAs using Metascape (Zhou et al., 2019) revealed a particularly strong enrichment of ontology terms associated with RNA metabolism and the cell cycle, as well as enrichment of several other terms including those related to trafficking and the microtubule cytoskeleton (Table S4).

We retested the activity of the shortlisted crRNAs toward BICD2N-tethered peroxisomes in rapamycin-treated U-2 OS PEX cells, as well as toward early endosomes in untreated, unmodified U-2 OS cells. As the primary screen indicated specificity of some factors for a subset of cargoes, we also assessed the effects of each of the crRNA pools on perinuclear localization of the Golgi apparatus (marked with TGN46 antibodies) in untreated, unmodified cells. Studies in other cell types have shown that impairing dynein function causes dispersion of the Golgi (Harada et al., 1998; Palmer et al., 2009), and we confirmed this is also true in U-2 OS cells using *crDYNC1H1* and *crLIS1* (Fig. S5 A).

Each cargo was assayed in two independent biological replicates (each with four technical replicates), in which there was good agreement in general between the effects of the crRNA pools (Fig. S5 B). The cut-offs applied previously to the genome-wide data were relatively lenient to maximize the chance of capturing relevant hits in a "one-shot" screening format. Here, we used more stringent gating, which led to the removal of 81 crRNAs that impacted cell viability and morphology based on the range observed with *crLIS1*. Of the remaining 296 crRNA pools, 195 caused a significant change in distribution of at least one cargo (ratio of perinuclear to peripheral signal ±2.5 SD of NTC; Table S5). Our 66% validation rate is comparable to the 50% rate reported for an arrayed screen that used a subset of the same crRNA library to assess delivery of lipid nanoparticle-encapsulated mRNA (Ross-Thriepland et al., 2020).

To better visualize the effects of the 296 crRNA pools on cargo distribution, we grouped them via K-means clustering of their mean effect sizes on localization of peroxisomes, early endosomes, and the Golgi (Fig. 3). crRNAs targeting *AKTIP*, *FAM160A2*, and *HOOK3* were in the same cluster due to selective inhibition of early endosome localization to the perinuclear region, whereas *crPAFAH1B2* was unique in strongly promoting clustering of early endosomes to this location. The observation that targeting *PAFAH1B2* did not increase perinuclear localization of peroxisomes and the Golgi may reflect these cargoes already being tightly clustered at this site.

We also identified clusters of crRNAs that dispersed all three cargoes (Fig. 3). These included cluster 14, which had strong dispersion phenotypes and comprised crRNAs targeting *LIS1* and

three dynein components, as well as five other genes, and cluster 13, which had more modest cargo dispersion and comprised four dynactin components and 16 other genes (see Table S5 for constituents of each cluster). Other clusters contained crRNAs with more selective effects. These included cluster 9, which was characterized by dispersion of peroxisomes and the Golgi but not early endosomes. We conclude from these experiments that the primary screen successfully identified genes that influence the distribution of dynein cargoes, including many that affect a subset of cargo types.

To further evaluate the phenotypes in the secondary screen, we determined the effects of the original shortlist of 377 crRNAs pools on the microtubule cytoskeleton by staining assay plates with γ-Tubulin and α-Tubulin antibodies (Fig. 4, A and B; and Table S6). These reagents mark the MTOC and microtubule network, respectively. γ-Tubulin staining revealed that, compared with the NTC condition, 243 cRNA pools (64.5%) decreased the proportion of cells with one MTOC. Of these pools, 193 (51.2% of the total) increased the proportion of cells with more than one MTOC and 50 (13.3% of the total) increased the proportion with no MTOC. Several of the pools that increased MTOC number targeted known regulators of mitosis, including PLK1, AURKA, and CHMP4B. The crRNAs that impaired MTOC formation included those targeting the centriole protein SAS6 (*crSASS6*), the γ-Tubulin ring complex component GCP4 (*crTUBGCP4*), and subunits of the tubulin chaperone T-complex protein Ring Complex/Chaperone Containing TCP-1 (TRiC/CCT). crRNAs targeting LIS1, the dynein subunits DYNC1H1, DYNC1I2, and the dynactin subunit ACTR1A also lowered MTOC number, in keeping with the role of the motor in delivering components of the centrosome (Young et al., 2000; Jia et al., 2013).

α-Tubulin staining showed that none of the crRNAs caused a strong depletion of microtubules, such as that seen with nocodazole (Fig. 4 C; and Table S6). However, 40 (11%) of the hits were associated with more subtle alterations in morphology of the microtubule network. These included genes encoding several tubulin isotypes and TRiC/CCT components, as well as LIS1, the dynein light chain DYNLL1, and the dynactin component DCTN5. The observation that crRNAs targeting dynein–dynactin can cause α-Tubulin phenotypes is consistent with the motor's ability to modulate interphase microtubule networks (Koonce et al., 1999; Ma et al., 1999; Burakov et al., 2008).

We conclude from these analyses that a subset of crRNA pools influences the microtubule cytoskeleton, including several that target dynein–dynactin components.

## Identification of co-functional genes by unsupervised phenotypic clustering

The above analysis used a small number of features to give a coarse-grain phenotypic assessment of screen hits. Our findings indicated that whereas core dynein–dynactin components affect the localization of multiple cargoes and the organization of the microtubule cytoskeleton, other hits influence a subset of these processes. To systematically classify the hits, we adapted an established image-based profiling workflow (Bray et al., 2016; Caicedo et al., 2017; Chandrasekaran et al., 2021) to generate detailed phenotypic fingerprints for each crRNA pool. These were compared with each other to identify sets of genes that

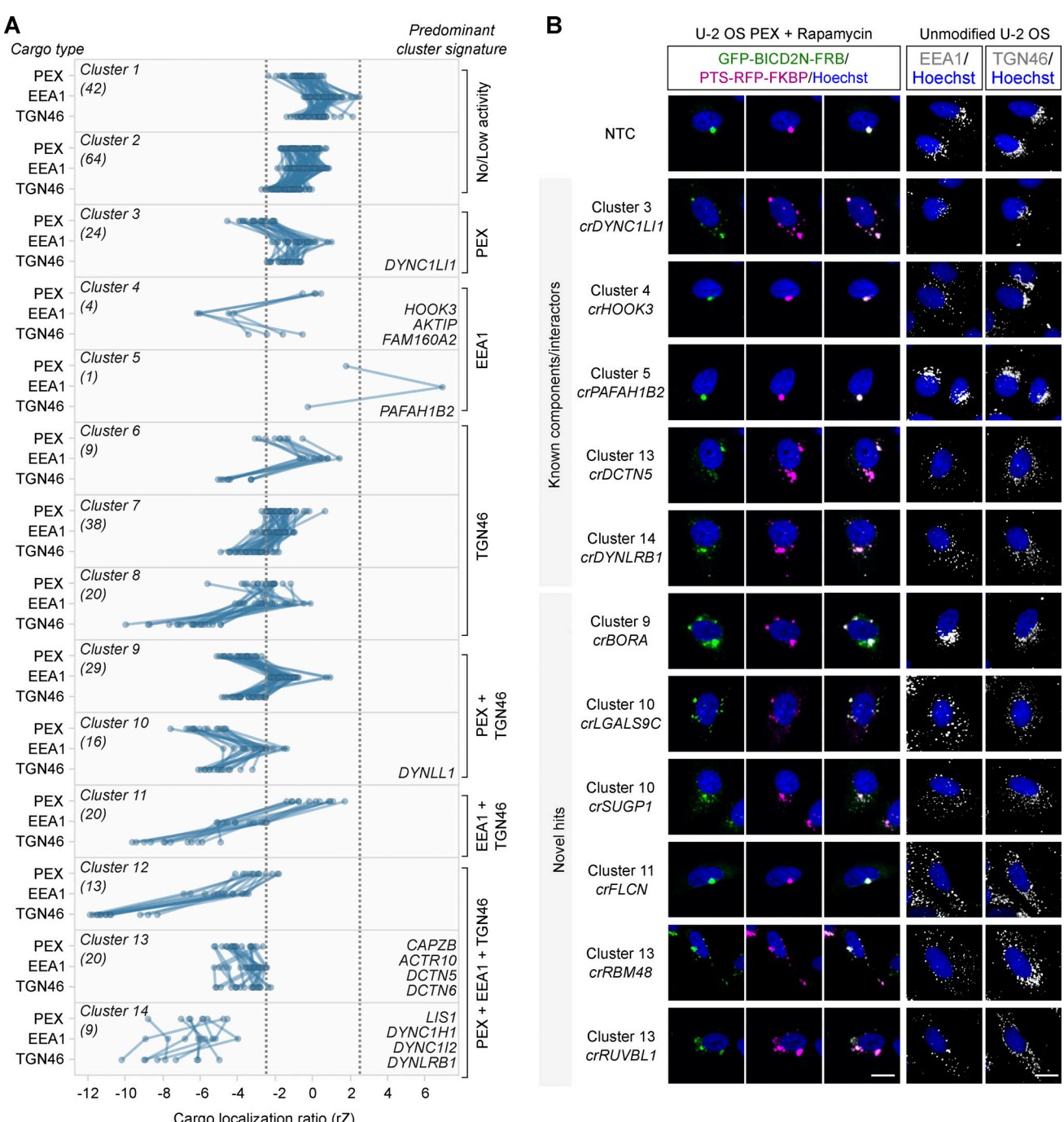

Figure 3. **Categorization of screen hits based on effects on different dynein cargoes. (A)** Grouping of hits based on effects in the secondary screen on localization of peroxisomes (PEX; average of GFP-BICD2N-FRB and PTS-RFP-FKBP), early endosomes (EEA1), and *trans*-Golgi network (TGN46). Cargo localization ratio was calculated by dividing the spot number in the perinuclear region by the spot number in the peripheral region (negative values indicate increased dispersion). Grouping was performed with K-means clustering with Euclidean distance. Data points represent the average rZ (central reference = NTC) from two independent experiments (at least four wells per crRNA per experiment), with each line representing a crRNA pool. Dashed lines show ± 2.5*SD of NTC. Individual clusters are labeled with the number of constituent genes (parentheses), examples of constituent genes, and manual annotation of predominant cargo signature. See Table S5 for the list of genes in each cluster. **(B)** Representative images of cargo localization in cells edited with crRNAs targeting known components and well-characterized interactors of the dynein machinery, as well as novel hits. Third column from the left is a merge of GFP-BICD2N-FRB and PTS-RFP-FKBP signals. Scale bar, 20 µm.

have similar phenotypes and are therefore candidates to function in the same process.

To generate phenotypic fingerprints, we first extracted 2003 quantitative phenotypic parameters related to signals from GFP-BICD2N-FRB, PTS-RFP-FKBP, EEA1, TGN46, α-Tubulin, γ-Tubulin, and Hoechst (Fig. 5 A). The number of features was subsequently reduced to 278 by removing those that were highly variable or redundant via linear regression. Visualizing the

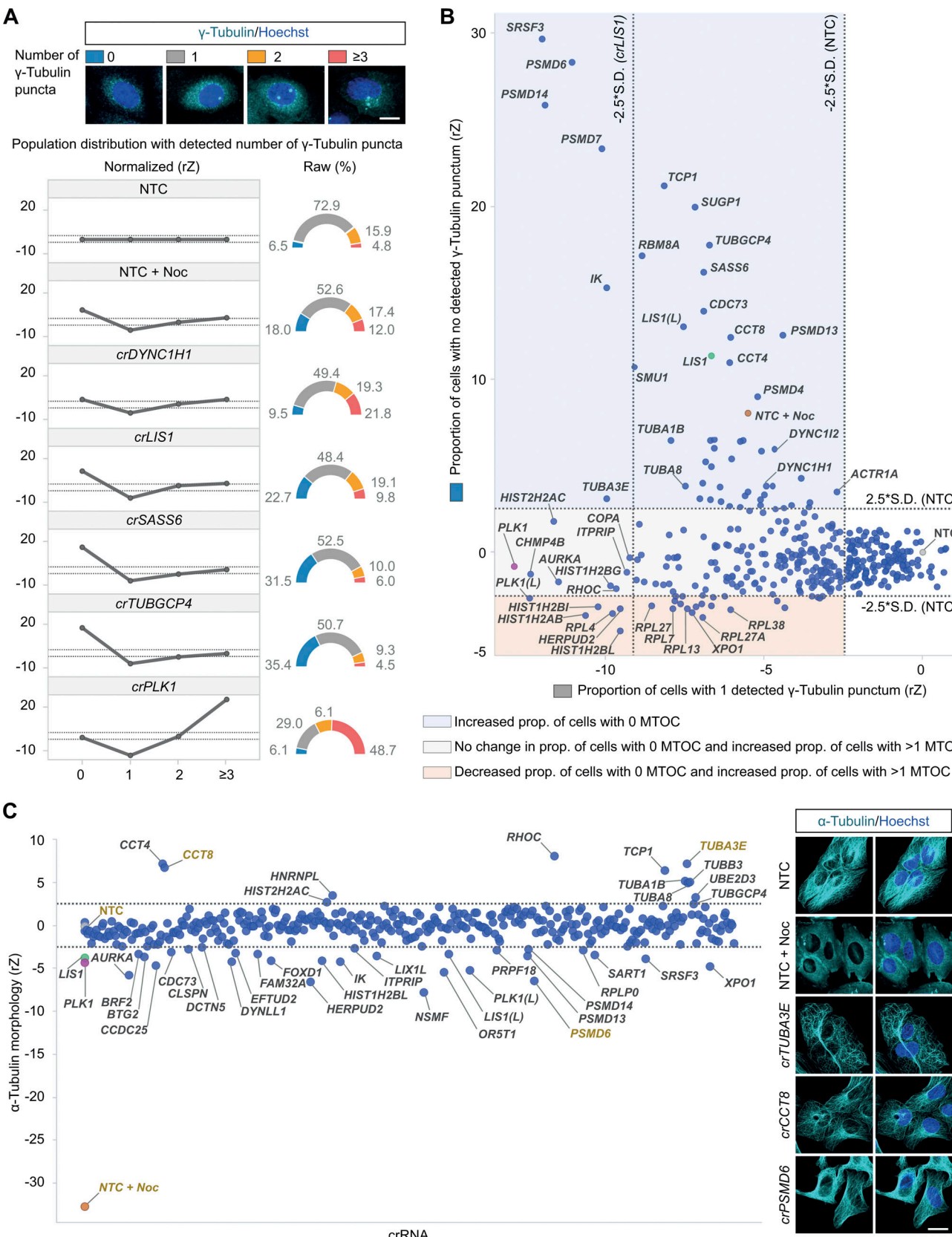

Figure 4. **Analysis of γ-Tubulin and α-Tubulin signals in the secondary screen. (A)** Representative images of unmodified U-2 OS cells with different numbers of detected γ-Tubulin puncta (from population of NTC controls) and distribution of γ-Tubulin puncta numbers in cells treated with nocodazole (Noc) or the indicated crRNA pools. Line charts show normalized data (rZ normalization, central reference = NTC) and pie charts show raw data (percentage of cells). Scale bar, 50 μm. **(B)** Scatterplot of normalized values (rZ normalization, central reference = NTC) for the proportion of cells with 1 versus 0 detected γ-Tubulin

punctum. crRNAs with values to the left of the vertical −2.5*SD (NTC) cut-off line were classed as causing a decrease in the proportion (prop.) of cells with one detected γ-Tubulin punctum (243/377 crRNA pools). This phenotype could be associated with either an increase in the proportion of cells with >1 MTOC (193 crRNAs; values below the horizontal 2.5*SD [NTC] cut-off line; regions shaded in pink and dark pink) or an increase in the proportion of cells with no MTOC (50 crRNAs; values above the horizontal 2.5*SD [NTC] cut-off line; region shaded in blue). An additional threshold based on *crLIS1* controls (−2.5* SD) was introduced for the one punctum feature to categorize crRNAs that yielded a stronger phenotype than *crLIS1*. **(C)** Quantification of α-Tubulin morphology in U-2 OS PEX cells generated by linear discriminant analysis (rZ, central reference = NTC) and representative immunofluorescence images of examples (labeled in gold text in the plot). In B and C, library copies of *crPLK1* and *crLIS1* are labeled with "(*L*)." Scale bar, 25 μm. Data points in A–C represent the mean values of two independent experiments with at least four wells per crRNA per experiment. Hits were defined as exceeding ± 2.5*SD of (A–C) NTC and/or (B) *crLIS1*. See Table S6 for full dataset.

distribution of the high-dimensional phenotypic points on a Uniform Manifold Approximation and Projection (UMAP) plot revealed grouping of genes encoding members of the same protein complexes, such as histones, ribosomal proteins, RNA polymerase II, the RUVBL and TRiC/CCT chaperonins, FAM160A2-AKTIP-HOOK3, and dynein–dynactin (Fig. 5 B; and Table S7). Remarkably, there were even separate groupings of components of the core (A/B) and regulatory (C/D) particles of the proteasome. These observations highlight the utility of our procedures for identifying co-functional genes.

Next, we performed unsupervised hierarchical gene clustering based on highly correlated phenotypic features, as signified by proximity in dendrograms (Fig. S6 [explorable in Data S1] and Table S8). This process revealed clusters related to the same protein complexes highlighted by the UMAP plot. The dynein–dynactin cluster (Fig. 5 C) consisted of neighboring subclusters of genes encoding a subset of dynein (*DYNC1H1, DYNC1I2,* and *DYNLRB1*) and dynactin (*ACTR10, DCTN6, ACTR1A,* and *DCTN5*) constituents. Also present in the dynein–dynactin cluster were *FAM160A2, AKTIP,* and *HOOK3* (which themselves formed a subcluster), *DYNLL1, LIS1,* and *BICD2,* as well as six additional genes. These six genes were *SUGP1* (also known as *SF4*), which encodes an RNA-binding protein with a remarkably similar phenotypic fingerprint to that of the dynein and dynactin components, a grouping of *FLCN, RBM48,* and *TMED2* (encoding a GTPase-activating protein, RNA-binding protein, and transmembrane protein, respectively), as well as *NRF1* (encoding a stress sensor), and *RNF103-CHMP3* (encoding a readthrough product that has sequences from the E3 ligase RNF103 and the multivesicular body component CHMP3), which both grouped with *FAM160A2, AKTIP,* and *HOOK3*.

However, not all components of dynein and dynactin identified in the screen were present in the dynein–dynactin cluster. The phenotypic signatures of *DYNC1LI1* and the dynactin component *CAPZB* were divergent from those of other known dynein–dynactin components, as well as from each other (Fig. S6 and Data S1). This could reflect these factors taking part in a subset of motor functions or having additional, dynein-independent functions that influence cellular organization. Consistent with these ideas, the DYNC1LI1 and DYNC1LI2 light intermediate chains have non-overlapping functions in some dynein-based trafficking events (Kumari et al., 2021) and CAPZB also has a role in capping F-actin (Cooper and Sept, 2008). Nonetheless, our analysis shows that phenotypic clustering is a valuable tool for revealing novel gene associations in our dataset, as well as for highlighting factors that are candidates to work closely with dynein.

**Hit verification with independent crRNAs**

The strong clustering of phenotypic fingerprints for crRNAs that target genes encoding components of the same protein complex strongly suggests that the phenotypes produced by our editing procedures are not driven significantly by off-target effects. To further evaluate the potential for off-target effects in our phenotypic readouts, we targeted a subset of hits from the secondary screen with unrelated crRNA pools. These reagents were selected from the Vienna Bioactivity CRISPR (VBC) collection, which preferentially disrupts functional protein domains while minimizing off-target cutting (Michlits et al., 2020). For these experiments, we selected five of the six additional genes that clustered with dynein–dynactin components, as well as 17 other genes whose targeting caused mislocalization of at least one dynein cargo.

As in the secondary screen, we assessed the effects of the crRNAs on subcellular localization of GFP-BICD2N-FRB and PTS-RFP-FKBP in U-2 OS PEX cells treated with rapamycin, as well as EEA1 and TGN46 in untreated, unmodified U-2 OS cells. We also included a new readout of dynein activity by staining lysosomal compartments in untreated, unmodified cells with an antibody to LAMP1. These structures have previously been shown to rely on dynein for trafficking toward the perinuclear region (Harada et al., 1998; Jordens et al., 2001; Tan et al., 2011), and we corroborated this conclusion with *crDYNC1H1* and *crLIS1* (Fig. S5 A).

Significant effects on cargo localization were confirmed for 19 out of the 22 hits (86.4%) using the VBC crRNA pools (Fig. 6 and Fig. S7, A–C). The lack of activity for at least two of the other three pools (which were directed against *CDH23, FAM86B2,* and *RNF183*) appeared to reflect inefficient cutting of the target gene (Fig. S8, A–C). Overall, however, these experiments revealed a high rate of hit replication. Thus, while we cannot rule out their contribution to a subset of phenotypes, these data provide further evidence that off-target effects do not significantly drive the effects we observe with our pipeline.

The analysis with independent crRNA pools additionally revealed that those targeting *NRF1, SUGP1, FLCN,* and *LGAL9SC* caused dispersion of lysosomal compartments (Fig. 6 and Fig. S7, B and C). Moreover, we found that, in addition to reducing enrichment of GFP-BICD2N-FRB and PTS-RFP-FKBP in the perinuclear region, crRNAs to *CDC5L* and *FAM32A* enhanced clustering of lysosomal compartments at this site. crRNAs to *BORA, INTS2,* and *UVRAG* increased perinuclear clustering of both lysosomal compartments and early endosomes, while causing dispersion of peroxisomes (Fig. 6 and Fig. S7, A–C). These observations raise the possibility of interplay between different dynein-based trafficking processes.

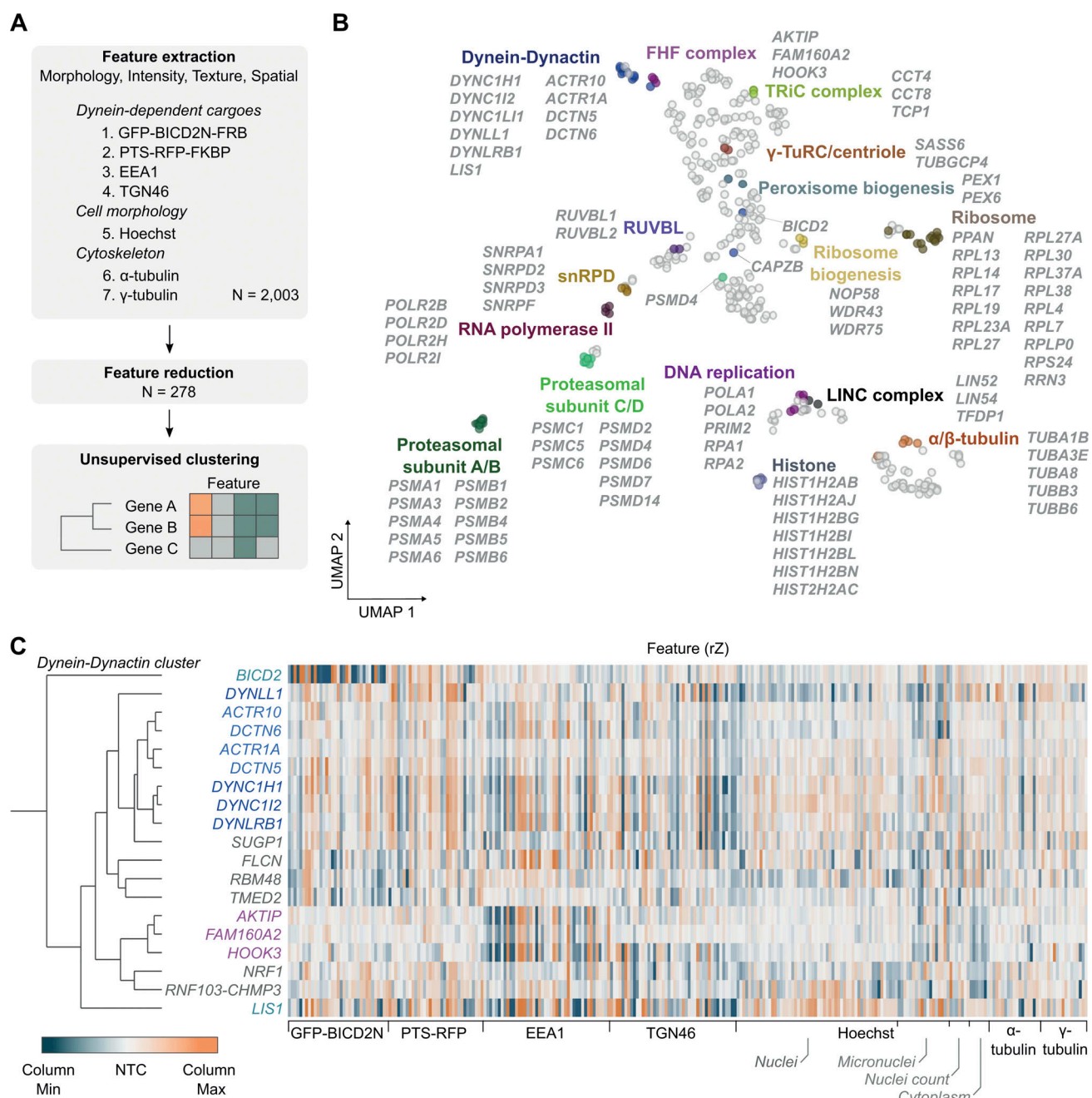

Figure 5. **Unsupervised image-based profiling identifies a functional cluster containing known components of the dynein machinery and novel factors. (A)** Workflow for phenotypic profiling using images collected from the secondary screen. **(B)** UMAP plot for phenotypes of genes selected from the primary screen. The highlighted clusters of genes were manually curated and annotated based on data in the UNIPROT database and primary literature. See Table S7 for source data. **(C)** Phenotypic feature heatmap of the dynein–dynactin gene cluster. Features are grouped in the x-axis according to the marker (see Data S1 and Table S8 for names of individual features). Genes encoding dynein and dynactin components, as well as the associated proteins BICD2 and LIS1, are labeled in different shades of blue. FHF component genes are shown in magenta. Novel genes are labeled in gray. The scales of rZ values (central reference = NTC) were adjusted based on minimum and maximum values of individual features. "Cytoplasm" refers to features associated with the background Hoechst staining in the cytoplasm.

## SUGP1 sustains functional levels of LIS1 mRNA

Finally, we investigated the mode of action of *SUGP1*. Targeting this gene dispersed all cargoes tested and had similar effects to targeting dynein–dynactin components on features of the nucleus and microtubule cytoskeleton (Fig. 5 C). *SUGP1* encodes a 72-kDa protein containing two SURP domains and a G-patch domain

(Fig. 7 A). These motifs are found in many RNA-processing enzymes, leading to annotation of SUGP1 as an RNA-binding protein. However, little is known about the functions of SUGP1, other than roles in SF3B1-associated 3′-splice site recognition during pre-mRNA processing (Sampson and Hewitt, 2003; Alsafadi et al., 2021) and regulation of cholesterol metabolism (Kim et al., 2016).

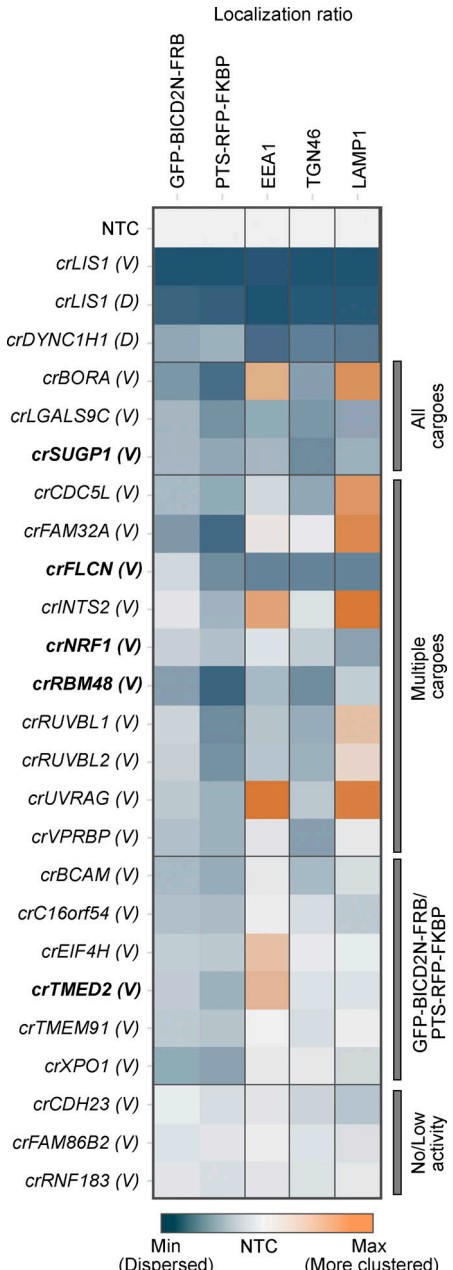

Figure 6. **Confirmation of cargo localization phenotypes with independent crRNAs.** Heatmap displaying localization ratio of dynein cargoes at the perinuclear versus peripheral region of cells treated with indicated crRNAs. GFP-BICD2N-FRB and PTS-RFP-FKBP were evaluated in U-2 OS PEX cells treated with rapamycin, whereas other markers were evaluated in unmodified, untreated U-2 OS cells. crRNAs were synthesized based on the VBC score (labeled "V"), except for LIS1 and DYNC1H1 crRNAs from the initial "Horizon Discovery" set (labeled "D"), which were used as additional positive controls. Bold labeling indicates crRNAs that target novel constituents of the dynein–dynactin cluster previously generated by unsupervised profiling. Color scales of individual features were adjusted based on their minimum and maximum values. Categories of affected cargoes were manually annotated based on statistically significant effects (see Fig. S7). Data represent relative change of mean per cell values aggregated at well level compared with NTC from a minimum of three independent experiments (minimum of 100 cells analyzed per well; four wells analyzed per condition).

We first corroborated the requirement for SUGP1 in dynein-based trafficking by revealing the inhibitory effects of seven out of eight individual crRNAs on both SUGP1 protein expression and peroxisome relocalization (Fig. 7 B and Fig. S8 D). As SUGP1 is enriched in the nucleus (Fig. 7 B) and is a predicted RNA-binding factor, we hypothesized that its depletion affects expression or processing of components of the dynein machinery or its regulators. To test this notion, we performed RNA sequencing (RNA-seq) of polyA-enriched RNAs in U-2 OS cells edited with a single, highly active SUGP1 crRNA (Fig. 7 B and Fig. S8 D). As controls, we processed cells treated with a crRNA from the NTC pool or a crRNA that targets the XCR1 gene. Because XCR1 is exclusively expressed in dendritic cells (Dorner et al., 2009), its targeting should report on any changes in the U-2 OS transcriptome that are induced purely by a double-strand break in DNA. We confirmed that the selected SUGP1 and XCR1 crRNAs edited the target sites in the samples used for RNA-seq (Fig. S8, E–G) and that the XCR1 crRNA had no effect on peroxisome relocalization (Fig. S8 D).

Only a single mRNA, XIRP1, met our criteria for a significant change in abundance in NTC versus crXCR1 samples (minimum absolute $\log_2$ fold change $\geq 0.5$ and false discovery rate [FDR] $\leq 0.05$; Fig. S9 A and Table S9). In contrast, the abundance of 149 mRNAs was significantly altered in SUGP1-edited cells compared with both NTC and crXCR1 controls (Fig. 7 C; Fig. S9, A and B; and Table S9). This list included two mRNAs encoding known components or regulators of the dynein machinery—DYNC1I2 and LIS1—which exhibited, respectively, ~40% and ~35% reductions in abundance compared to the controls. By comparison, SUGP1 mRNA levels were reduced by ~75% relative to the controls. The changes in DYNC1I2 and LIS1 mRNA levels were independently validated by quantitative RT-qPCR (RT-qPCR) in both U-2 OS cells and ARPE-19 cells, including via treatment with another SUGP1 crRNA (Fig. S9 C). Thus, levels of DYNC1I2 and LIS1 mRNA are reduced in SUGP1-edited cells.

Analysis of the RNA-seq data with the rMATs pipeline (Shen et al., 2014) revealed 189 genes with significantly altered splicing patterns in SUGP1-edited cells compared to both controls (Fig. S9, D–F; and Table S10), while the LABRAT package (Goering et al., 2021) showed that 17 genes had 3′-end usage that was affected by SUGP1 targeting (Fig. S9 G and Table S11). However, no known components or regulators of the dynein machinery exhibited altered splicing or 3′-end usage (Tables S10 and S11). Thus, the reductions in DYNC1I2 and LIS1 mRNA levels in SUGP1-edited cells are not associated with changes in splicing or alternative polyadenylation of these transcripts.

We next investigated the consequences of reduced DYNC1I2 and LIS1 mRNA levels in SUGP1-edited cells on the abundance of their protein products using immunofluorescence. DYNC1I2 protein signal was not altered by crSUGP1 (Fig. 7 D), revealing compensation for the lower concentration of its mRNA. In contrast, LIS1 protein signal was significantly reduced in SUGP1-edited cells (29.5 and 38% reduction for two different SUGP1 guides compared with NTC; Fig. 7 D). The deficit in LIS1 protein could be fully restored by transfection of a crRNA-resistant

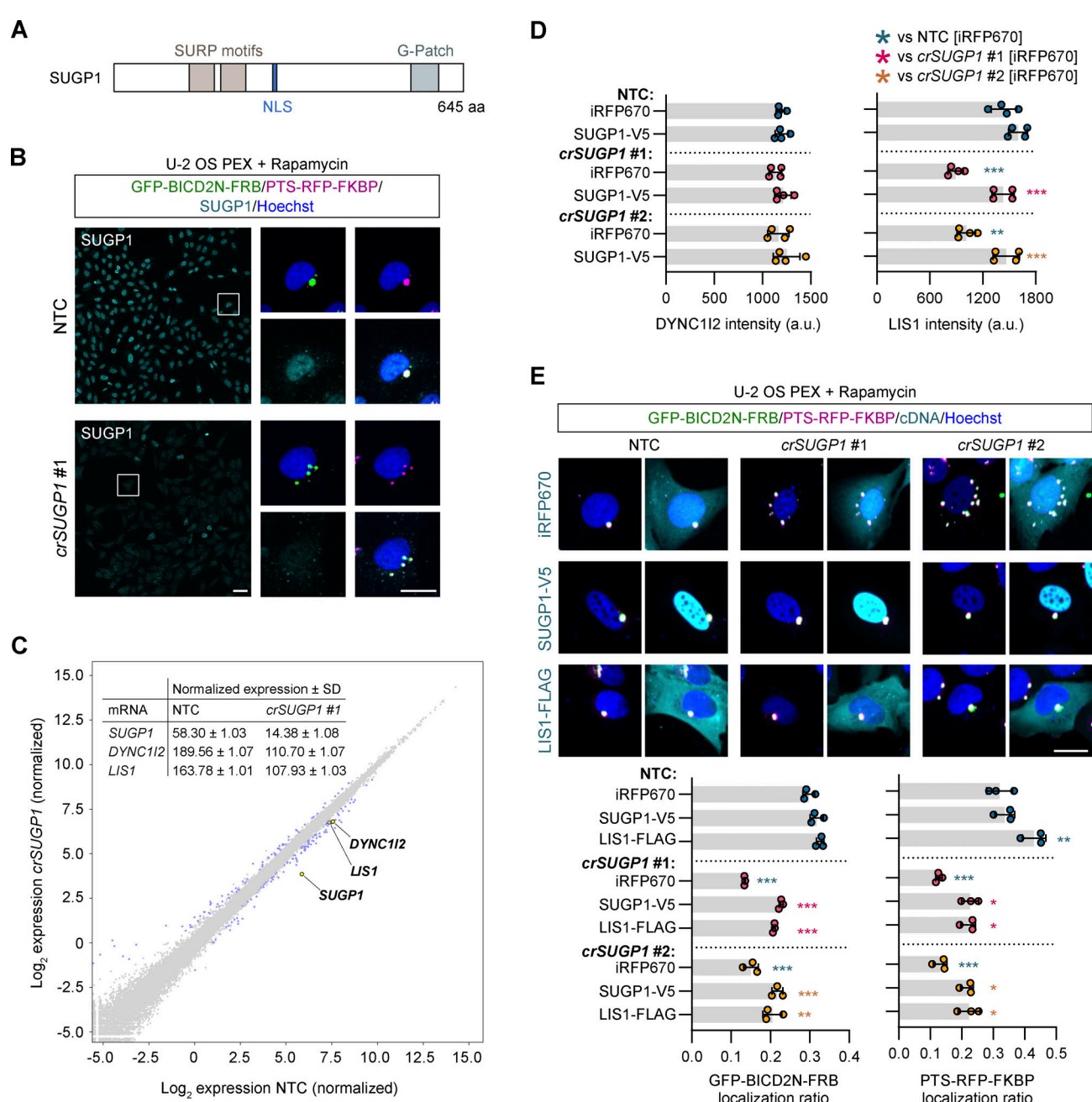

Figure 7. **SUGP1 sustains functional levels of *LIS1* mRNA and protein. (A)** Schematic of SUGP1 domain structure. NLS, nuclear localization signal. **(B)** Representative images of SUGP1 intensity and GFP-BIC2N-FRB and PTS-RFP-FKBP localization in U-2 OS PEX cells treated with *crSUGP1* #1. Scale bar, 25 µm. **(C)** Scatter plot of mRNA abundance for *crSUGP1* #1-edited versus NTC-treated U-2 OS cells (mean log₂ normalized values from three independently performed experiments). mRNAs meeting the threshold for inclusion (minimum absolute log₂ fold change ≥0.5; FDR ≤ 0.05) are labeled in blue, except *SUGP1*, *DYNC1I2*, and *LIS1*, which are labeled in yellow. The inset table shows non-logarithmic values for *SUGP1*, *DYNC1I2*, and *LIS1*. See Table S9 for full results. **(D)** Quantification of endogenous DYNC1I2 and LIS1 protein signal (determined by immunofluorescence) in unmodified U-2 OS cells treated with NTC or *crSUGP1* #1 or #2 and transfected with a control (iRFP670) or crRNA-resistant SUGP1-V5 expression plasmid. **(E)** Representative images and quantification (perinuclear versus peripheral localization ratio) of GFP-BICD2N-FRB and PTS-RFP-FKBP localization in U-2 OS PEX cells treated with NTC or *crSUGP1* #1 or #2 and transfected with a control (iRFP670), crRNA-resistant SUGP1-V5, or LIS1-FLAG expression plasmid. Scale bar, 25 µm. In D and E, cells were transfected with crRNA 96 h before fixation, and with expression plasmid 48 h after crRNA transfection. Data points represent mean per cell intensity values (D) or mean per cell localization ratio values (E) aggregated at well level from four independent experiments (minimum of 100 transfected cells analyzed per well; four wells analyzed per condition). Error bars signify SD. *P < 0.05, **P < 0.01, ***P < 0.001 (two-way ANOVA with Tukey's multiple comparison; colors of asterisks indicate comparison group).

*SUGP1* cDNA (Fig. 7 D). These observations indicate that SUGP1 promotes the expression of LIS1 protein by controlling the level of its mRNA.

Since even a small reduction in LIS1 protein abundance can have significant functional consequences (Reiner et al., 1993; Lo Nigro et al., 1997; Pilz et al., 1998; Cardoso et al., 2002; Hebbar et al., 2008; Dix et al., 2013), we hypothesized that the reduced LIS1 level contributes to dynein trafficking defects in *SUGP1*-edited cells. If this notion were correct, increasing LIS1 levels should suppress the cargo mislocalization phenotype. We therefore transfected *SUGP1*-edited and NTC U-2 OS PEX cells with a *LIS1* cDNA and treated them with rapamycin. In NTC cells, the *LIS1* cDNA elicited no change in GFP-BICD2N-FRB distribution and only a very modest increase in perinuclear clustering of the PTS-RFP-FKBP signal compared to controls in which a cDNA encoding the fluorescent protein iRFP670 was transfected (Fig. 7 E). In contrast, the *LIS1* cDNA significantly boosted perinuclear localization of both GFP-BICD2N-FRB and PTS-RFP-FKBP in *SUGP1*-edited cells (Fig. 7 E). The magnitude of suppression of the mislocalization phenotype was very similar to that observed in *crSUGP1* cells transfected with the crRNA-resistant *SUGP1* cDNA (Fig. 7 E). These data indicate that a key function of SUGP1 in the context of dynein-based trafficking is sustaining *LIS1* mRNA levels.

## Discussion

Current knowledge of dynein regulation is mostly limited to the effects of a small number of binding partners that include dynactin, LIS1, and a handful of activating adaptors. To gain new insights into dynein biology, we performed a genome-wide loss-of-function CRISPR screen using the subcellular distribution of the motor's cargoes as a readout. Our findings represent a valuable resource for mechanistic analysis of dynein-based trafficking, as well as other aspects of cellular organization.

### Leveraging arrayed CRISPR screening

Genome-wide CRISPR screening drives new discoveries by unbiased interrogation of gene function. The vast majority of CRISPR screens are in a pooled format, in which a crRNA library is introduced into a cell population en masse, followed by selection for a phenotype and identification of associated crRNAs by high-throughput sequencing (Bock et al., 2022). While this method has the advantage that crRNA delivery is straightforward, it is typically limited to readouts based on chemical selection or flow cytometry and therefore is not well suited for assessing subcellular phenotypes. To address this issue, innovative pooled screening methods have recently been developed, involving photoconversion of cells with desirable phenotypes followed by cell sorting (Kanfer et al., 2021; Yan et al., 2021), direct sorting of cells based on spatial phenotypes (Schraivogel et al., 2022), and in situ sequencing of crRNAs (Feldman et al., 2019; Wang et al., 2019; Funk et al., 2022). Nevertheless, the arrayed format remains the most direct approach for establishing genotype–phenotype associations and is well-suited for complex imaging-based endpoints (Hultquist et al., 2016; Tan and Martin, 2016; Strezoska et al., 2017; Kim et al., 2018;

O'Shea et al., 2020; Ross-Thriepland et al., 2020). However, this method is costly in terms of upfront investment in arrayed crRNA libraries, automation, and assay development.

We sought to maximize the output of the arrayed screening format by optimizing gene disruption procedures, including through the development of a scalable protocol for efficient mRNA-based delivery of Cas9, and multivariate analysis of the distribution of two dynein cargoes that depend on distinct activating adaptors—BICD2N-bound peroxisomes and HOOK-bound early endosomes. The effectiveness of the screening and analysis pipeline was demonstrated by recovery of core components of the dynein transport machinery, as well as known endosomal adaptors for the motor complex. However, 12 of 23 (52%) dynein–dynactin components were not identified as hits for either cargo. This observation indicates that not all genes that contribute to dynein function were identified in the screen. This is presumably due, at least in part, to inherent limitations of high-throughput genetic screens—functional redundancy, perdurance of long-lived proteins following gene disruption, or some genes not being targeted efficiently or not being expressed in the screened cell type. We cannot, however, rule out that some bona fide regulators of dynein function have subtle phenotypes that might be picked up with novel image analysis tools or replicates of the genome-wide screen. Nonetheless, we validated 195 novel hits from the primary screen, which are candidates for future mechanistic analysis. Identifying biochemical interactors of these proteins may reveal additional players in dynein-based trafficking that were refractory to our CRISPR screening approach.

### Phenotypic clustering reveals novel gene associations

Using morphological profiling, we generated fine-grained phenotypic signatures of hits from the primary screen. This strategy has previously been used to assess cell state (Way et al., 2021), predict biological activity of small molecules (Simm et al., 2018), and identify co-functional genes in functional genomics data (Rohban et al., 2017; de Groot et al., 2018; Duan et al., 2021; Funk et al., 2022). Remarkably, features associated with just seven markers were sufficient to cluster components of known protein complexes, including dynein–dynactin. We also observed clustering of six genes with dynein–dynactin that do not encode components of the core transport machinery. Of these genes, four had not previously been linked physically or functionally with the motor complex. The importance of three of these factors—*RBM48*, *SUGP1*, and *TMED2*—for cargo distribution was confirmed with independent crRNAs. The other two genes that clustered with dynein and dynactin components were *NRF1* and *FLCN*. *NRF1*, targeting of which caused mild peroxisome and lysosome dispersion, encodes a transcriptional factor that can complex with DYNLL1 and DYNLL2 (Herzig et al., 2000). However, it is not known if these interactions are related to the light chains' function in the motor complex or their independent role in assembling protein complexes (Rapali et al., 2011). FLCN is a GTPase-activating protein that promotes lysosomal trafficking by dynein (Cantalupo et al., 2001; Jordens et al., 2001; Johansson et al., 2007; Rocha et al., 2009), potentially by stabilizing the interaction of the activating adaptor RILP with Rab34

(Starling et al., 2016). In addition to lysosomes, early endosomes, and the Golgi were dispersed when *FLCN* was targeted. As RILP function appears to be restricted to late endosomes, lysosomes, and autophagosomes (Cantalupo et al., 2001; Jordens et al., 2001; Khobrekar et al., 2020; Khobrekar and Vallee, 2020), our findings raise the possibility that FLCN has a RILP-independent role in trafficking other dynein cargoes.

While the hits from the dynein–dynactin cluster should be prioritized for mechanistic studies, we anticipate that a subset of other hits will directly influence motor function. This view is supported by our observation that the phenotypic profiles of the dynein light intermediate chain isoform *DYNC1LI1* and the dynactin component *CAPZB* do not cluster with those of other dynein–dynactin subunits. Of the hits outside the dynein–dynactin cluster, *LGALS9C* is a strong candidate to follow up. crRNAs targeting this gene, which codes for a galectin-9 isoform, disrupt localization of all dynein cargoes tested and give a phenotypic profile that groups with that of *CAPZB*.

### Selectivity and interplay during cargo trafficking

We found a number of genes whose targeting altered localization of only a subset of dynein cargoes. Mechanisms that could account for such a phenomenon include modulating the expression, localization, or function of a subset of cargo adaptors, or regulating the activity of dynein only when it is engaged with certain cargoes. We also found that disrupting some genes increases perinuclear clustering of at least one type of dynein cargo, while dispersing at least one other. These factors might influence the recruitment of limiting components of the transport machinery to one or more cargo types, thereby affecting their availability for others. One gene in this category is *INTS2*, which encodes a nuclear protein that is best known as a mediator of small nuclear RNA processing (Baillat et al., 2005) but also promotes dynein recruitment to the cytoplasmic face of the nuclear envelope (Jodoin et al., 2013). We observed increased perinuclear clustering of endosomes and lysosomes when *INTS2* was targeted, which could conceivably arise from dyneins that otherwise would be located at the nuclear surface becoming accessible for other cargoes. The implication of these ideas is that activities of different dynein-driven processes are finely balanced.

### SUGP1 sustains functional levels of LIS1

To further demonstrate the utility of our screen, we investigated the mode of action of the RNA-binding protein SUGP1. This factor has a phenotypic signature that is very similar to that of core components of the dynein transport machinery. We found that SUGP1 disruption lowers *LIS1* mRNA and protein concentration and that restoring LIS1 protein levels suppresses the associated defects in cargo localization. While we cannot rule out additional roles of SUGP1 in dynein-based trafficking, our experiments suggest a key function of this protein is promoting expression of LIS1. Although the reduction of LIS1 protein in SUGP1-edited cells was modest, lowering LIS1 abundance by a similar amount is sufficient to impair dynein-mediated trafficking and cause neurodevelopmental defects in animal models (Gambello et al., 2003; Dix et al., 2013). In keeping with these

findings, loss of one copy of the *LIS1* gene causes severe brain malformation in humans (Reiner et al., 1993; Lo Nigro et al., 1997; Pilz et al., 1998; Cardoso et al., 2002). As the splicing and 3′-end usage of *LIS1* mRNA was not altered by disruption of SUGP1, the protein presumably plays a role in transcription or stabilization of *LIS1* mRNA or affects processing of mRNAs whose products control these events. Further investigation of the molecular mechanism by which SUGP1 promotes *LIS1* expression could shed light on post-transcriptional regulation of dynein-based trafficking, which is a largely unexplored topic.

### A resource for shedding light on other aspects of cellular organization

While our main objective was to reveal genetic requirements for dynein-based trafficking, we expect that our dataset will be valuable for those interested in several other aspects of cell biology. Our primary screen additionally highlighted crRNAs that affect the morphology of early endosomal compartments and peroxisomes. We also identified novel factors that influence nuclear morphology, formation or micronuclei, organization of the microtubule network, or MTOC number. The unbiased phenotypic clustering of hits in the secondary screen additionally revealed many novel gene associations, including those involving known regulators of gene expression and protein homeostasis. Exploring these associations is likely to provide new insights into the molecular control of these processes. Moreover, the primary screen images can be mined to extract additional phenotypes and identify genes that affect them. To facilitate this process, the images are available through AstraZeneca's Open Innovation Platform (https://openinnovation.astrazeneca.com/preclinical-research/preclinical-data.html).

## Materials and methods

### Cell culture

The U-2 OS human bone osteosarcoma cell line stably expressing GFP-BICD2N-FRB and PTS-RFP-FKBP was described previously (Vincent et al., 2020) (Research Resource Identifier [RRID]: CVCL_D4XX). U-2 OS (RRID: CVCL_0042) HEK-293 (RRID: CVCL_0045), and IMR-90 (RRID: CVCL_0347) cells were maintained in McCoys 5A, DMEM, and Eagle's MEM, respectively, whereas ARPE-19 (RRID: CVCL_0145) and SH-SY5Y (RRID: CVCL_0019) cells were cultured in DMEM/Nutrient Mixture F12 Ham. All media was purchased from Sigma-Aldrich and was supplemented with 10% (vol/vol) FBS and 1% (vol/vol) GlutaMAX (Thermo Fisher Scientific). All cell lines were certified free of Mycoplasma either internally using the MycoSEQ Mycoplasma detection kit (Thermo Fisher Scientific) or by IDEXX BioAnalytics using STAT-Myco testing. The identities of cell lines were authenticated by short tandem repeat fingerprinting by IDEXX BioAnalytics.

### mRNA-Cas9 transfection

For optimization of mRNA delivery, cells were seeded at ~70% confluency and reverse transfected with indicated concentrations of mRNA-Cas9-HA containing 5-methoxyuridine (5moU) or mRNA-eGFP 5moU (TriLink) in MessengerMAX

(Thermo Fisher Scientific) or RNAiMAX (Thermo Fisher Scientific). Samples were fixed either 6 or 24 h after transfection and processed for immunofluorescence. Cas9 expression was monitored by the intensity of the HA signal, which was quantified using Columbus 2.9.1 (Perkin Elmer). Unless stated otherwise, the concentration of mRNA-Cas9-HA was fixed at 40 ng per well of a 384-well plate. Cells were reverse transfected with mRNA-Cas9-HA coupled with 1% (vol/vol) MessengerMAX in OptiMEM for 6 h prior to crRNA transfection.

## Synthetic crRNA preparation, transfection, and automated liquid handling

Synthetic two-part crRNA was used for all experiments. The whole-genome human crRNA library (GC-006500-E2; Horizon Discovery) was designed using Horizon Discovery's Edit-R algorithm that maximizes functional protein knockout while minimizing off-target cutting by considering gaps and bulges between the crRNA and genome, as well as the position of mismatches (i.e., in the non-seed or seed positions [more or less likely, respectively, to be tolerated by Cas9]) when ranking crRNAs for specificity. None of the crRNAs selected for the library has a perfect match in the genome other than the target site, and only a very small proportion (4.78%) have a single mismatch or gap in the non-seed region versus another sequence (a situation necessitated because design space is limited for their targets). The library was prepared with pools of four equimolar crRNAs targeting each gene, as described previously (Ross-Thriepland et al., 2020), as we found in pilot experiments that the use of only one crRNA was often not sufficient to disrupt the function of the target gene in the majority of cells. Controls included in the genome screen were: NTC (a pool of four pre-designed crRNAs [Horizon Discovery] with at least three mismatches to potential protospacer adjacent motif (PAM)–neighboring targets in the human genome; 38 wells per plate) as the neutral control; crLIS1 (a pool of four crRNAs targeting LIS1 [Horizon Discovery]; 13 wells per plate) as the positive control, and crPLK1 (a pool of four crRNAs targeting PLK1 [Horizon Discovery]; 13 wells per plate) as the editing control. For hit confirmation, identical pools of crRNAs (n = 377 genes) and/or a new pool of four crRNAs per gene selected based on the VBC score (Michlits et al., 2020) (n = 22 genes) were custom synthesized by Horizon Discovery (Table S12). The VBC guides (Table S12) were designed based on off-target prediction scores against all exons with any mismatches weighted by their position in the crRNA sequence relative to the PAM (Michlits et al., 2020). All the sequences had zero predicted off-targets, with the exception of the crLGALS9C crRNAs, which were predicted to potentially target closely related LGALS9 genes. In the secondary screens, individual NTC crRNAs (NTC 1–4), the crLIS1 pool, and crPLK1 pool were included in the synthesis of the library plate as additional controls. These were used for quality control of individually synthesized batches of crRNAs via assessment of their performance in the phenotypic assays. crRNAs were dispensed into 384-well Phenoplates (Perkin Elmer) via an Echo 555 instrument (Labcyte) at a final concentration of 50 nM. For the genome-wide screen, automatic liquid handling was executed by docking a Steristore (HighRes), Echo 555, Multidrop combi

(Thermo Fisher Scientific), and washer dispenser EL406 (Biotek) to a Star6 automation platform (HighRes). Library and assay plates were stored in the Steristore at 8°C during dispensing and were brought to room temperature before reverse transfection. Cas9-transfected U-2 OS cells (~1,500 cells per well of the 384-well plate) were reverse transfected with crRNAs with 1% (vol/vol) RNAiMAX (Thermo Fisher Scientific) in serum-free media dispensed via a Multidrop combi. Plates were incubated for 72 h before rapamycin treatment, fixation, and immunostaining (all via automatic liquid handling).

## Drug treatment

To induce heterodimerization of GFP-BICD2N-FRB and PTS-RFP-FKBP, cells were treated with 2 nM (unless indicated otherwise) rapamycin (SelleckChem) in serum-free media via a Multidrop combi for 2.5 h before fixation. Nocodazole (3 μM; Sigma-Aldrich) and DMSO as vehicle control were added via a Tecan D3000 dispenser.

## Immunostaining

Cells were fixed in 4% (wt/vol) paraformaldehyde containing 0.04 mg/ml (wt/vol) Hoechst 33342 (Thermo Fisher Scientific) for 20 min, followed by permeabilization and blocking in 5% (wt/vol) bovine serum albumin (BSA) and 0.25% (vol/vol) Triton X-100 in phosphate-buffered saline (PBS) for 1 h. Cells were then incubated in 1% (wt/vol) BSA with primary antibodies at 4°C overnight followed by secondary antibodies at room temperature for 2 h (see Table S13 for details of primary and secondary antibodies [including catalog numbers and RRID portal numbers] and their pairings). All washes were performed with PBS on a Washer Dispenser EL406 (Biotek). For the secondary screen, U-2 OS PEX cells were stained with α-Tubulin antibodies and Hoechst (in addition to directly visualizing GFP-BICD2N-FRB and PTS-RFP-FKBP), whereas unmodified U-2 OS cells were stained with TGN46, EEA1, and γ-Tubulin antibodies, as well as Hoechst.

## High-content imaging, feature calculations, and data normalization

Image acquisition was performed at room temperature on the CellVoyager 8000 imaging system (Yokogawa) with either a 20× or 40× water immersion objective (1.0 and 0.95 NA, respectively) with a minimum of four fields of view per well. The imaging medium was PBS. Images were acquired as complete Z-stacks and were post-processed to generate maximum projections. 2 × 2 binning was performed on the images acquired for the genome screen to reduce file size. All analysis was performed on binned 20× images except for the analysis of α-Tubulin in the secondary screens, which was performed on unbinned 40× images.

Cellular segmentation and image analysis were performed using Columbus 2.9.1. Details of the workflow for the genome-wide screen are provided in Data S2. To summarize, nuclei were detected by virtue of strong Hoechst signal, whereas cytoplasm was detected using outlines of the α-Tubulin antibody signal (during assay development and secondary screens, the outline of the weak Hoechst background signal was used to define the

cytoplasmic region in instances when α-Tubulin staining was not used). Nuclei were subsequently filtered based on cellular morphology, and those with cell boundaries located at the image boundary were excluded. Cell debris and cells that were not viable or undergoing mitosis were excluded based on cut-offs for nuclear morphology, and size of the nucleus and cytoplasm. The cytoplasmic area was then resized to include a region overlaying the periphery of the nucleus as this is where the MTOC, and therefore the target of dynein-dependent cargo movement, is located in many cells. For the peroxisome relocalization assay, cells were further selected for GFP- and RFP-positive cells based on their intensity profiles. Calculations of morphological features or the number of spots were performed with pre-existing Columbus algorithms, with parameters optimized using a series of images of U-2 OS cells treated with NTC crRNAs, crLIS1, or nocodazole. This involved tuning parameters including "detection sensitivity," which defines how intense a spot must be for detection, and "splitting coefficient," which determines if adjacent spots are split or merged, and evaluating the proportion of real spots that were identified by visual inspection of the results. In the case of RFP spot detection, a further background subtraction step (see Data S2) was applied to ensure that the desired spots were captured. To determine the number of spots in different regions of the cytoplasm, cells were segmented into three rings—the perinuclear region (with an outer border of –7 μm and inner border of 7 μm of the resized cytoplasm), the intermediate region (with a distance of 7–14 μm from the resized cytoplasm), and the outer region (with a distance of 14–35 μm [based on the maximum cytoplasm size, relative to the nuclear envelope, for NTC U-2 OS cells])—for spot detection. An absolute cut-off of 7 μm was selected for the inner ring as it gave lower variance in pilot studies using the images of U-2 OS cells treated with NTC crRNAs, crLIS1, or nocodazole than using different absolute cut-off values or cut-offs based on the percentage of total cell area. Textural features of nuclear, GFP, and RFP signals were also captured, as was the intensity of the α-Tubulin signal. For early endosome signals, textural, morphological, and intensity features in the cell were captured and the cytoplasm was resized into two compartments based on the percentage of cell area—an inner ring containing the nucleus and perinuclear region and a peripheral ring comprising the region between the outer border of the inner ring and just beyond the edge of the cell boundary (to capture any endosomes in protrusions). The cut-offs for the rings were decided based on robust performance in pilot studies with cells treated with NTC or crLIS1 crRNAs, or nocodazole. The localization ratio for early endosomes was calculated by dividing the number of EEA1 spots (identified using settings optimized with the positive and negative control image sets, as described for peroxisome signals) in the inner region by the number in the peripheral region. To identify cells with one or more micronuclei, a more lenient gating for removing cell debris, and mitotic and inviable cells was first applied to the images. Subsequently, micronuclei were identified with the "Find micronuclei" routine in Columbus, which identifies a small region in the cell that has a higher Hoechst intensity than its surroundings and is outside the defined nucleus. Subsequently, features related to micronucleus number, size, intensity, and positioning were captured.

In the follow-up experiments, localization ratios for cargoes were determined using methods described above for EEA1, with spot detection parameters adjusted to effectively capture TGN46 and LAMP1 signals. The exception was for calculation of peroxisome localization in the secondary screen, in which the perinuclear region had an inner border of –7 μm and an outer border of 7 μm of the resized cytoplasm, and the peripheral region had an inner border of 7 μm and an outer border of 35 μm of the resized cytoplasm. Linear discriminant analysis (LDA) was then applied to both mean GFP and RFP localization ratios per well.

Data normalization and transformation including LDA were performed with Genedata Screener 19.0.1 (Genedata). LDA was used as a linear classifier by combining multiple individual features that have an rZ′ ≥ 0.1 to facilitate separation of the NTC versus a positive control and/or to generate a single feature for hit calling. LDA was also used for morphological and/or textural analysis when no individual feature was sufficient to define the phenotype of interest. Either one-point normalization (rZ score with NTC as a central reference) or two-point normalization (NTC minus positive control; 0–100) was used. For the genome-wide screen, genes were initially filtered based on the number of cells, cellular morphology, and micronuclei before hit calling. Thresholds were based on mean ± SD and were assessed and adjusted based on the desired stringency (at least ±2*SD) compared with the specific control (e.g., NTC) for individual features. The rationale for using a range of thresholds rather than a fixed threshold in the genome-wide screen was to take into consideration the limitations of the one-shot screening (i.e., with no replicates for almost all crRNA pools), as well as the number of hits that we had the capacity to take forward to the secondary screen. For the subsequent hit validation activities, the increased number of technical and biological replicates provided more confidence when hit calling and, therefore, we standardized on a minimum threshold of at least ±2.5* SD of NTC. Thresholds used in specific instances are given in the figure legends.

## Image-based profiling

Mean profiles of features ($n$ = 2003) were generated with the pre-existing algorithms in Columbus 2.9.1. Data were transformed into rZ scores with NTC as a central reference and were aggregated by median per crRNA pool. Feature reduction was performed by removing highly variable features (defined by SD ≥3 of crLIS1 controls that were scattered around the 384-well plates) and highly correlated (i.e., redundant) features ($R^2$ ≥ 0.8). The remaining features ($n$ = 278) were subjected to hierarchical clustering to group crRNAs using complete linkage and correlation for distance measures with a normalization of scaling between 0 and 1. Genes encoding proteins that function in the same protein complexes or processes were assigned manually following inspection of the UNIPROT database (RRID: SCR_002380) and primary literature.

## Molecular cloning and DNA transfection

The cDNA sequence for human SUGP1 (based on RefSeq: NM_172231) fused with a C-terminal V5 epitope tag was

synthesized and cloned into the KpnI and XbaI restriction sites in pcDNA3.1(+) by Azenta Biosciences. The synthesized *SUGP1* sequences had synonymous mutations in the PAM or target site sequences to prevent cutting of the cDNA by *SUGP1* crRNAs in rescue experiments. The expression construct for human LIS1 (RefSeq: NM_000430) fused with a C-terminal FLAG epitope tag was obtained from OriGene. An iRFP670 expression plasmid (synthesized by Azenta Biosciences) in the pcDNA3.1 (+) backbone was used as a transfection control. Cells were transfected with endotoxin-free plasmid DNA 48 h after crRNA transfection and incubated for an additional 48 h before fixation. DNA transfection was performed with Lipofectamine 3000 (Thermo Fisher Scientific) according to the manufacturer's instructions (for each well of a 384-well plate: 6 ng plasmid DNA, 1% [vol/vol] Lipofectamine 3000, 0.5% [vol/vol] p3000 reagent). Transfected cells were selected based on the intensity profile of either iRFP670, V5, or FLAG.

### Tracking of indels by decomposition (TIDE)

Genomic DNA was isolated from cells collected from 96-well plates using Direct PCR lysis reagent (Viagen) supplemented with 1 mg/ml (wt/vol) Proteinase K (Sigma-Aldrich) according to the manufacturer's instructions. PCR reactions were carried out with Phusion Flash PCR Master Mix (Thermo Fisher Scientific), again as instructed by the manufacturer. PCR products were purified using the QIAquick kit (Qiagen), and capillary sequencing was performed by Azenta Biosciences. Sequencing traces from cells treated with crRNAs targeting the gene of interest or from NTC controls were analyzed using the standard parameters of the TIDE webtool (RRID: SCR_023704) (Brinkman et al., 2014).

### RNA-seq and bioinformatics

Approximately $1 \times 10^6$ U-2 OS cells per sample were harvested from an individual T25 flask and processed for RNA extraction and sequencing. The samples were from three independent experiments that each included U-2 OS cells transfected with predesigned commercially available NTC guide 1 (targeting 5′-GATACGTCGGTACCGGACCG-3′), *crSUGP1* guide 1 (targeting 5′-TACTTGTACCCTTGGCTATT-3′), or *crXCR1* guide 1 (targeting 5′-GTGTTTCTCCTCAGCCTAGT-3′) (Horizon Discovery). To confirm editing by the guides, DNA from a proportion of cells from each sample pool was purified and analyzed using TIDE, as described above. To confirm depletion of SUGP1 protein specifically in *crSUGP1*-treated cells, a proportion of each sample pool was reseeded at high density in a 384-well plate and incubated for 24 h before fixation and processing for immunostaining with α-SUGP1 antibodies.

Quality control on the extracted RNA was performed with a Qubit 4.0 Fluorometer (Thermo Fisher Scientific) and Agilent 3500 Fragment Analyzer (Agilent). Subsequent steps of RNA-seq were outsourced to Azenta, which generated sequencing libraries from polyA-enriched RNA (captured with oligo-dT beads) using the NEBNext Ultra II RNA library prep kit for Illumina (NEB), multiplexed them, and loaded them on an Illumina NovaSeq 6000 machine. Samples were sequenced using a 2 × 150 pair-end configuration v1.5, followed by removal of adaptor sequences and poor-quality sequences with the Illumina bcl2fastq program (version 2.20; RRID: SCR_015058).

We validated the overall quality of the sequencing data using FastQC (version 0.11.5; https://www.bioinformatics.babraham.ac.uk/projects/fastqc/; RRID: SCR_014583) and determined the absence of potential sources of contamination using the FastQ Screen tool (version 0.14.1; RRID: SCR_000141) (Wingett and Andrews, 2018). We then performed further quality trimming of the sequences using Trim Galore! (version 0.6.7; https://www.bioinformatics.babraham.ac.uk/projects/fastqc/; RRID: SCR_011847) and its dependency Cutadapt (version 2.4; RRID: SCR_011841) (Martin, 2011) in paired-end mode.

The FASTQ reads were mapped to the human reference genome GRCh38 (version 102) using the STAR aligner (version 2.7.9a; RRID: SCR_004463) (Dobin et al., 2013) that was distributed with the transcript splicing analysis software rMATS (RRID: SCR_023485) (Shen et al., 2014) inside a Docker container (made available on Docker Hub by the Xing group [https://hub.docker.com/r/xinglab/rmats]). The primary alignments of the mapped data (in the form of BAM files) were imported into the genome browser SeqMonk v1.48.0 (https://www.bioinformatics.babraham.ac.uk/projects/seqmonk/; RRID: SCR_001913). Following data import, genes exhibiting an appreciable level of expression were identified using the *RNA-seq Quantitation Pipeline* in SeqMonk. The pipeline counted the number of read pairs mapping to exons for every gene. These gene-level expression values were normalized by dividing by the total number of mapped read pairs per sample. The results were then $\log_2$ transformed. Genes with an expression score of greater than −2 in at least one of the nine samples were selected for differential expression analysis. The number of read pairs mapping to each exon was determined again using SeqMonk's *RNA-seq Quantitation Pipeline* but, on this occasion, the raw counts were recorded and no subsequent normalization and $\log_2$ transformation steps were performed. The raw counts were used to identify differentially expressed genes using DESeq2 (version 1.32.0; RRID: SCR_015687) (Love et al., 2014), which was launched from SeqMonk using default settings. Only the count data from genes previously shown to exhibit an appreciable level of expression in one of the datasets were passed to DEseq2 for analysis. DESeq2 was run using R (version 4.1.0 [Camp Pontanezen] Patched 2021-06-08 r80465; RRID: SCR_001905). Differentially expressed genes were defined as having an adjusted P-value ≤0.05 after multiple testing correction and an absolute $\log_2$ fold change ≥0.5.

We converted the rMATs Docker container (v4.1.2) into a Singularity container (using Singularity version 3.8.0), and from within that container executed the rMATS Python script *rmats.py* to perform the STAR mapping and subsequent splicing analysis. The samples were mapped in paired-end mode using the default *rmats.py* settings. We also specified the *rmats.py* option *readLength* to optimize the software for sequence reads of 150 bp in length. In addition, the flag variable-read-length allowed the processing of reads of varying lengths, which was necessary owing to the quality trimming prior to rMATs analysis. This process produced pairwise splicing comparisons between the different conditions (NTC versus *crSUGP1*, *crXCR1* versus *crSUGP1*, and NTC versus *crXCR1*).

Alternative polyadenylation for samples processed for RNA-seq (NTC, *crXCR1*, and *crSUGP1*) was quantified using LABRAT v0.3.0 (RRID: SCR_025006) (Goering et al., 2021). LABRAT performs this analysis by quantifying the relative abundances of the last two exons of all transcript isoforms using the transcript quantification tool Salmon (Patro et al., 2017). Genes whose total abundance across all isoforms was less than five transcripts per million in any sample were excluded from the analysis. FDR values were calculated using LABRAT's linear mixed effects model followed by multiple hypothesis correction. Isoform structures and transcript sequences were derived from GENCODE v28 (RRID: SCR_014966). Genes with alternative polyadenylation were defined as having $\Delta\psi \geq 0.05$ (Goering et al., 2021) and FDR $\leq 0.05$.

### Taqman real-time PCR assay
RNA was extracted using the RNeasy mini kit (Qiagen) according to the manufacturer's instructions. QuantiTect reverse transcriptase (Qiagen) was used for cDNA synthesis with an input of 800 ng RNA per sample, with the product diluted 1:4 in distilled water prior to Taqman assays. The reaction mix included 1X Taqman real-time PCR master mix, 1X Taqman probe(s), and cDNA template. Predesigned Taqman probes (Thermo Fisher Scientific) were used for *LIS1* (Hs00181182_m1; FAM-MGB), *DYNC1I2* (Hs00909737_g1; FAM-MGB), and *B2M* (Hs00187842_m1; VIC-MGB) as a housekeeping control. Real-time PCR was performed in MicroAmp optical 384-well plates using the QuantStudio 6 Flex RT-PCR system (Thermo Fisher Scientific). The default thermocycling program was used (95°C for 10 min, followed by 40 cycles of 95°C for 15 s and 60°C for 1 min). Data were analyzed using the Quantstudio real-time PCR system software (Thermo Fisher Scientific). Relative quantification values were obtained by normalizing against the expression level of NTC samples.

### Statistics
Statistics analyses were performed with Prism 9.3.1 (Graphpad; RRID:SCR_002798). Appropriate statistical tests were selected based on data distributions (Gaussian or non-Gaussian), variance (equal or unequal), sample size, and number of comparisons (pairwise or multiple). Gaussian versus non-Gaussian distributions were determined in Prism 9.3.1 with Anderson-Darling, D'Agostino-Pearson, Shapiro–Wilk, and Kolmogorov–Smirnov normality tests; a failure of one or more of these tests to determine that the data were Gaussian resulted in a nonparametric test being used. Details of statistical tests applied for each dataset are provided in the figure legends.

### Online supplemental material
Figs. S1 and S2 show optimization of gene disruption and peroxisome relocalization procedures. Fig. S3 shows successful scaling of gene disruption procedures. Fig. S4 shows metrics for cell viability, micronuclei, and peroxisome morphology from the genome-wide screen. Fig. S5 shows dynein-dependent Golgi and lysosomal compartment localization, and reproducibility of data in secondary screen biological replicates. Fig. S6 shows an overview of phenotypic fingerprint clustering. Fig. S7 shows additional data on cargo localization in the independent crRNA screen. Fig. S8 shows additional data from *RNF183*-, *FAM86B2*-, and *SUGP1*-edited cells. Fig. S9 shows additional analysis of RNA-seq data from *SUGP1*-edited and control cells. Table S1 shows features and data acquired in the genome-wide screen. Table S2 lists hits identified using different features of cellular organization. Table S3 shows components of the dynein machinery identified with different assay endpoints. Table S4 lists genes taken forward to the secondary screen and associated Gene Ontology (GO) terms. Table S5 shows K-means clusters and cargo localization scores from the secondary screen. Table S6 shows γ-Tubulin and α-Tubulin datasets from the secondary screen. Table S7 shows the source data for the UMAP plot in Fig. 5 B. Table S8 shows clustered phenotypic fingerprints. Table S9 shows differentially expressed genes in *SUGP1*-edited and control cells. Table S10 shows differential splicing events in *SUGP1*-edited and control cells. Table S11 shows differential 3′-end usage events in *SUGP1*-edited and control cells. Table S12 shows sequences of secondary screen and VBC-based crRNAs. Table S13 shows antibody pairs used for immunofluorescence. Data S1 shows a high-resolution, explorable version of the phenotypic heatmap. Data S2 shows image acquisition and analysis workflow for genome-wide CRISPR screen.

### Data availability
RNA-seq data has been uploaded to Gene Expression Omnibus (accession number: GSE218249). Raw images from the primary screen are available upon request via AstraZeneca's Open Innovation Platform (https://openinnovation.astrazeneca.com/preclinical-research/preclinical-data.html). Other data are included in the manuscript or supplement or are available from S.L. Bullock upon reasonable request.

## Acknowledgments
We thank members of the Bullock Group at Medical Research Council (MRC) Laboratory of Molecular Biology (LMB), and the Functional Genomics and Open Innovation groups at AstraZeneca for advice, support, and sharing resources; we are particularly grateful to Davide Gianni, Ceri Wiggins, Samantha Peel, James Pilling, Martin Booth, Roger Clark, and Paul Twyman at AstraZeneca. We also thank Llywelyn Griffith (Francis Crick Institute) and Mariann Bienz (LMB) for advice on splicing analysis and feedback on the manuscript, respectively.

The work was supported by the LMB-AstraZeneca BlueSky Fund (BSF31), the UK Medical Research Council (file reference numbers MC_U105178790 [to S.L. Bullock] and MC_UP_1201/9 [to Madeline Lancaster, who supports S.W. Wingett]), and the National Institutes of Health (R35-GM133385 [to J.M. Taliaferro]). Open Access funding provided by MRC Laboratory of Molecular Biology.

Author contributions: S.L. Bullock and D. Ross-Thriepland conceived the project. C.H. Wong, D. Ross-Thriepland, and S.L. Bullock designed experiments. C.H. Wong performed experiments. C.H. Wong, S.W. Wingett, C. Qian, M. Hunter, and J.M. Taliaferro analyzed data. C.H. Wong and S.L. Bullock prepared the manuscript, which was edited and approved by all other authors.

Disclosures: C. Qian, M. Hunter, and D. Ross-Thriepland reported personal fees from AstraZeneca during the conduct of the study and outside the submitted work; and are employees and shareholders of AstraZeneca. No other disclosures were reported.

Submitted: 10 June 2023

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

# Supplemental material

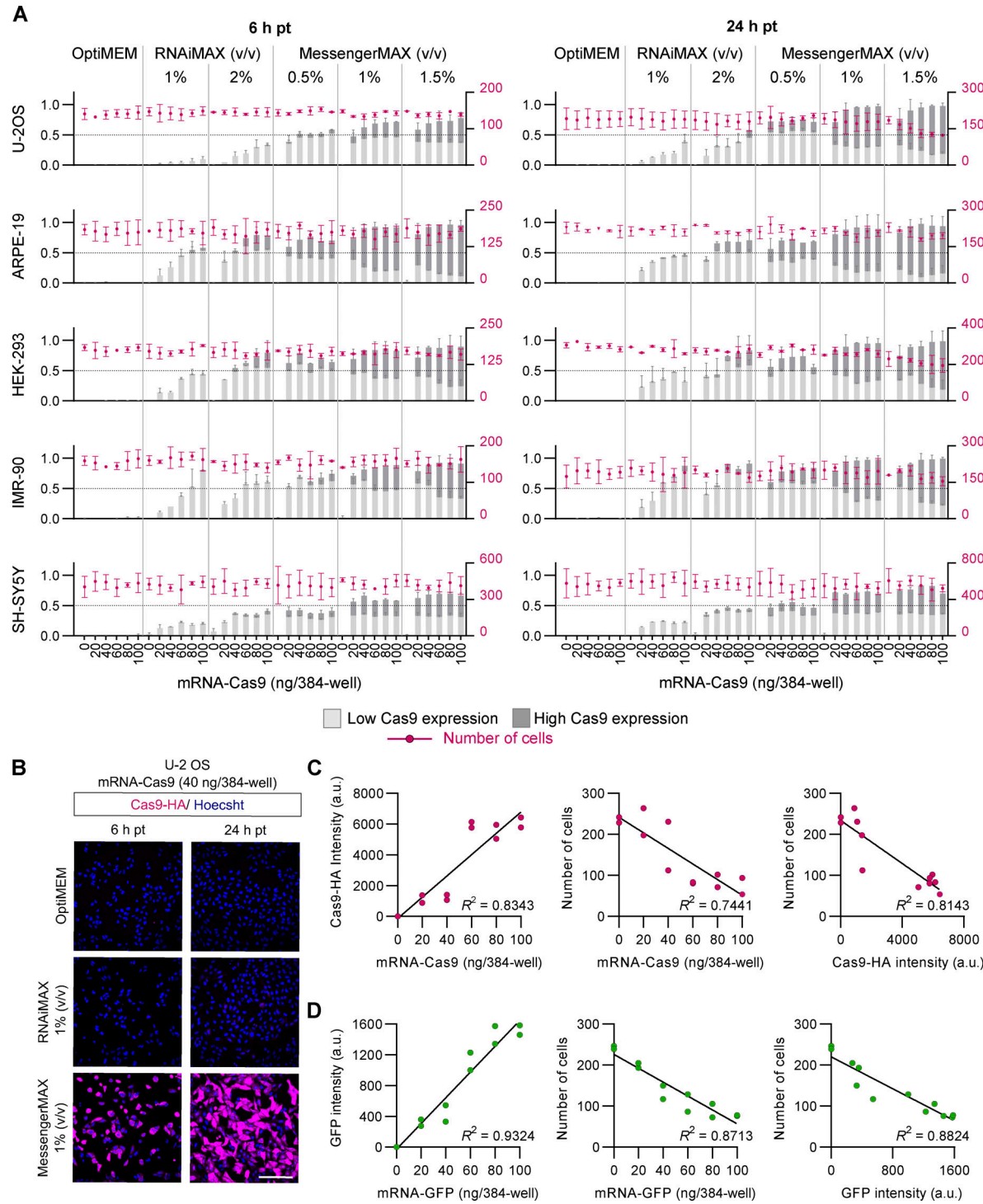

Figure S1. **Optimizing conditions for mRNA-Cas9 delivery. (A)** Efficiency and toxicity profile of mRNA-Cas9 delivery in a panel of five mammalian cell lines. Cells were transfected in a 384-well format with a titration of mRNA and transfection reagent and fixed either 6 or 24 h later (pt, post-transfection). OptiMEM (vehicle) was used as a control. Charts show the frequency of Cas9-positive cells (immunostaining for HA tag on Cas9 with gating for low and high expression based on fluorescence intensity; bar graph, left axis, labeled in light and dark gray), and the total number of cells (dot plot, right axis, labeled in magenta) as an indication of cytotoxicity. Data points represent mean per cell intensity values aggregated at well level from two independent experiments (minimum of 100 cells from two wells analyzed per condition). Error bars, SD. The condition selected for the study was 1% (vol/vol) MessengerMAX with 40 ng mRNA per well of a 384-well plate. **(B)** Representative images of U-2 OS cells stained for Cas9 (HA) and DNA (Hoechst) after transfection of mRNA-Cas9 (40 ng/well of a 384-well plate) coupled with 1% (vol/vol) RNAiMAX or MessengerMAX. Scale bar, 200 µm. **(C and D)** Reduced cell number with the highest tested MessengerMAX concentration is caused by high mRNA transfection efficiency and is not specific to Cas9 expression. U-2 OS cells were transfected with a titration of either mRNA-Cas9 or mRNA-GFP coupled with 1.5 % (vol/vol) MessengerMAX and fixed 24 h after transfection. Linear regression analysis showed a negative relationship between both Cas9-HA intensity (C) and GFP intensity (D) and cell number. The data point represents mean per cell intensity values from two independent experiments (minimum of 100 cells from two wells analyzed per condition).

Wong et al.
Arrayed genome-wide screen for dynein regulators

Journal of Cell Biology     S2

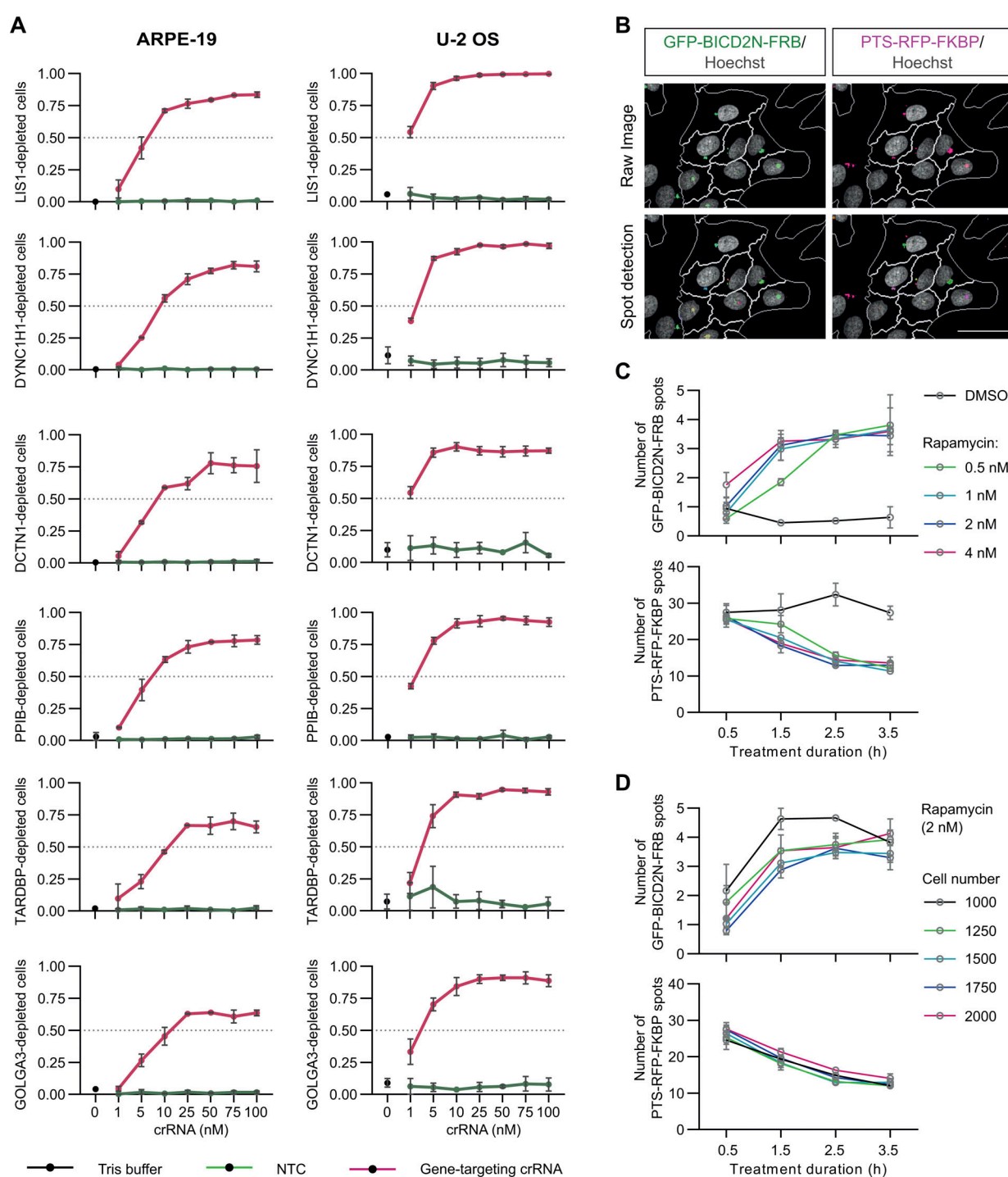

Figure S2. **Optimization of crRNA concentration and peroxisome relocalization assay. (A)** Quantification of CRISPR/Cas9 editing frequency in ARPE-19 and U-2 OS cells with varying crRNA concentration. Cells were transfected in a 384-well format with mRNA-Cas9 (40 ng/well) coupled with MessengerMAX (1%; vol/vol) for 6 h before transfecting with a panel of six gene-targeting crRNA pools (*crLIS1*, *crDYNC1H1*, *crPPIB*, *crDCTN1*, *crGOLGA3*, and *crTARDBP*) or the NTC pool at the indicated concentrations. The Tris buffer used for RNA resuspension was used as an additional control. Cells were fixed 72 h later and stained with antibodies to the corresponding protein products to evaluate the frequency of target protein depletion (gating based on the range of fluorescence signal of NTC cells). A final concentration of 50 nM crRNA per well was selected for the genome-wide screen. **(B)** Representative results of applying a spot detection mask on raw images of rapamycin-treated U-2 OS PEX cells. Scale bar, 50 μm. **(C)** Optimization of rapamycin concentration and treatment duration in the peroxisome relocalization assay. Note that, with these raw values, the number of GFP spots increases with perinuclear clustering because discrete puncta are otherwise relatively uncommon due to the dim signal, whereas RFP spot number decreases with perinuclear clustering as signals from multiple dispersed puncta coalesce at the MTOC. A 2.5-h treatment with 2 nM rapamycin was selected for the genome-wide screen. **(D)** Impact of the number of seeded cells per well (384-well plate format) on the number of GFP-BICD2N-FRB and PTS-RFP-FKBP spots. Cells were seeded for 72 h prior to treatment with rapamycin (2 nM). 1,500 cells per well were seeded for the genome-wide screen. In A, C, and D, data points represent mean aggregation at well level (minimum of 100 cells from at least two wells analyzed per condition). Error bars, SD.

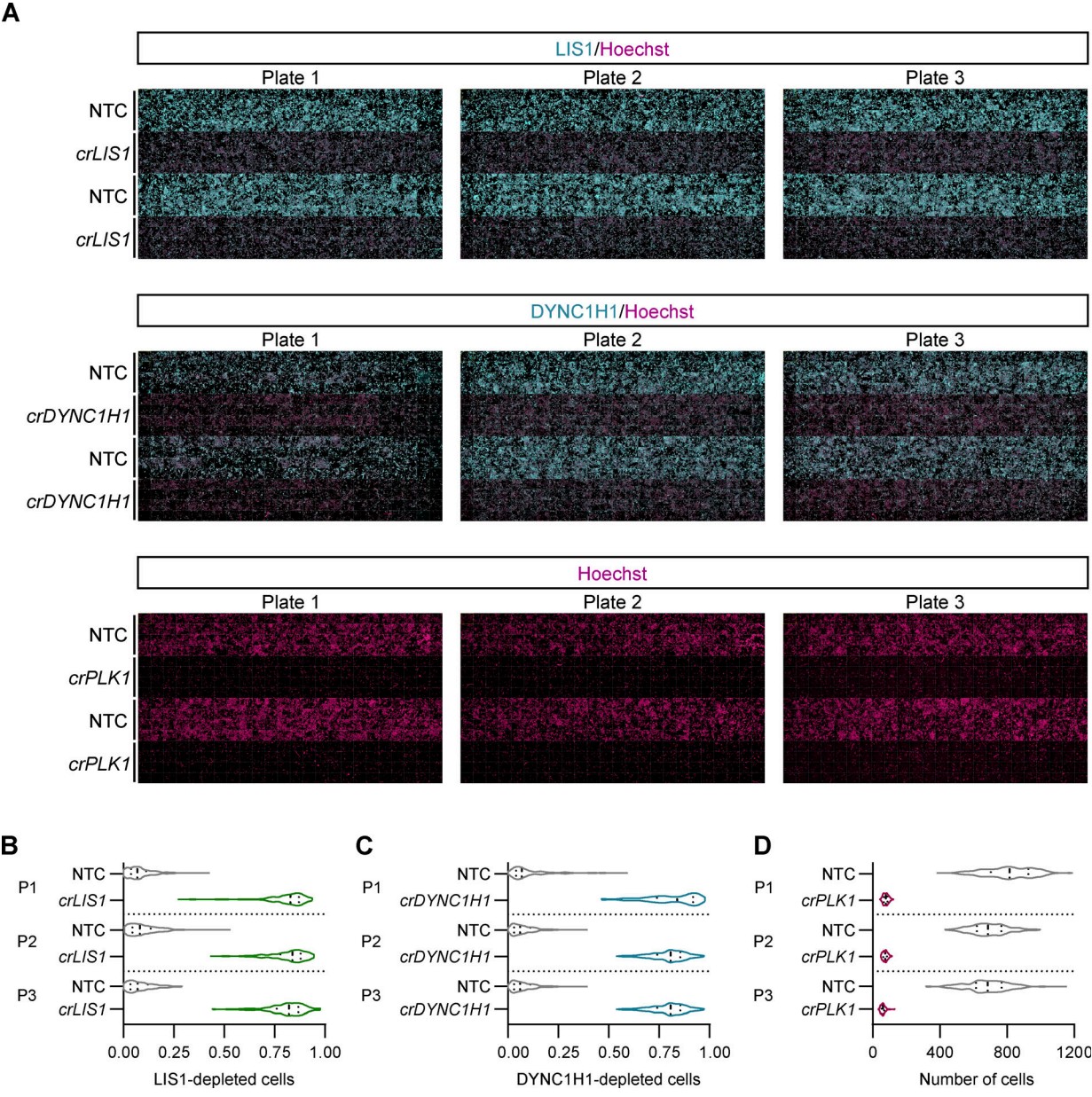

Figure S3. **Assay scaling for high-throughput editing. (A)** Representative low-magnification view of 384-well plate regions showing consistent editing in U-2 OS PEX cells treated with *crLIS1*, *crDYNC1H1*, and *crPLK1*. *crLIS1* and *crDYNC1H1* activities were assessed by immunostaining for the target proteins, whereas activity of *crPLK1* was read out by a reduction in cell number (revealed by Hoechst staining). **(B–D)** Violin plots (median, bold line; first/third quartile, dashed lines) of frequency of cells depleted for LIS1 (B) or DYNC1H1 (C), or the number of cells (D), after transfection with *crLIS1*, *crDYNC1H1*, or *crPLK1*, respectively. Gating of target-depleted cells was based on the range of the fluorescence signal of NTC cells. Datapoints represent mean per cell intensities aggregated at the well level (minimum of 100 cells from 192 wells per plate) for three individual plates (P).

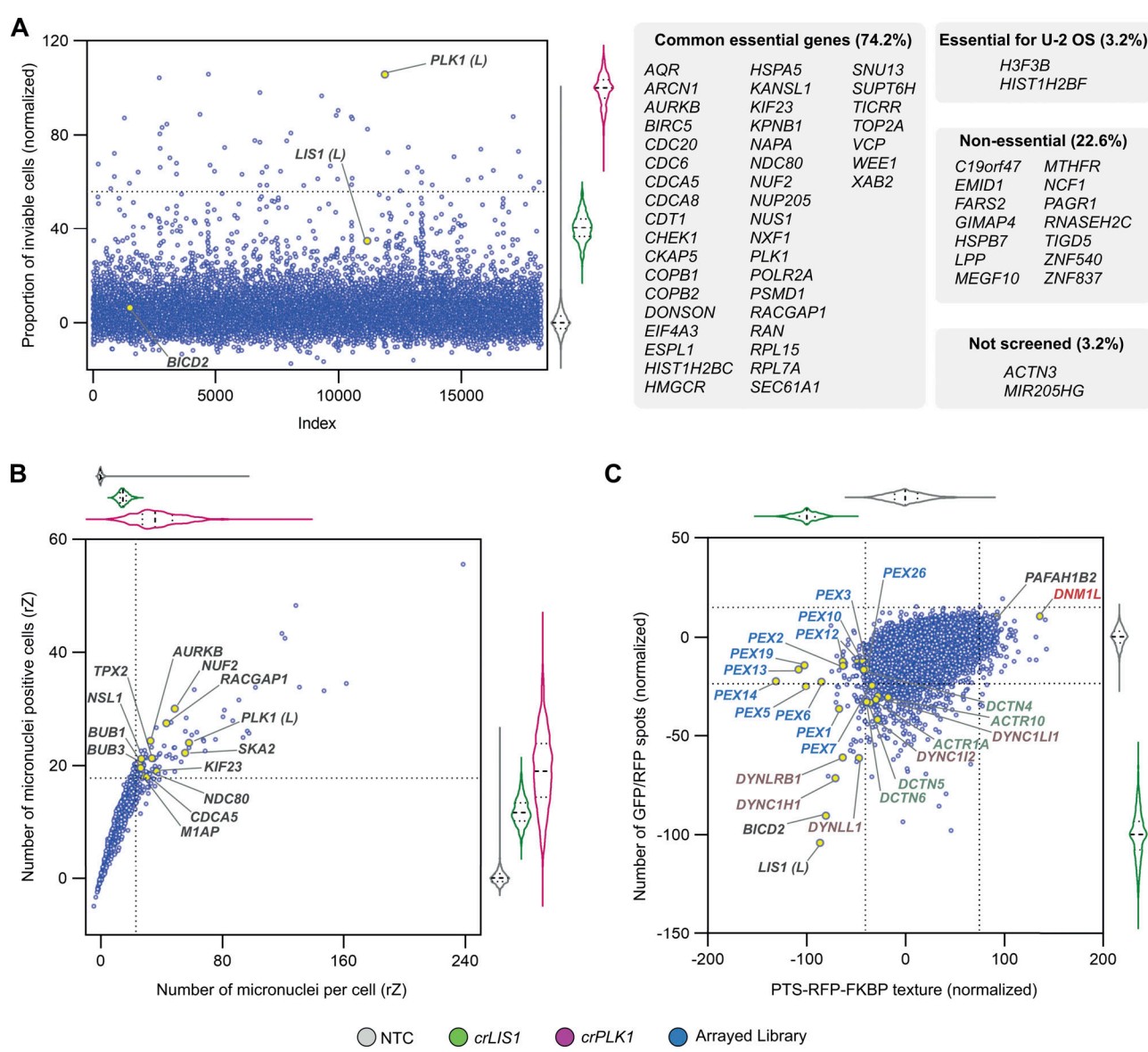

| NTC | crLIS1 | crPLK1 | Arrayed Library |

Figure S4. **Additional genome-wide screen endpoints. (A)** Effects of arrayed library crRNAs on cell viability and comparison to results from previous cell viability studies. Scatter plot of library results and corresponding violin plot for controls (median, bold dashed line; first/third quartile, dashed lines; color code in key at bottom of figure), proportion of inviable cells (gated based on nuclear morphology of NTC cells). Data points represent normalized values based on the neutral (NTC, 0) and lethal editing (crPLK1, 100) controls. Dashed line on y-axis represents 2.5*SD of crLIS1, the threshold for calling lethal crRNAs. Library copies of crPLK1, crBICD2 and crLIS1 are labeled with "(L)." The genes targeted by the lethal crRNAs were cross-referenced with their gene essentiality categorization from the Cancer Dependency Map project (DepMap 22Q2 Public+Score; https://depmap.org/portal): "common essential genes" are classed as essential for growth and survival in ≥90% of cancer cell lines; "essential for U-2 OS" genes are those classed as not essential across multiple cell lines but essential in U-2 OS cells; "non-essential" genes are those not identified as essential in a panel of cell lines; "not screened" genes are those that are not represented in the DepMap screening dataset. Numbers in parentheses refer to the percentage of essential genes in our dataset found in each Depmap category. **(B)** Effects of arrayed library crRNAs on micronuclei incidence. Scatter plot of library results and corresponding violin plot of controls (median, bold dashed line; first/third quartile, dashed lines; color code in key at bottom of figure) for the number of micronuclei per cell (x-axis) and number of cells containing micronuclei (y-axis). Data points represent rZ normalization (central reference = NTC). Dashed lines on the x- and y-axes represent 2.5*SD of crLIS1, the threshold for calling crRNAs that cause a micronucleus phenotype. The library copy of crPLK1 is labeled with "(L)." Labeled genes were functionally enriched (FDR ≤ 0.005) for GO terms associated with regulation of chromosome segregation, including, "nuclear division," "mitotic sister chromatid segregation," and "nuclear chromosome segregation" (http://bioinformatics.sdstate.edu/go/). **(C)** Effects of arrayed library crRNAs on PTS-RFP-FKBP texture and number of PTS-RFP-FKBP and GFP-BICD2N-FRB spots. Scatterplot of library results and corresponding violin plots of controls (median, bold dashed line; first/third quartile, dashed lines; color code in key at bottom of figure) of normalized values based on the neutral control (NTC, 0) and crLIS1 (−100). Dashed lines on both axes represent ± 2.5*SD of NTC and arrayed library, the threshold for hit calling. Both endpoints were generated from a linear discriminant analysis; PTS-RFP-FKBP texture was generated from six textural features based on filtered images, while "GFP/RFP spots" were generated by two features (total number of GFP-BICD2N-FRB and PTS-RFP-FKBP spots). Core components of the dynein complex and dynactin complex that met the threshold for hit calling for either endpoint are labeled in purple and teal text, respectively. Library copy of crLIS1 is labeled with "(L)." Known peroxisome biogenesis genes are labeled in blue text (note duplicate of the PEX12 crRNA pool in the library). crPAFAH1B2 and crDNM1L are labeled as examples of crRNA pools that affect PTS-RFP-FKBP texture in the opposite way to crRNAs targeting dynein–dynactin components.

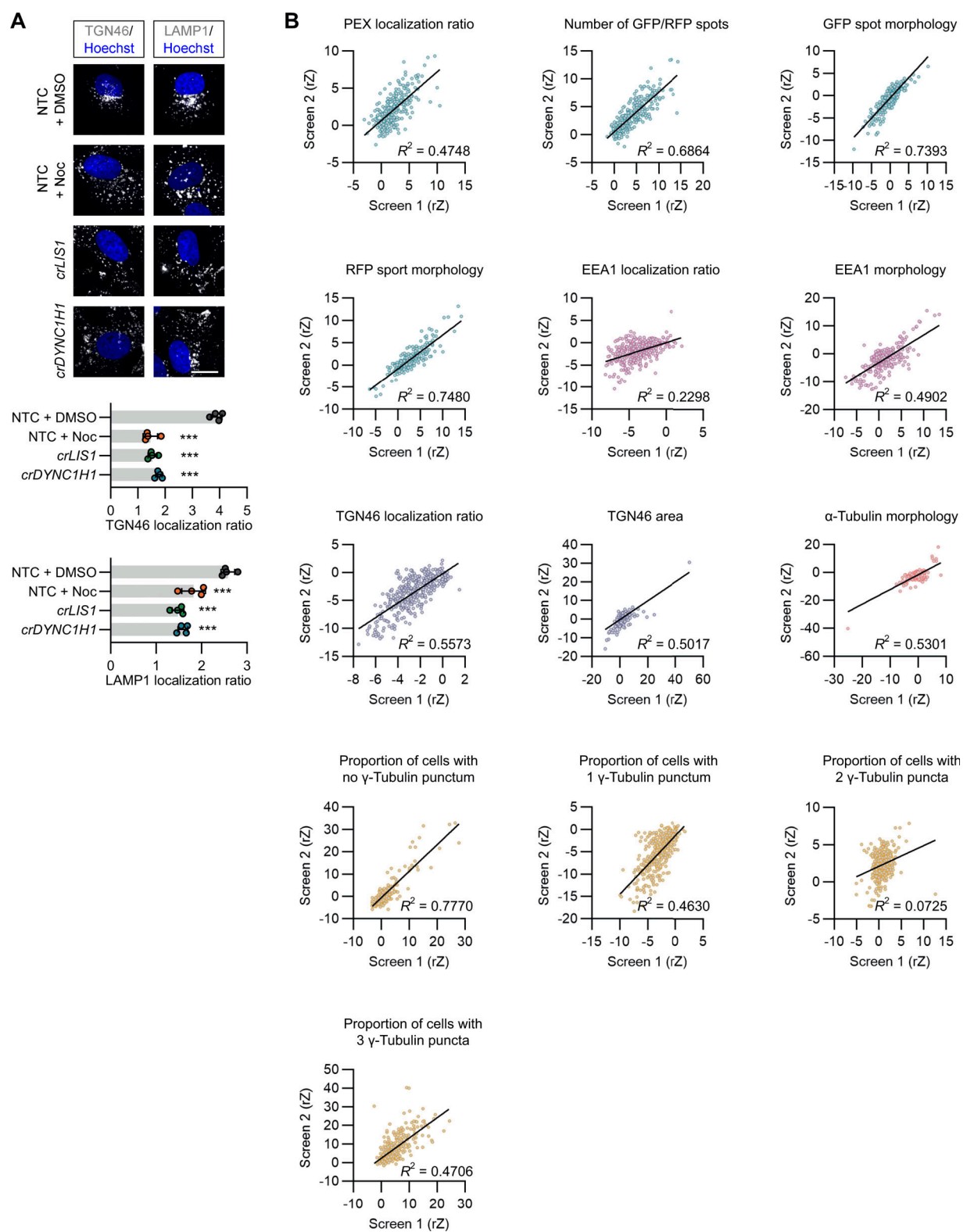

Figure S5. **Supplemental data on imaging readouts for downstream analysis of screen hits. (A)** Confirmation that dynein promotes perinuclear enrichment of Golgi and lysosomal membranes in U-2 OS cells. Shown are representative images and quantification of dispersion of trans-Golgi network (TGN46) and lysosomal membranes (LAMP1) in U-2 OS cells following the indicated treatments (Noc, nocodazole). Bar graphs show the ratio of spot number in the perinuclear region versus the peripheral region (lower values indicate increased dispersion). Data points represent mean per cell intensity values aggregated at the well level (minimum of 100 cells analyzed per well; four wells analyzed per condition). Error bars signify SD. ***P < 0.001 (one-way ANOVA with Dunnett's multiple comparison versus NTC + DMSO). Scale bar, 20 μm. **(B)** Reproducibility of secondary screen biological replicates. Linear regression was performed on the main endpoints used in the two independent runs in the secondary screen. Data points represent the median rZ (central reference = NTC) of individual crRNA pools. The "proportion of cells with two γ-Tubulin puncta" had an $R^2$ score of <0.2 (0.07254) and therefore was not used for hit calling.

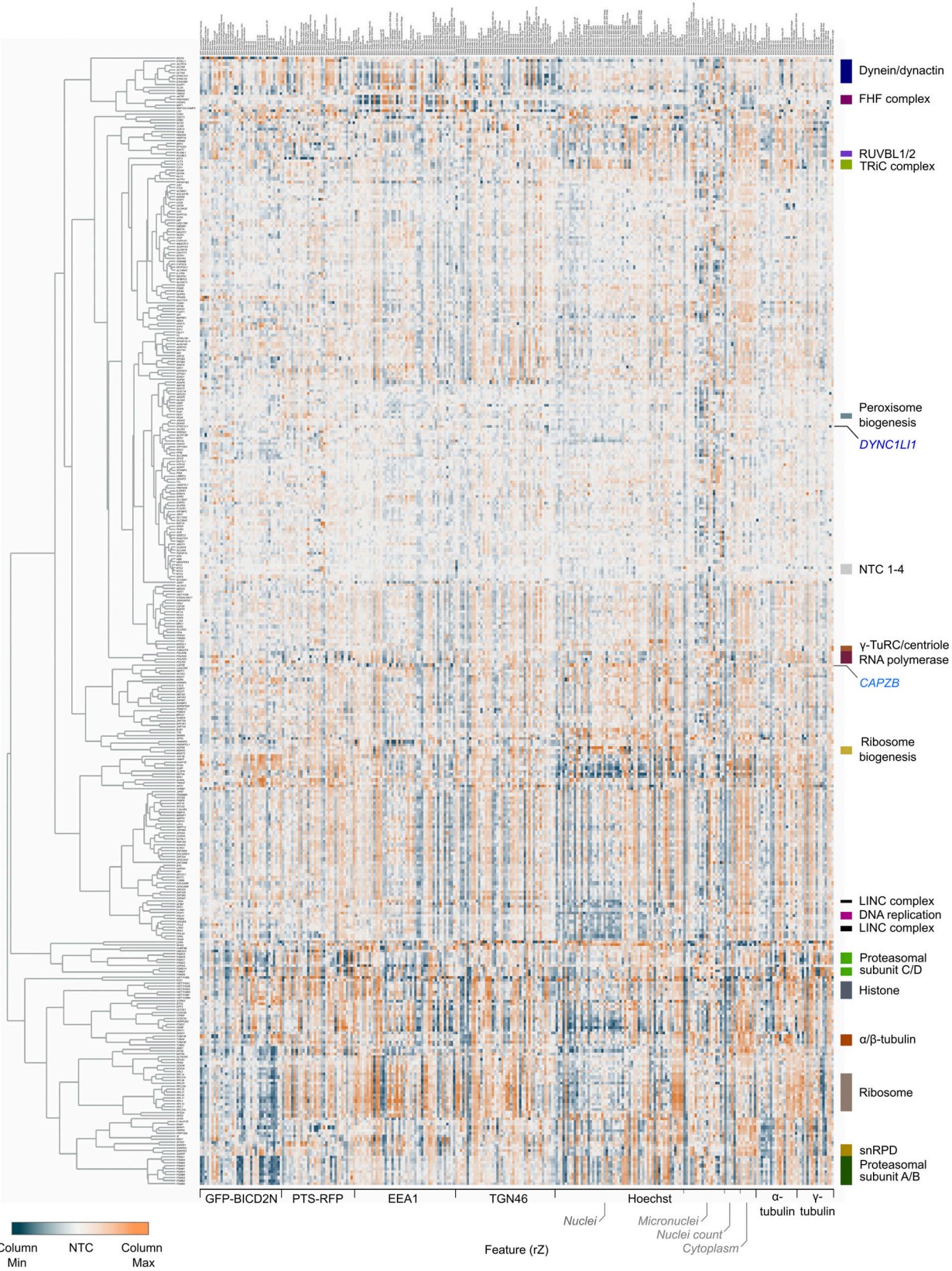

Figure S6. **Overview of functional clusters from image-based profiling of secondary screen data.** Displayed is the phenotypic feature heatmap generated by hierarchical clustering. The data, including gene names and feature titles, can be explored by zooming in within Data S1 as well as in Table S8. The scale of rZ values (central reference = NTC) was adjusted based on the minimum and maximum values of individual features. Labels to the right highlight a subset of functional clusters, as well as the positions of the NTC (neutral control) crRNAs and crRNAs targeting *CAPZB* and *DYNC1LI1*. "Cytoplasm" refers to features associated with the background Hoechst staining in the cytoplasm.

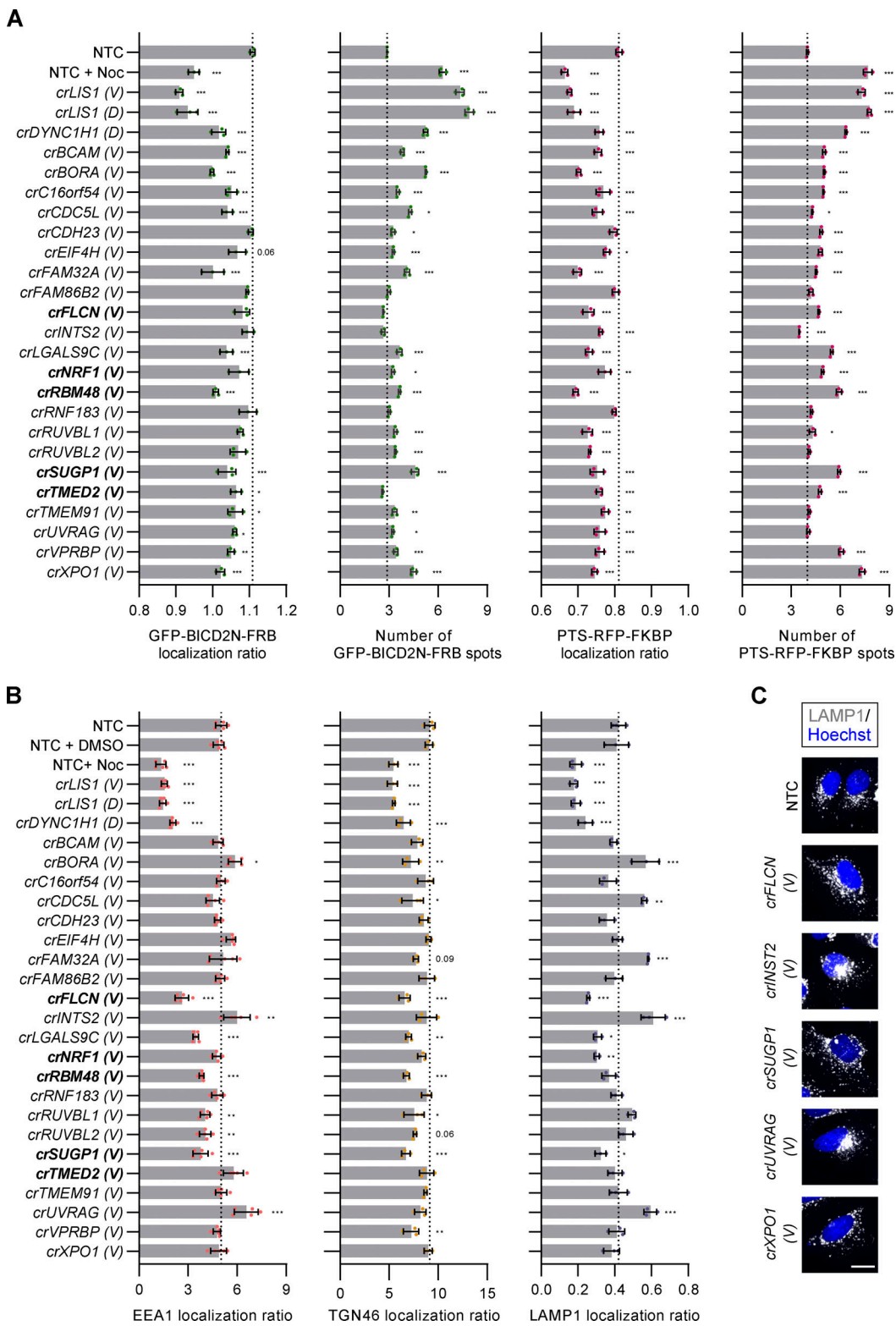

Figure S7. **Readouts of orthogonal cargo localization screen. (A and B)** Bar graphs showing readouts generated from the screen with VBC-generated crRNAs. Quantification of (A) localization ratio (perinuclear versus peripheral) and the total number for GFP-BICD2N-FRB and PTS-RFP-FKBP spots in U-2 OS PEX cells treated with rapamycin and (B) localization ratio (perinuclear versus peripheral) of EEA1, TGN46, or LAMP1 spots in unmodified, untreated U-2 OS cells. "(V)" and "(D)" indicate crRNAs synthesized based on the VBC score or from the original "Horizon Discovery" set, respectively. Data points represent mean aggregation from at least three independent experiments (minimum of 100 cells analyzed per well; four wells analyzed per condition). EEA1 ratio values were log-transformed for normal distribution. Bold lettering indicates crRNAs that were novel components of the dynein–dynactin cluster of phenotypic profiles. Error bars signify SD. *P < 0.05, **P < 0.01, ***P < 0.001 (one-way ANOVA with Dunnett's multiple comparison against NTC). **(C)** Representative images of LAMP1 staining in U-2 OS cells treated with the indicated crRNAs. Scale bar, 20 µm.

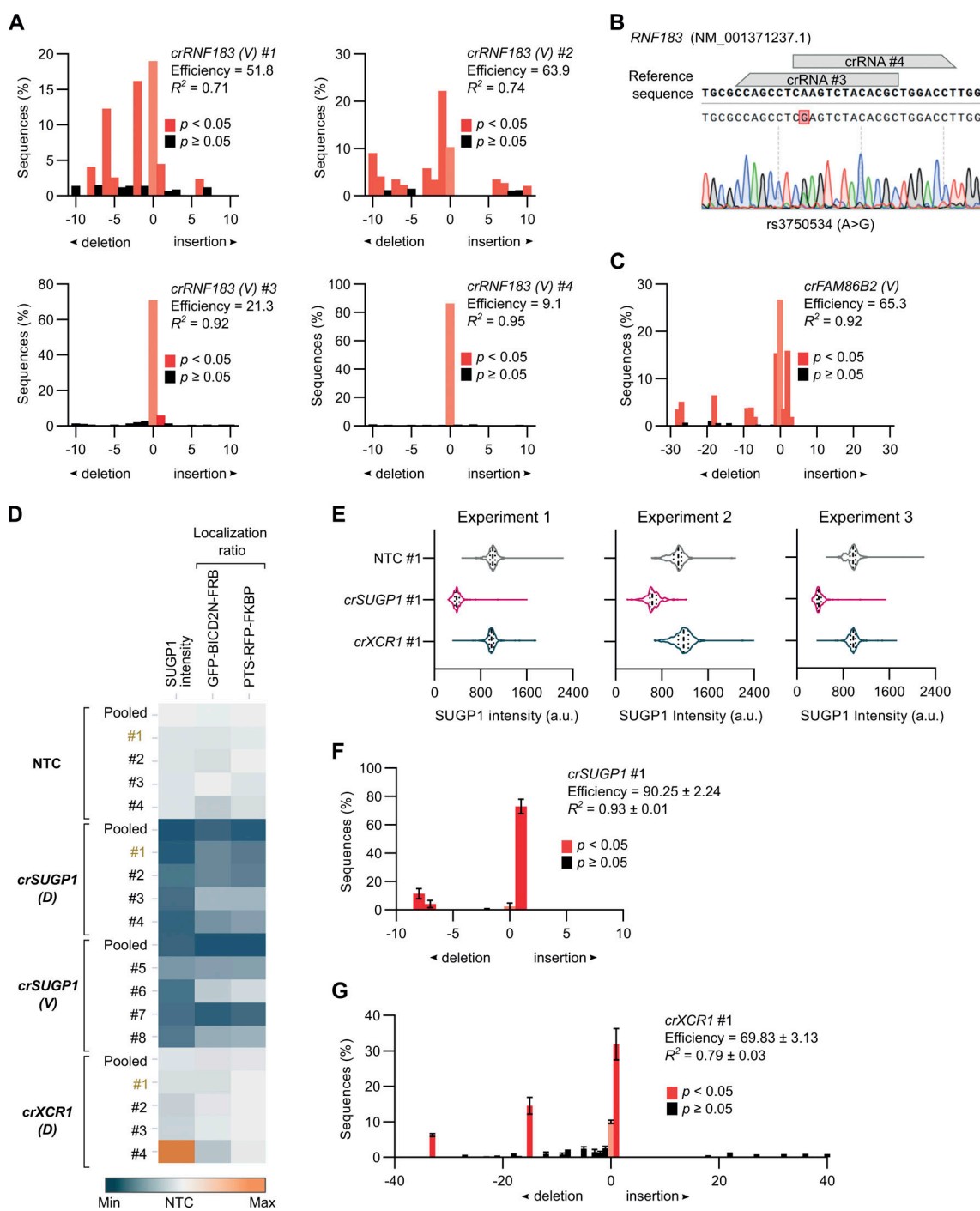

Figure S8. **Supplemental data on activity of *RNF183*, *FAM86B2*, and *SUGP1* crRNAs. (A)** Indel distribution (from TIDE analysis) in sequences of target regions in unmodified U-2 OS cells transfected with VBC-derived (*V*) *RNF183* crRNAs. **(B)** Incomplete targeting with the *crRNF183* pool may be associated with an overlapping target region and/or spanning of a common polymorphism (rs3750534) within a target sequence for *crRNF183* (*V*) #3 and *crRNF183* (*V*) #4, which have particularly low activity. **(C)** Indel distribution (from TIDE analysis) in sequences of target regions in unmodified U-2 OS cells transfected with the VBC-derived (*V*) *crFAM86B2* crRNA pool. Insufficient targeting of the *crFAM86B2* pool may be due to all four crRNAs targeting overlapping regions, and therefore competing with each other. **(D)** Phenotyping of individual and pooled *SUGP1* crRNAs. Heatmap displaying quantification of SUGP1 protein signal and localization ratio (perinuclear versus peripheral) of GFP-BICD2N-FRB and PTS-RFP-FKBP spots in U-2 OS PEX cells with the indicated treatments. "(*D*)" and "(*V*)" indicate source of crRNA design (*D*, Discovery; *V*, VBC score). Data points represent mean per cell values aggregated at well levels (minimum of 100 cells analyzed per condition; four wells per condition). Color scale of individual features was adjusted based on minimum and maximum raw values. The guides selected for RNA-seq are labeled in gold text. **(E–G)** Quality control for samples submitted for RNA-seq. **(E)** Violin plots of intensity of SUGP1 protein signal at the single cell level (median, bold line; first/third quartile, dashed lines; minimum of 100 cells analyzed per condition). **(F and G)** Indel distribution (from TIDE analysis) in sequences of target regions in unmodified U-2 OS cells transfected with *crSUGP1* #1 (F) or *crXCR1* #1 (G). In A, C, F, and G, bar graphs display mean ± SD, and the efficiency scores are out of 100. Corresponding sequences from cells transfected with NTC crRNAs #1–4 (A and C) and NTC crRNA #1 (F and G) were used as a reference.

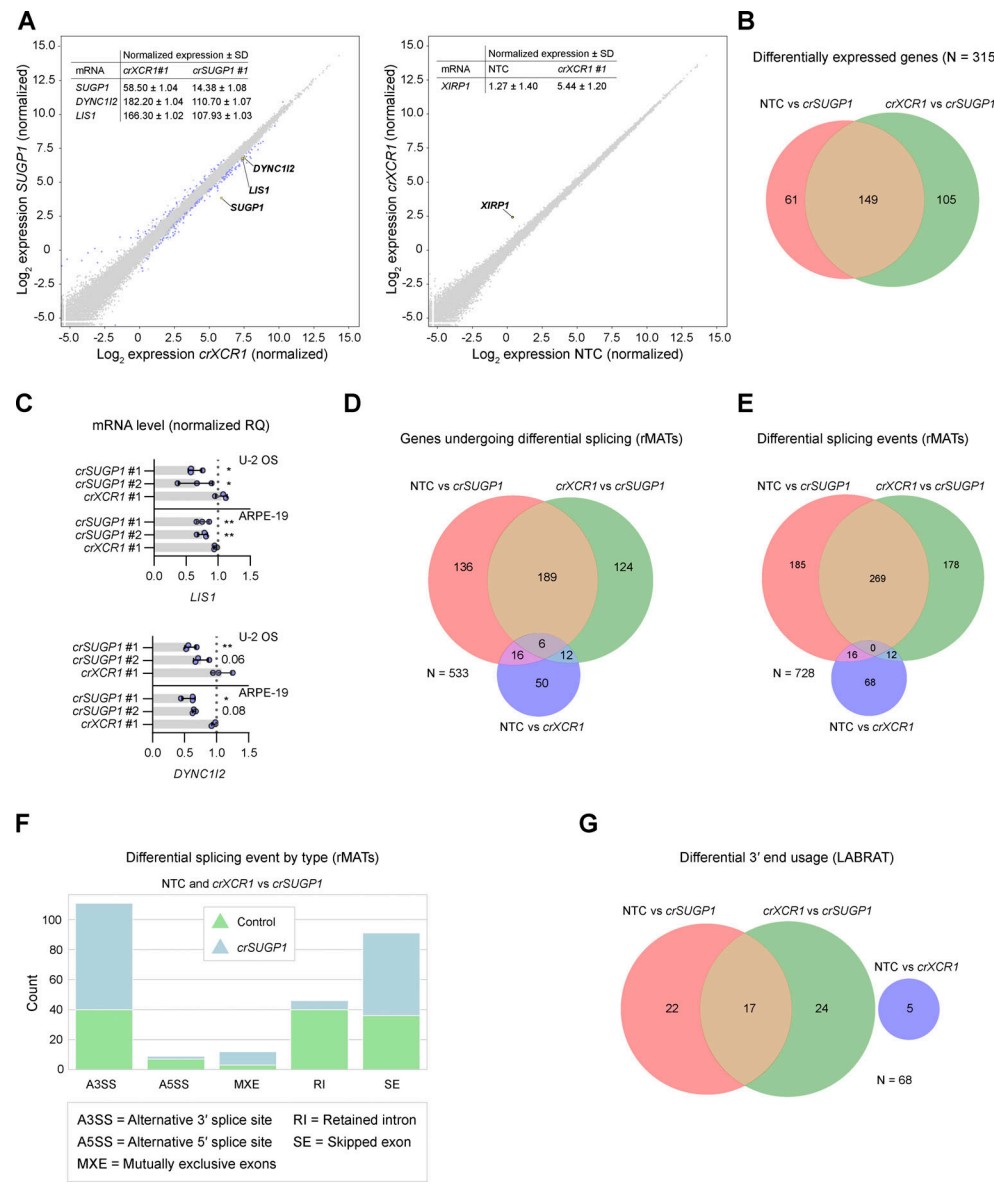

Figure S9. **Supplemental data for differential expression and splicing analysis. (A)** Scatter plot of mRNA abundance for (left panel) *XCR1*-edited versus *SUGP1*-edited U-2 OS cells and (right panel) NTC versus *XCR1*-edited U-2 OS cells (mean log₂ normalized values from three independent experiments). mRNAs meeting threshold for inclusion (minimum absolute log₂ normalized fold change ≥0.5 and FDR ≤ 0.05) are labeled in blue, except (left panel) *SUGP1*, *DYNC1I2*, and *LIS1*, and (right panel) *XIRP1* (the only differentially expressed gene in the NTC versus *crXCR1* comparison), which are labeled in yellow. Inset tables show non-logarithmic values for (left panel) *SUGP1*, *DYNC1I2*, and *LIS1* and (right panel) *XIRP1* mRNAs. See Table S9 for full results. **(B)** Venn diagram showing overlap of differentially expressed genes in the NTC versus *crSUGP1* and *crXCR1* versus *crSUGP1* comparisons. **(C)** Quantification of *LIS1* and *DYNC1I2* mRNA level, determined by TaqMan-based real-time qPCR, in *SUGP1*-edited and *XCR1*-edited U-2 OS and ARPE-19 cells. Data points represent the mean of three independent experiments (RQ = relative quantification based on NTC). Error bars signify SD. *P < 0.05, **P < 0.01 (one-way ANOVA with Dunnett's multiple comparison against NTC). **(D and E)** Venn diagrams showing the overlap of genes that undergo differential splicing (D) and differential splicing events (E) in the datasets, as determined with rMATs (note that some genes have >1 differential splicing event). The threshold for classifying an event as differential was: absolute IncLevelDifference ≥0.2, total read count (inclusion count + skipping count) ≥10, and FDR ≤ 0.05. See Table S10 for full results. **(F)** Classes of alternative splicing events common to both comparisons (i.e., NTC versus *crSUGP1* or *crXCR1* versus *crSUGP1*), as identified by rMATS. Blue and green represent events that were enriched in control and *crSUGP1* samples, respectively. rMATS reports 5 splicing categories: (i) alternative 3' splice sites (A3SS); (ii) alternative 5' splice sites (A5SS); (iii) mutually exclusive exons (MXE); (iv) retained introns (RI), and (v) skipped exons (SE). The A3SS and A5SS events involve the splicing together of two exons separated by a single intron. For A3SS events, alternative splicing causes a downstream exon to extend partially into neighboring intronic sequence. A5SS is defined by an alternative splicing event causing an upstream exon to extend partially into the adjoining intron. MXEs describe the splicing of adjacent exons (separated by a single intron) in which one exon is retained but the other is excluded, or vice versa. The graph reports MXE events in which the upstream exon was selected. Events classified as RI are those in which an intron is not spliced out and hence is retained in the mature transcript. SE denotes splicing events in which an exon is skipped over and not included in the processed RNA molecule. **(G)** Venn diagrams showing overlap of differential 3'-end usage events in the datasets, as determined by LABRAT. LABRAT quantifies alternative polyadenylation sites and reports upstream or downstream shifts in the usage of those sites for each gene as compared to the control (see Table S11 for full results). The threshold for classifying an event as differential was Δψ ≥0.05 and FDR ≤ 0.05.

Provided online are 13 tables and two datasets. Table S1 shows features and data acquired in the genome-wide screen (Excel file). Table S2 lists hits identified using different features of cellular organization (Excel file). Table S3 shows components of the dynein machinery identified in assay endpoints (Excel file). Table S4 lists the genes taken forward to secondary screens and associated GO data (Excel file). Table S5 shows K-means clusters and cargo localization ratios from the secondary screen (Excel file). Table S6 shows γ-Tubulin and α-Tubulin datasets from secondary screen (Excel file). Table S7 shows source data for UMAP in Fig. 5 B (Excel file). Table S8 shows phenotypic fingerprints for clustering (Excel file). Table S9 shows differentially expressed genes in NTC, *crSUGP1*, and *crXCR1* samples (DEseq) (Excel file). Table S10 shows differential slicing events in NTC, *crSUGP1*, and *crXCR1* samples (rMATS) (Excel file). Table S11 shows differential 3′-end usage events in NTC, *crSUGP1*, and *crXCR1* samples (LABRAT) (Excel file). Table S12 shows sequences of secondary screen and VBC-based independent crRNAs (Excel file). Table S13 lists primary and secondary antibodies used for immunofluorescence (Excel file). Data S1 shows a high-resolution, explorable version of the phenotypic heatmap. Data S2 shows image acquisition and analysis workflow for genome-wide CRISPR screen.

