## [Peer Review File · The Journal of Cell Biology]

Genome-scale requirements for dynein-based transport revealed by a high-content arrayed CRISPR screen

Chun Hao Wong, Steven Wingett, Chen Qian, Morag Rose Hunter, Matthew J Taliaferro, Douglas Ross-Thriepland, and Simon L Bullock

Corresponding Author(s): Simon L Bullock, MRC Laboratory of Molecular Biology

Review Timeline:

Submission Date:	2023-06-10
Editorial Decision:	2023-06-21
Revision Received:	2024-01-10
Editorial Decision:	2024-02-12
Revision Received:	2024-02-16
Accepted:	2024-02-19

Monitoring Editor: Melissa Gardner

Scientific Editor: Andrea Marat

Transaction Report:

DOI: <https://doi.org/10.1083/jcb.202306048>

Revision 0

Review #1

1. Evidence, reproducibility and clarity:

Evidence, reproducibility and clarity (Required)

Cytoplasmic dynein-1 (dynein) is the predominant minus-end directed microtubule-based motor involved in the transport of numerous cellular cargoes in addition to mitotic functions. Although in vitro analysis of dynein assembly and interactions are becoming more common, cell biological studies that aim at identifying different dynein cargoes and functions are lacking. To shed light on dynein's function, Wong et al., performed a genome-wide CRISPR loss-of-function screen using peroxisome-tethered and endosome localization assays as readouts. Detailed data analysis and supervised and unsupervised phenotypic clustering of targets from an extensive RNA gene library (~20,000 genes) revealed ~200 genes disrupted in cargo trafficking processes. The authors also showed that a novel gene SUGP1, identified in their screen, promotes the expression of a critical dynein activator, LIS1. The generated datasets provide a rich library of genes that can be further mined by other researchers in the field.

****Major comments:****

This manuscript reads well, and the conclusions are mostly supported by experimental data. The authors went to great lengths to optimize their high throughput assay setup by testing different cell lines and transfection conditions and included different positive and negative controls, which is a strength of the study. The use of two functional readouts (early endosome and BICD2-driven peroxisome distribution) in the initial screen followed up by a validation screen of a smaller subset of genes using readouts for dynein disruption phenotypes (Golgi fragmentation, lysosome clustering) is another strength. In addition, the follow-up identification of an RNA-binding protein SUGP1 as a regulator of LIS1 mRNA levels provides an interesting new way of regulating dynein function.

I have a few concerns about the experimental design and conclusions.

1. The images in Fig. 1E and 1G for the dynein control (crDYNHC1) show some clustering around the cell nucleus while crLIS1 knockout shows no perinuclear clustering. Is this expected? Shouldn't dynein knockout prevent perinuclear clustering? Is it possible that crDYNHC1 does not lead to a complete knockout? Given that this is a proof-of-principle control in the assay, a more detailed validation of DYNHC1 knockout using western blotting and RT-qPCR, in addition to the validation shown in Fig. 1C would strengthen the claim that this control works as expected. These experiments should be fast and easy to do.
2. The use of multiple crRNAs together to target a single gene can increase off-target effects, however, the authors never test for off-target phenotypes or address the possibility of off-targeting. Can the authors show using a few examples that their approach does not lead to

significant off-targeting? This should also be addressed in the text.

3. It is unclear to me how the authors established the limits for the quantification of localization ratios. As described in methods, the perinuclear region was defined as having an outer limit of 7 μm from the nuclear envelope. However, the cells are not the same size (also seen in representative images), which could skew the calculation of ratios solely based on fixed distance limits. Have the authors considered taking into account cell size? Perhaps a more accurate calculation would be to measure the distance from the nucleus to the cell periphery for each cell and normalize this value to the cell size to account for cell size differences. The perinuclear region could then be defined as the percentage of the distance from the nuclear envelope of the normalized cell radius. It is also unclear how the size and intensity of each "spot" are accounted for in the analysis as this is an important aspect of the quantification given that the "spots" are not the same size/intensity. Redoing this analysis would not require the authors to collect any new data but could help in gene identification, especially given that the authors only identified ~50% of the known dynein-dynactin complex components to be disrupted in their assay. These genes might have more subtle phenotypes that could be amplified by doing more precise image analysis and quantification.

****Minor comments:****

1. It would be helpful if the authors could change gene names to a bold or brighter font in scatter plots in all figures. It is hard to read the names the way they are right now.
2. Line 185 the authors say: "We also analysed the induction of micronuclei (Supplementary figure 6B), which to our knowledge has not been assessed in earlier screens." What screens are the authors referring to? Could you add references here?
3. Line 256: "Each cargo was assayed in two independent screens, in which there was good agreement in general between the effects of the crRNA pools (Supplementary figure 8)." The authors also indicate in the legend for Supplementary figure 8 that "The only metric with a poor R2 score (proportion of cells with two γ - Tubulin puncta) was not used for hit calling." However, the EEA1 localization ratio also shows poor R2 score, shouldn't this screen also be excluded? In general, what was the cutoff for R2 score? This information should be included.

2. Significance:

Significance (Required)

This work is the first genome-wide loss-of-function CRISPR study (to my knowledge) aimed at identifying dynein-driven trafficking disruption phenotypes. In general, the data generated in this study will enrich the field's understanding of how dynein is regulated and how it achieves its broad cargo and functional specificity. This manuscript will also provide a resource and experimental setup for the design of other genome-wide loss-of-function CRISPR studies.

I have broad expertise in the cytoskeleton field with a detailed understanding of dynein's function from a mechanistic and functional perspective. I have minimal experience with high throughput screening, but I am experienced with CRISPR-based assays and cell imaging techniques.

3. How much time do you estimate the authors will need to complete the suggested revisions:

Estimated time to Complete Revisions (Required)

(Decision Recommendation)

Less than 1 month

Yes

Review #2

1. Evidence, reproducibility and clarity:

Evidence, reproducibility and clarity (Required)

In this manuscript the authors have conducted a genome-wide CRISPR loss-of-function screen in human cells to find regulators of cytoplasmic dynein, a microtubule-based motor that plays a major role in the transport of cargo towards microtubule minus ends. The screen was carried out to address how dynein is synthesised and assembled, and how its activity is controlled to enable the motor to selectively transport a wide variety of cargoes. Cells were fixed cells 72 hours after transfection and fluorescently stained for intracellular markers. Several read-outs were used in the screen, of which the major ones were the distribution of dynein-tethered peroxisomes and early endosomes. The authors used a 384 well format (61 unique 384-well plates) and a fluorescence microscopy-based imaging readout to gauge dynein activity. From a guide RNA library targeting 18,253 genes, the authors recovered 195 validated hits. For one gene (SUGP1) follow-up studies demonstrate that the protein encoded by this gene controls the levels of the dynein activator LIS1 and thereby promotes cargo trafficking by dynein. The dataset reported here represents a source for investigating proteins that might be involved in minus-end

microtubule-based transport, as well as in other aspects of cellular organisation that were captured in the high-content imaging approach.

I find this an interesting and well written resource manuscript, both from the perspective of how to conduct and analyse a high-content imaging screen, as well as from the dynein biology view. Results presented in this manuscript deserve follow-up studies. I do have a number of comments.

****Major comments****

1. On page 6 the authors state they used 61 x 384 wells. This equals 23,424 wells, but the authors state they analysed (8,150,065 cells from) 24,576 wells. What causes this difference in number? More importantly, the authors target 18,253 genes with four guides per gene. If I understand correctly these four guides per gene are present in a single well and the high-content imaging experiment was only done once. Although many cells were analysed per well (four fields of view per well; median of 345 cells analysed per well) and results are interesting and appear solid, I do think a replicate experiment is necessary.
2. The screen was developed based on the U-2 OS PEX line, in which tethering of dynein to peroxisomes is achieved by addition of rapamycin acting via a split BICD2 protein. Thus, the assay depends on the BICD2 adapter. Is this limiting when one is looking for dynein regulators?
3. Related to the question above, the authors do not recover BICD1 in their screen. Is this because BICD1 is not expressed in the cell systems used or is there another reason?
4. It has very recently been shown (doi.org/10.1038/s41467-023-38116-1) that BICD2 phosphorylation by CDK1 in the G2 phase of the cell cycle promotes its interaction with PLK1. This is followed by PLK1 phosphorylation in the N-terminus of BICD2, which in turn facilitates interaction with dynein and dynactin, allowing the formation of active motor complexes. Thus, adaptor activation through phosphorylation regulates dynein activity. In the present manuscript the authors use PLK1 as a read-out of cell viability. However, PLK1 also appears to regulate dynein via BICD2 phosphorylation. Given the latest results would the authors interpret their PLK1 data differently? Would it be preferable to screen for regulators of dynein activity in non-dividing cells?
5. Using the 377 genes listed in Supplementary Table 4 I performed a Metascape analysis. The results suggest that many of the hits are proteins involved in RNA metabolism or the cell cycle and that many of the encoded proteins form complexes. Based on this I wonder whether the screen yielded many proteins that are involved in controlling the steady state levels of dynein, microtubules, or of the dynein regulators. SUGP1 is an example of this. I suggest that the authors include an extensive Metascape analysis in a new version of the manuscript.
6. On page 11 a UMAP plot is described, which is shown in Figure 4B. How were the "members of the same protein complexes, such as histones, ribosomal proteins, RNA polymerase II, the RUVBL and TRiC/CCT chaperonins, FAM160A2- AKTIP-HOOK3 and dynein-dynactin" identified?
7. How do the complexes identified in Figure 4B relate to the MCODE-based complexes identified in Metascape?

2. Significance:

Significance (Required)

I think the present manuscript is an interesting resource paper for the dynein community. The advance is technical rather than conceptual.

I am a cell biologist with an interest in microtubules and how this cytoskeletal network controls cell shape and function. I analyse this using fluorescence microscopy and -omics approaches. I am not an expert in high-content imaging screens and analyses but the data presented here seem solid and novel to me.

3. How much time do you estimate the authors will need to complete the suggested revisions:

Estimated time to Complete Revisions (Required)

(Decision Recommendation)

Between 1 and 3 months

Yes

Review #3

1. Evidence, reproducibility and clarity:

Evidence, reproducibility and clarity (Required)

In this manuscript, the authors performed an arrayed CRISPR loss-of-function screen targeting 18,253 genes with the goal of uncovering gene products that regulate cytoplasmic dynein-1 motor function. In order to assess the impact of gene knockout, the authors optimized a protocol

for transfecting pools of cells with mRNA encoding Cas9 and scalably delivering arrayed pools of synthetic guides targeting a single gene to knock-out. In order to link gene knockouts to dynein-1 function the authors employed (1) a previously developed cell model U-2 OS PEX and (2) anti-EEA1 and anti- α -Tubulin antibodies and (3) hoechst as high-content fluorescent readouts for their genome-wide screen.

The authors then picked a subset of genes to move forwards with that were deemed as hits. A secondary round of screening was performed on these hit genes and unsupervised phenotypic clustering was performed on the feature vectors derived from the high-content images. These analyses revealed several distinct phenotypic clusters that can be categorized by the dynein cargoes or other functional categories including proteostasis related functions. The authors identified the gene SUGP1, which has never previously been linked to dynein-dynactin functionality.

The authors then show that targeting SUGP1 reduces the mRNA of both LIS1 and DYNC112 and the subsequent protein abundance of only LIS1.

In summary, the authors provide an optimized method for performing what they have termed 'one-shot' genome wide arrayed screening with pools of synthetic guides. They additionally have generated a data resource for others interested in understanding early endosome pathways and dynein-dynactin functionality.

The technical feat of generating such a large dataset and optimizing the protocol for arrayed synthetic guide pools will undoubtedly be useful for the community. However, this work has several limitations including (1) lack of adequate documentation for reproducing the analyses and (2) minimal mechanistic insight into the function of SUGP1.

****Major Comments:****

- The authors do not provide code or even pseudocode for the algorithms used to generate the features from the high-content images. If the authors are claiming that this would be a resource for the community to use then the authors need to provide an easy way for others to recreate their analysis.
- The authors mention that they will make the images from their screen publicly available, which is an essential part of making their work a useful resource for the community. However, more details need to be provided about how they will share the results. While a "data dump" of images will be useful to a narrow group of computationally savvy scientists, the broader community will require an interactive interface to enable browsing of the data. The authors should establish such a platform and make it available to reviewers of the revised manuscript to evaluate its usefulness.
- The authors highlight SUGP1 as an example for "novel mechanistic insights" - but the insights they provide are really minimal. If they authors want to claim mechanistic insights, they should experimentally address questions such as: Does SUGP1 physically interact with LIS1 mRNA? Which region of LIS1 mRNA confers regulation by SUGP1? Can the authors generate a version of LIS1 resistant to SUGP1 regulation to show that the effect of SUGP1 loss is mediated by LIS1 (and not additional factors?).

****Minor Comments:****

- Primary and Secondary antibody pairs are described nowhere in this paper. This would be impossible for anyone to recreate with just the list of primary and secondary antibodies used here.
- The authors provide no description of how the segmentation was performed or any reference to the code that they used for segmentation regarding the definition of perinuclear region. Considering so many of the results are based on these values it is important that others are able to recreate these values.
- Line 132: The authors do not explain what a min-max analysis is anywhere in the paper. This should be explained.
- There is no discussion of how the authors quantify micronuclei formation. If they state that they are the first to do this and that this is a novel technique they at the minimum need to explain the methods for quantifying micronuclei.
- Supplemental Fig 4C if a per cell intensity quantification is done I would like to see a metric for the segmentation accuracy on these cells overlaid with a cytoskeletal stain.
- It would be nice to have examples of nuclei or morphology that were excluded from downstream analysis, perhaps in a supplemental figure.
- Nowhere in the manuscript is it explained how the SUGP1 intensity measurement in Figure 6D is calculated, is this one a per well basis or a per cell basis?

2. Significance:

Significance (Required)

The generation of the dataset described in this manuscript is impressive. However, to reach its full significance and usefulness for the scientific community, the authors should provide relevant technical details, in particular of their analysis pipeline, and share the screen results in an accessible, interactive interface. If they want to claim mechanistic insights into SUGP1, more mechanistic work is required.

3. How much time do you estimate the authors will need to complete the suggested revisions:

Estimated time to Complete Revisions (Required)

(Decision Recommendation)

Between 3 and 6 months

4. Review Commons values the work of reviewers and encourages them to get credit for their work. Select 'Yes'

below to register your reviewing activity at Web of Science Reviewer Recognition Service (formerly Publons); note that the content of your review will not be visible on Web of Science.

Yes

Revision Plan

Manuscript number: RC-2023-01931

Corresponding author(s): Simon Bullock

1. General Statements [optional]

We are very grateful to the reviewers for their thorough evaluation of our manuscript and constructive feedback. Please see the cover letter for statements about goals and highlights of our study. Please note that the reviewers' introductory comments, as well as their significance statements, are not included in the point-by-point replies below.

2. Description of the planned revisions

Reviewer 1:

It is unclear to me how the authors established the limits for the quantification of localization ratios.

We previously mentioned briefly in the Materials and Methods that the 7 μm cut-off was established based on pilot experiments using nocodazole as a positive control. We should have provided more information about this process in the manuscript and will do so in the full revision as part of an extensive response to the request of Reviewer 3 for more information on how the image analysis was performed. We thank the reviewer for pointing out our oversight.

As described in methods, the perinuclear region was defined as having an outer limit of 7 μm from the nuclear envelope. However, the cells are not the same size (also seen in representative images), which could skew the calculation of ratios solely based on fixed distance limits. Have the authors considered taking into account cell size? Perhaps a more accurate calculation would be to measure the distance from the nucleus to the cell periphery for each cell and normalize this value to the cell size to account for cell size differences. The perinuclear region could then be defined as the percentage of the distance from the nuclear envelope of the normalized cell radius.

During the assay development phase of the project, we experimented with multiple ways of quantifying cargo localisation phenotypes and selecting endpoints for hit calling. This included segmenting the ring region based on the percentage of total cell area, as suggested by the reviewer. However, this procedure produced greater variance than the method with the 7 μm ring, which meant that it had weaker power to identify bona fide hits such as components of the dynein machinery. The selected metrics of 'number of spots' and 'localisation ratio' were those that were most robust during our trials with positive and negative controls. In response to the reviewer's comment, we plan to include in the full revision more detail in the Materials and Methods of our attempts to optimise the image analytics, including the efforts to factor in cell size.

Revision Plan

It is also unclear how the size and intensity of each "spot" are accounted for in the analysis as this is an important aspect of the quantification given that the "spots" are not the same size/intensity.

We will provide this information in a full revision.

Redoing this analysis would not require the authors to collect any new data but could help in gene identification, especially given that the authors only identified ~50% of the known dynein-dynactin complex components to be disrupted in their assay. These genes might have more subtle phenotypes that could be amplified by doing more precise image analysis and quantification.

We did not just rely on spot number and localisation ratio to shortlist genes from the screen for further experimentation. We also included measures of morphology, such as peroxisome spot texture (Supplementary figure 6C) and EEA1 morphology (Figure 2F), as we were aware that each metric could have limitations (more information of what each metric captures will be included in the full revision). Supplementary table 3 shows which dynein-dynactin subunits and co-factors were hits with each metric.

Given that we used many different metrics to analyse the data, we think it is unlikely that dynein-dynactin components were missed because their phenotypes escaped our attention. Currently, we think it is much more likely that these genes were missed for other reasons, including redundancy and protein perdurance (as stated in the Discussion from line 505 to 508). Nonetheless, during the revision period, we intend to examine the possibility that the missing dynein-dynactin components have subtle phenotypes by plotting their positions in scatter plots to see if they are close to the cut offs we applied.

Minor comments -

1. It would be helpful if the authors could change gene names to a bold or brighter font in scatter plots in all figures. It is hard to read the names the way they are right now.

This will be addressed in the full revision, most likely by using bold font (in the preliminary revision we have experimented with bold font in the updated scatter plots in Supplementary figure 6, which we believe makes the gene names much easier to read). We thank the reviewer for this suggestion.

Reviewer 2:

5) Using the 377 genes listed in Supplementary Table 4 I performed a Metascape analysis. The results suggest that many of the hits are proteins involved in RNA metabolism or the cell cycle and that many of the encoded proteins form complexes. Based on this I wonder whether the screen yielded many proteins that are involved in controlling the steady state levels of dynein,

Revision Plan

microtubules, or of the dynein regulators. SUGP1 is an example of this. I suggest that the authors include an extensive Metascape analysis in a new version of the manuscript.

6) On page 11 a UMAP plot is described, which is shown in Figure 4B. How were the "members of the same protein complexes, such as histones, ribosomal proteins, RNA polymerase II, the RUVBL and TRiC/CCT chaperonins, FAM160A2- AKTIP-HOOK3 and dynein-dynactin" identified?

7) How do the complexes identified in Figure 4B relate to the MCODE-based complexes identified in Metascape?

We thank the reviewer for these questions, which we address together here as they are all related to analysis of known functions of hits from the screen. In our previous submission, genes encoding proteins that are members of the same complex or operate in the same processes were identified manually by inspection of the UNIPROT database, which pools together many datasets, as well as of associated primary literature. We went down this route in order to mitigate against any errors introduced by automated curation of large datasets by a single platform. We have now included information on how the co-functional genes were defined in the legend to Figure 4B and the Materials and Methods of the preliminary revision.

In response to the reviewer's comments, we will perform the proposed analysis of the hits using Metascape or an equivalent tool. We will compare our current results to those from the Metascape (or equivalent) analysis and update the manuscript as necessary. However, it is unlikely that this process will substantially alter the conclusions of Figure 4B as the vast majority of the associations are supported by a large body of evidence. We also plan to include in the full revision an additional supplementary table that includes the source data for Figure 4B so that readers can analyse the identity and position of each gene in the plot. This will facilitate exploration of functional clusters.

Reviewer 3:

Major comments –

- The authors do not provide code or even pseudocode for the algorithms used to generate the features from the high-content images. If the authors are claiming that this would be a resource for the community to use then the authors need to provide an easy way for others to recreate their analysis.

We agree with the reviewer that we should make it clear how we generated the features from the high-content images using the Columbus software. In the full revision, we will provide documentation of how the features were generated, including with example images. We thank the reviewer for this suggestion.

Revision Plan

Minor Comments:

- Primary and Secondary antibody pairs are described nowhere in this paper. This would be impossible for anyone to recreate with just the list of primary and secondary antibodies used here.

This information will also be included in the full revision.

- The authors provide no description of how the segmentation was performed or any reference to the code that they used for segmentation regarding the definition of perinuclear region. Considering so many of the results are based on these values it is important that others are able to recreate these values.

As described above, we will provide documentation of how segmentation was performed in the full revision, with further examples of the output also shown.

- There is no discussion of how the authors quantify micronuclei formation. If they state that they are the first to do this and that this is a novel technique they at the minimum need to explain the methods for quantifying micronuclei.

There seems to be a slight misunderstanding here, which results from us not being sufficiently clear in the previous submission. We did not mean to emphasise that our quantification method was novel but rather that other genome-scale screens had not looked specifically at micronuclei formation. We have now modified the language in lines 191 to 194 to address this point. We will also provide details in the full revision of how the quantification was performed for the micronuclei analysis.

- Supplemental Fig 4C if a per cell intensity quantification is done I would like to see a metric for the segmentation accuracy on these cells overlaid with a cytoskeletal stain.....It would be nice to have examples of nuclei or morphology that were excluded from downstream analysis, perhaps in a supplemental figure.

These are also excellent points that will help us further clarify how these image analyses were performed. In the full revision, we intend to provide new supplementary data including example images and outputs of segmentation, as suggested.

- Nowhere in the manuscript is it explained how the SUGP1 intensity measurement in Figure 6D is calculated, is this one a per well basis or a per cell basis?

Figure 6D shows intensity measurements for DYNC1I2 and LIS1, not SUGP1. Nonetheless, the reviewer makes another important point. We have now clarified in this legend that the values per independent experiment shown in the graph were derived from aggregating per cell intensity values at the per well level (at least 4 wells from a 384-well plate per experiment) and then

calculating the mean per well value per experiment to show on the plot. In the full revision we will also include more information on the process used for calculating intensity values from immunofluorescence images throughout the study.

3. Description of the revisions that have already been incorporated in the transferred manuscript

Please insert a point-by-point reply describing the revisions that were already carried out and included in the transferred manuscript. If no revisions have been carried out yet, please leave this section empty.

Reviewer 1:

crDYNC1H1 data in Figure 1: The images in Figure 1E and 1G for the dynein control (crDYNH1C1) show some clustering around the cell nucleus while crLIS1 knockout shows no perinuclear clustering. Is this expected? Shouldn't dynein knockout prevent perinuclear clustering? Is it possible that crDYNH1C1 does not lead to a complete knockout?

We did not choose a particularly representative image for *crDYNC1H1* in Figure 1E and have now corrected this oversight. Nonetheless, the reviewer is correct that *crDYNC1H1* does not generally cause as strong a phenotype at *crLIS1* (quantified on multiple occasions in the manuscript). This does indeed correlate with an incomplete removal of DYNC1H1 in the cell population, as shown by the quantification of immunofluorescence in Figure 1C. We do not expect all crRNA pools to cause a complete loss of protein in the population due to incomplete cutting at target sites across the population or perdurance of protein that was made before crRNA treatment. Moreover, because *LIS1* is a haploinsufficient gene, it may be more sensitive to incomplete perturbation than *DYNC1H1*. We have now modified the text to make these points clear (lines 140 – 143):

'crLIS1 caused a stronger dispersal phenotype than crDYNC1H1 (Figure 1F), which may be related to more efficient reduction in levels of the target protein (Figure 1B and C) or the strong sensitivity of dynein-based transport to even partial reductions in LIS1 concentration²⁰⁻²³.'

Minor comments:

2. Line 185 the authors say: "We also analysed the induction of micronuclei (Supplementary figure 6B), which to our knowledge has not been assessed in earlier screens." What screens are the authors referring to? Could you add references here?

We have now cited the screens that assess nuclear morphology but did not assess micronuclei formation (line 194).

Revision Plan

3. Line 256: "Each cargo was assayed in two independent screens, in which there was good agreement in general between the effects of the crRNA pools (Supplementary figure 8)." The authors also indicate in the legend for Supplementary figure 8 that "The only metric with a poor R2 score (proportion of cells with two γ - Tubulin puncta) was not used for hit calling." However, the EEA1 localization ratio also shows poor R2 score, shouldn't this screen also be excluded? In general, what was the cutoff for R2 score? This information should be included.

We were able to cross-correlate the data for EEA1 localisation ratio with that from EEA1 morphology, which had a high R² score (0.4902 vs 0.2296), when selecting hits (Figure 2F). This process improves confidence in the hits that were selected. Moreover, although the R² score for EEA1 localisation ratio was sub-optimal, it was substantially higher than the score for 'proportion of cells with two γ -Tubulin puncta' (0.07254). We have addressed these points in the legend to Supplementary figure 8.

Reviewer 2:

Major comments -

1) On page 6 the authors state they used 61 x 384 wells. This equals 23,424 wells, but the authors state they analysed (8,150,065 cells from) 24,576 wells. What causes this difference in number?

We inadvertently including some additional control wells in the well calculation and therefore misreported these numbers. The correct numbers (23,424 wells and 8,150,247 viable cells) have been used in the preliminary revision. We thank the reviewer for spotting our mistake.

3) The screen was developed based on the U-2 OS PEX line, in which tethering of dynein to peroxisomes is achieved by addition of rapamycin acting via a split BICD2 protein. Thus, the assay depends on the BICD2 adaptor. Is this limiting when one is looking for dynein regulators?

Whilst we thank the reviewer for evaluating this aspect of the manuscript, we do not understand this comment. In our genome-wide screen, we simultaneously assayed peroxisome and early endosome localisation, which reports on dynein-based transport by a second type of adaptor. This was a very large undertaking and a key strength of our approach.

3) Related to the question above, the authors do not recover BICD1 in their screen. Is this because BICD1 is not expressed in the cell systems used or is there another reason?

BICD1 is moderately expressed in U2OS cells according to proteomic data (Beck et al., 2011; <https://www.emboPress.org/doi/full/10.1038/msb.2011.82>). We assume that *BICD1* was (i) not returned in the early endosome screen because BICD family members are not implicated in trafficking of these cargoes and (ii) not returned in the peroxisome screen as these organelles were transported by induced tethering with BICD2. We now mention briefly (lines 214 and 215)

Revision Plan

that *BICD1* was not recovered despite being expressed in U2OS cells and cite the Beck et al. study:

'The gene encoding the paralogous BICD1 protein was not recovered, despite being expressed at moderate levels in U-2 OS cells⁴⁸.'

4) It has very recently been shown (doi.org/10.1038/s41467-023-38116-1) that BICD2 phosphorylation by CDK1 in the G2 phase of the cell cycle promotes its interaction with PLK1. This is followed by PLK1 phosphorylation in the N-terminus of BICD2, which in turn facilitates interaction with dynein and dynactin, allowing the formation of active motor complexes. Thus, adaptor activation through phosphorylation regulates dynein activity. In the present manuscript the authors use PLK1 as a read-out of cell viability. However, PLK1 also appears to regulate dynein via BICD2 phosphorylation. Given the latest results would the authors interpret their PLK1 data differently? Would it be preferable to screen for regulators of dynein activity in non-dividing cells?

We thank the reviewer for pointing out this interesting study. *PLK1* is a strong hit in many genome-wide screens that are unrelated to dynein-based trafficking because of the proliferation defects associated with its disruption (<https://orcs.thebiogrid.org/Gene/5347>). For this reason, it is an effective positive control for editing efficiency in screening projects (e.g. Strezoska et al. 2017; Ross-Thriepland et al. 2020). The proliferation phenotype of *crPLK1* is almost certainly unrelated to its interaction with BICD2, as *crBICD2* does not elicit a mitotic phenotype even though it is active in the peroxisome relocalisation assay. We now show in Supplementary figure S6A that *crBICD2* does not affect proliferation and mention this in the context of the study referred to by the reviewer (lines 183 – 186).

'Despite PLK1 having recently been implicated in regulation of BICD2 function⁴², the library copy of crBICD2 did not affect cell viability (Supplementary figure 6A). This observation suggests that PLK1's role in cell proliferation does not involve BICD2.'

Reviewer 3:

Major comments -

- The authors highlight SUGP1 as an example for "novel mechanistic insights" - but the insights they provide are really minimal. If they authors want to claim mechanistic insights, they should experimentally address questions such as: Does SUGP1 physically interact with LIS1 mRNA? Which region of LIS1 mRNA confers regulation by SUGP1? Can the authors generate a version of LIS1 resistant to SUGP1 regulation to show that the effect of SUGP1 loss is mediated by LIS1 (and not additional factors?).

We respectfully disagree that the insights from our work on how SUGP1 affects dynein-based transport are 'really minimal'. In order to provide a proof-of-principle of how our screen can lead

to new knowledge, we performed extensive transcriptomic analysis of SUGP1-edited cells, as well as functional experiments to test a strong hypothesis generated from this approach. Our data indicate that *LIS1* is a key target of SUGP1 in the context of dynein-based trafficking and that this effect does not involve changes in *LIS1* splicing or 3'-end usage. As Reviewer 1 points out, this work provides an 'interesting new way of regulating dynein function'. Addressing the detailed molecular mechanism by which SUGP1 controls LIS1 levels would constitute a new long-term study and therefore goes well beyond the scope of what can be included in the paper, which already covers a lot of ground. Therefore, in the preliminary revision we have taken the steer of the reviewer to remove the specific term 'mechanistic insight', which after all has different meanings to different people.

Minor comments:

- Line 132: The authors do not explain what a min-max analysis is anywhere in the paper. This should be explained.

We now see that min-max was not a helpful term. We have now changed this to 'feasibility study' (line 132).

4. Description of analyses that authors prefer not to carry out

Please include a point-by-point response explaining why some of the requested data or additional analyses might not be necessary or cannot be provided within the scope of a revision. This can be due to time or resource limitations or in case of disagreement about the necessity of such additional data given the scope of the study. Please leave empty if not applicable.

Reviewer 1:

DYNC1H1 expression data: Given that this is a proof-of-principle control in the assay, a more detailed validation of DYNHC1 knockout using western blotting and RT-qPCR, in addition to the validation shown in Figure 1C would strengthen the claim that this control works as expected. These experiments should be fast and easy to do.

As described above, it was regrettable that we did not choose a more representative example of the *crDYNC1H1* phenotype in Figure 1E. This error has now been corrected. We do not believe that performing Western blotting and RT-qPCR will provide additional information to that gained from the quantification of the IF in Figure 1C, which already shows that *crDYNC1H1* causes a less strong reduction of its protein target than *crLIS1*. Moreover, western blotting and RT-qPCR would only report at the bulk level, whereas the IF provides useful information about the heterogeneity of targeting at the cell-to-cell level. Additionally, RT-qPCR would only look at changes in mRNA levels, which may not cause proportional changes in protein levels (e.g. see DYNC1I2 data in Figure 6D). We should also point out that the ability of *crDYNC1H1* to significantly disperse dynein's cargoes was validated on multiple occasions in the manuscript

Revision Plan

(Figure 1E, 2F, 3A, 5A and Supplementary figure S6C, S7 and S12), showing that this control works as expected. We also recovered other dynein-dynactin subunits and co-factors in the peroxisome and endosome screens, further demonstrating that the localisation processes we assay are dependent on dynein. In section 2 of the revision plan we discussed the potential reasons for incomplete dispersion of dynein cargoes when DYNC1H1 is targeted.

Off target considerations: The use of multiple crRNAs together to target a single gene can increase off-target effects, however, the authors never test for off-target phenotypes or address the possibility of off-targeting. Can the authors show using a few examples that their approach does not lead to significant off-targeting? This should also be addressed in the text.

Multiplexing of crRNAs is increasingly used in CRISPR screens to produce consistently efficient and reliable gene disruption at target sites. Indeed, in pilot experiments we found that using individual crRNAs was often not sufficient to cause disruption of target gene function in the majority of cells. As described in more detail below, the data in the previous submission make a strong case against a significant contribution of off-target effects to our results. This is consistent with extensive optimisation of the crRNA library by the manufacturer to minimise the potential for off-target cutting (e.g. none of the crRNAs for hits selected for our secondary screens has a perfect match to another sequence in the genome). In addition, transient expression of both Cas9 (from its mRNA) and synthetic gRNA is expected to reduce off-target editing compared to constitutively active expression in pooled lentiviral screens. We regret that we did not draw attention to these issues in the previous submission and thank the reviewer for prompting us to address this matter.

One strong, systematic piece of evidence that off-targets do not contribute significantly to the phenotypes observed is the tight clustering of phenotypic fingerprints associated with crRNA pools that target known co-functional genes (Figure 4B). This would not be seen if the phenotypes were driven significantly by disruption of non-target genes. We also set out to specifically address the contribution of off-targets by taking 22 hits from the secondary screen and attempting to validate the phenotypes with completely independent crRNAs (Figure 5). We were able to validate mislocalisation of dynein cargoes in 19/22 cases. We tested the editing efficiency of two of the pools that did not replicate a cargo mislocalisation phenotype and found this to be relatively low (Supplementary figure S13), offering a plausible explanation for the failure to reproduce earlier results with these reagents. We also provided in depth analysis that the *crSUGP1* phenotype is not associated with an off-target effect through replication with the individual crRNAs in the pools (Supplementary figure S14A), as well as phenotypic rescue with a cDNA (Figure 6E).

We therefore believe that genome-wide assessment of editing events of a few crRNAs, which is suggested by the reviewer as a possible experiment, would not add significantly to the study. Moreover, this sequencing-based approach is also very time-consuming and expensive and we do not have the resources to deliver it. Nonetheless, we agree that the treatment of off-target effects in the manuscript needs to be improved. In the preliminary revision we have therefore made changes from lines 369 – 372 as follows:

Revision Plan

'The strong clustering of phenotypic fingerprints for crRNAs targeting genes that encode components of the same protein complexes strongly suggested that the phenotypes produced by our procedures are not driven significantly by off-target effects of the crRNAs. To further evaluate the potential for off-target effects in our phenotypic readouts, we targeted a subset of hits from the secondary screen with unrelated pools of crRNAs.'

We also include an important caveat that off-targets cannot be ruled out entirely (line 390 – 392):

'Whilst, we cannot rule out their contribution to a small subset of phenotypes, these data provide further evidence that off-target effects do not significantly drive the phenotypes we observe with our procedures.'

We also plan to include more details in the Materials and Methods of the full revision on the design of the crRNA library by the supplier and how it minimises off-target effects. We feel that, collectively, these changes will greatly improve the treatment of off-target effects in the study. We thank the reviewer for leading us to make these changes.

Reviewer 2:

N number of genome-wide screen: More importantly, the authors target 18,253 genes with four guides per gene. If I understand correctly these four guides per gene are present in a single well and the high-content imaging experiment was only done once. Although many cells were analysed per well (four fields of view per well; median of 345 cells analysed per well) and results are interesting and appear solid, I do think a replicate experiment is necessary.

We were very surprised to see this comment. Of course, running a duplicate of any genome-wide screen would improve the quality of the output by removing some of the false positives and reducing the number of false negatives. However, it is simply not practical to perform such a time consuming and (extremely) expensive experiment twice. For this reason, the strategy of analysing data from only one run is commonplace for high-throughput imaging-based screens. To give some context for our specific pipeline, a single screen takes three weeks of full-time use to the AstraZeneca functional genomics platform (to which we no longer have access) and the cost of the crRNA library is > £80K.

In any case, the problem of false positives in a single genome-wide screen can be removed by rigorous downstream efforts to validate hits. In our case, we ran two independent biological replicates each with four technical replicates with the hits from the primary screen, providing high confidence validated hits. False negatives are of course a risk with any genome-wide screen. Nonetheless, we recovered ~200 validated hits from the screen, which represents a rich source of new hypotheses for the field. Following the reviewer's comment, we now give an example in the manuscript of how these hits can be used to bootstrap the identification of players in dynein-based trafficking that may have been missed in the primary screen (lines 509 – 511).

'Analysis of biochemical interaction partners of these proteins may offer a means to identify additional players in dynein-based trafficking that were refractory to our CRISPR screening approach.'

Reviewer 3:

- The authors mention that they will make the images from their screen publicly available, which is an essential part of making their work a useful resource for the community. However, more details need to be provided about how they will share the results. While a "data dump" of images will be useful to a narrow group of computationally savvy scientists, the broader community will require an interactive interface to enable browsing of the data. The authors should establish such a platform and make it available to reviewers of the revised manuscript to evaluate its usefulness.

The images from the project will be hosted on the established AstraZeneca Open Innovation platform, from which they will be able to be batch downloaded. Whilst, just like with transcriptomic datasets, reprocessing of the data to mine new information will require computational skills, other users will be able to visualise images for genes (or sets of genes) that hold a particular interest. We do not believe that constructing a website to allow browsing of individual images will add significant value to the resource. Browsing individual images in isolation will have limited use without being able to analyse the numerous controls simultaneously. For the vast majority of users, much of the value of the resource comes from the quantitative data that we have already extracted from the images. These data can be easily plotted with users' favourite tools, even for those that are not computationally savvy.

Moreover, building and maintaining a website to browse individual images will be an enormous amount of work and we currently do not have the resources to deliver this. We note that several other large-scale imaging-based CRISPR screens (e.g. Yan et al. 2021, J Cell Biol, <https://doi.org/10.1083/jcb.202008158>; Kanfer et al., J Cell Biol, doi.org/10.1083/jcb.202006180; Ross-Thrieland et al., SLAS Discovery 2020, [https://slas-discovery.org/article/S2472-5552\(22\)06576-5/fulltext](https://slas-discovery.org/article/S2472-5552(22)06576-5/fulltext)) have not produced an interface to allow their images to be browsed, presumably due to the issues described above.

June 21, 2023

Re: JCB manuscript #202306048T

Dr. Simon L Bullock
MRC Laboratory of Molecular Biology
Cell Biology
Francis Crick Avenue
Cambridge CB2 0QH
United Kingdom

Dear Dr. Bullock,

Thank you for submitting your transfer manuscript entitled "Genome-scale requirements for dynein-based trafficking revealed by an arrayed CRISPR screen" from Review Commons. We agree that a suitably revised study seems potentially appropriate as a JCB Tool, and therefore invite you to submit a revision as outlined in your response to the Referees from Review Commons.

GENERAL GUIDELINES:

Text limits: Character count for an Transfer is < 40,000, not including spaces. Count includes title page, abstract, introduction, results, discussion, and acknowledgments. Count does not include materials and methods, figure legends, references, tables, or supplemental legends.

Figures: Transfers may have up to 10 main text figures. Figures must be prepared according to the policies outlined in our Instructions to Authors, under Data Presentation, <https://jcb.rupress.org/site/misc/ifora.xhtml>. All figures in accepted manuscripts will be screened prior to publication.

*****IMPORTANT:** It is JCB policy that if requested, original data images must be made available. Failure to provide original images upon request will result in unavoidable delays in publication. Please ensure that you have access to all original microscopy and blot data images before submitting your revision. ***

Supplemental information: There are strict limits on the allowable amount of supplemental data. Transfers may have up to 5 supplemental figures. Up to 10 supplemental videos or flash animations are allowed. A summary of all supplemental material should appear at the end of the Materials and methods section.

Please note that JCB now requires authors to submit Source Data used to generate figures containing gels and Western blots with all revised manuscripts. This Source Data consists of fully uncropped and unprocessed images for each gel/blot displayed in the main and supplemental figures. Since your paper includes cropped gel and/or blot images, please be sure to provide one Source Data file for each figure that contains gels and/or blots along with your revised manuscript files. File names for Source Data figures should be alphanumeric without any spaces or special characters (i.e., SourceDataF#, where F# refers to the associated main figure number or SourceDataFS# for those associated with Supplementary figures). The lanes of the gels/blots should be labeled as they are in the associated figure, the place where cropping was applied should be marked (with a box), and molecular weight/size standards should be labeled wherever possible.

The typical timeframe for revisions is three to four months. While most universities and institutes have reopened labs and allowed researchers to begin working at nearly pre-pandemic levels, we at JCB realize that the lingering effects of the COVID-19 pandemic may still be impacting some aspects of your work, including the acquisition of equipment and reagents. Therefore, if you anticipate any difficulties in meeting this aforementioned revision time limit, please contact us and we can work with you to find an appropriate time frame for resubmission. Please note that papers are generally considered through only one revision cycle, so any revised manuscript will likely be either accepted or rejected.

When submitting the revision, please include a cover letter addressing the reviewers' comments point by point. Please also

highlight all changes in the text of the manuscript.

Thank you for this interesting contribution to Journal of Cell Biology. You can contact us at the journal office with any questions, cellbio@rockefeller.edu or call (212) 327-8588.

Sincerely,

Melissa Gardner
Monitoring Editor

Andrea L. Marat
Senior Scientific Editor

Journal of Cell Biology

We are very grateful to the reviewers for their thorough evaluation of our work and their helpful feedback. We have now revised the manuscript according to the reviewer comments and a revision plan approved by the editors at *Journal of Cell Biology*. Changes include the provision of much more information on the image analysis methods (including a new, 29-page supplemental methods document), further consideration of the potential for off-target effects and false negatives in the screen, and toning down, caveating or qualifying some conclusions. We include below a point-by-point response to the reviewer comments in which the line numbers refer to the merged PDF generated by the manuscript submission system. To facilitate evaluation of the revision, we have uploaded a version of the manuscript with significant changes highlighted in blue text.

Reviewer #1 (Evidence, reproducibility and clarity (Required)):

Cytoplasmic dynein-1 (dynein) is the predominant minus-end directed microtubule-based motor involved in the transport of numerous cellular cargoes in addition to mitotic functions. Although in vitro analysis of dynein assembly and interactions are becoming more common, cell biological studies that aim at identifying different dynein cargoes and functions are lacking. To shed light on dynein's function, Wong et al., performed a genome-wide CRISPR loss-of-function screen using peroxisome-tethered and endosome localization assays as readouts. Detailed data analysis and supervised and unsupervised phenotypic clustering of targets from an extensive RNA gene library (~20,000 genes) revealed ~200 genes disrupted in cargo trafficking processes. The authors also showed that a novel gene SUGP1, identified in their screen, promotes the expression of a critical dynein activator, LIS1. The generated datasets provide a rich library of genes that can be further mined by other researchers in the field.

Major comments:

This manuscript reads well, and the conclusions are mostly supported by experimental data. The authors went to great lengths to optimize their high throughput assay setup by testing different cell lines and transfection conditions and included different positive and negative controls, which is a strength of the study. The use of two functional readouts (early endosome and BICD2-driven peroxisome distribution) in the initial screen followed up by a validation screen of a smaller subset of genes using readouts for dynein disruption phenotypes (Golgi fragmentation, lysosome clustering) is another strength. In addition, the follow-up identification of an RNA-binding protein SUGP1 as a regulator of LIS1 mRNA levels provides an interesting new way of regulating dynein function.

I have a few concerns about the experimental design and conclusions.

1. The images in Fig. 1E and 1G for the dynein control (crDYNHC1) show some clustering around the cell nucleus while crLIS1 knockout shows no perinuclear clustering. Is this expected? Shouldn't dynein knockout prevent perinuclear clustering? Is it possible that crDYNHC1 does not lead to a complete knockout?

Regrettably, we did not choose a particularly representative image of dispersion for *crDYNC1H1* in Fig. 1 E. We have now corrected this oversight. Nonetheless, the reviewer is correct that *crDYNC1H1* generally does not cause as strong a dispersion phenotype as *crLIS1* (quantified on multiple occasions in the manuscript). This does indeed correlate with less complete removal of DYNC1H1 than LIS1 with the respective crRNA pools, as shown by the quantification of immunofluorescence (IF) in Fig. 1B and C. We have now modified the text to make this point clear (lines 129–131):

'crLIS1 caused a stronger peroxisome dispersal phenotype than crDYNC1H1 (Fig. 1 F), which may be related to greater reduction in the level of its target protein (Fig. 1, B and C).'

We do not expect all crRNA pools to cause a complete loss of protein in the population due to incomplete cutting at target sites or differential perdurance of proteins that were made before crRNA treatment.

Given that this is a proof-of-principle control in the assay, a more detailed validation of DYNHC1 knockout using western blotting and RT-qPCR, in addition to the validation shown in Fig. 1C would strengthen the claim that this control works as expected. These experiments should be fast and easy to do.

We thank the reviewer for the suggestion but we do not believe that western blotting or RT-qPCR will provide additional information to that gained from the quantification of the IF in Fig. 1C. Moreover, western blotting and RT-qPCR would only report at the bulk level, whereas the IF provides useful information about the heterogeneity of targeting at the cell-to-cell level. RT-qPCR would also only look at changes in mRNA levels, which may not cause proportional changes in protein levels (e.g. see DYNC112 data in Fig. 7 C and D). We also want to point out that the ability of *crDYNC1H1* to significantly disperse dynein's cargoes was validated on multiple occasions in the manuscript (Fig. 1 E and F, 2 F, 6, and S7), showing that this control works as expected, and that we also routinely validated phenotypes and assays with *crLIS1*. We also recovered other dynein-dynactin subunits and co-factors in the peroxisome and endosome screens, further demonstrating that the localisation processes we assay are dependent on dynein.

2. The use of multiple crRNAs together to target a single gene can increase off-target effects, however, the authors never test for off-target phenotypes or address the possibility of off-targeting. Can the authors show using a few examples that their approach does not lead to significant off-targeting? This should also be addressed in the text.

Multiplexing of crRNAs is increasingly used in CRISPR screens to efficiently disrupt target genes by overcoming the performance variability of individual guides (e.g. PMID 27798563; bioRxiv 10.1101/2023.01.03.522655). Indeed, in pilot experiments we found that using individual crRNAs often did not disrupt the target gene in the majority of cells (now mentioned on lines 619–621). To reduce the risk of off-target effects with crRNA pools, we selected a commercial crRNA library from Horizon that has been extensively optimised to minimise the number of potential off-target cuts. This is achieved through Horizon's Edit-R algorithm, which outperforms other algorithms for identifying guides with potential off-target effects by also taking into account gaps in the alignment of genomic sequences with the guide, as well as the position of mismatches (i.e. in the non-seed or seed positions (more or less likely, respectively, to be tolerated by Cas9)) when ranking crRNAs for specificity. Horizon have informed us that none of the crRNAs selected for the library has a perfect match in the genome other than the target site and only a very small proportion (4.78%) have a single mismatch or gap in the non-seed sequence vs another sequence (a situation necessitated by design space being limited for their targets).

One strong (and systematic) piece of evidence that off-targets do not contribute significantly to the phenotypes we observe is the tight clustering of phenotypic fingerprints associated with crRNA pools that target genes encoding components of the same protein complex (Fig. 5 B and S6 (explorable in the 'Supplemental phenotypic heatmap' file and Table S8)). This would not be seen if the phenotypes were driven to a significant extent by disruption of non-target genes.

We also set out to experimentally address the contribution of off-target effects by taking 22 hits from the secondary screen and attempting to validate the phenotypes with completely independent crRNAs (themselves selected based on the lowest likelihood of off-target cutting (line 364)) (Fig. 6 and Fig. S7). We now appreciate that we should have made the rationale for this approach explicit and have corrected this mistake (see below). In these experiments,

we were able to validate mislocalisation of dynein cargoes in 19/22 cases. We tested the efficiency of target gene editing for two of the pools that did not reproduce a cargo mislocalisation phenotype and found this to be relatively low (Fig. S8 A–C); these results offer a plausible explanation for the failure to reproduce the initial phenotype. We also showed that the *crSUGP1* phenotype is not associated with an off-target effect through its replication with multiple individual crRNAs (Fig. S8 D), as well as its rescue with a crRNA-resistant cDNA (Fig. 7 E).

We therefore believe that genome-wide assessment of editing events of a few crRNAs, which the reviewer raises as a possible new experiment, would not add substantially to the study. Moreover, this sequencing-based approach would be time-consuming and expensive and unfortunately we do not have the resources to deliver it.

Nonetheless, the reviewer makes a very important point that the treatment of off-target effects in the manuscript needs to be improved. In the revision we have therefore made changes from lines 358–362 as follows:

‘The strong clustering of phenotypic fingerprints for crRNAs that target genes encoding components of the same protein complex strongly suggests that the phenotypes produced by our procedures are not driven significantly by off-target effects. To further evaluate the potential for off-target effects in our phenotypic readouts, we targeted a subset of hits from the secondary screen with unrelated crRNA pools...’

We also include an important caveat that off-targets cannot be ruled out entirely (lines 379–381):

‘Thus, whilst we cannot rule out off-target effects contributing to a subset of phenotypes, these data provide further evidence that they do not significantly drive the effects we observe.’

We have also included in the Materials and Methods (lines 610–617) the above points about minimising the risk of off-target cutting through library crRNA design. We feel that, collectively, these changes significantly improve the treatment of off-target effects in the study. We thank the reviewer for leading us to make these modifications.

3. It is unclear to me how the authors established the limits for the quantification of localization ratios. As described in methods, the perinuclear region was defined as having an outer limit of 7 μm from the nuclear envelope. However, the cells are not the same size (also seen in representative images), which could skew the calculation of ratios solely based on fixed distance limits. Have the authors considered taking into account cell size? Perhaps a more accurate calculation would be to measure the distance from the nucleus to the cell periphery for each cell and normalize this value to the cell size to account for cell size differences. The perinuclear region could then be defined as the percentage of the distance from the nuclear envelope of the normalized cell radius.

We should have made this clearer as well. During the assay development phase of the project, we randomly selected a set of images of U-2 OS PEX cells treated with the non-targeting control (NTC) crRNA pool, the *crLIS1* pool, or nocodazole. We used these images to experiment with multiple ways of quantifying peroxisome localisation phenotypes and selecting endpoints for hit calling. This included assessing the total number of fluorescent spots per cell, or the number of spots in cytoplasmic regions that were defined based on distance thresholds of different absolute sizes, or resizing the ring regions based on the percentage of total cell area (analogous to the approach suggested by the reviewer). This included attempting to increase the ability to detect subtle phenotypes by segmenting the cytoplasm into three regions. An absolute distance threshold of 7 μm from the nuclear envelope had less variance than a cut-off based on percentage of cell area so we reasoned it

would be the most effective ring-based method for identifying hits in a large-scale screen for peroxisome mislocalisation. We now explain how we developed the image analytics in the Materials and Methods (lines 679–705).

We are grateful that the reviewer led us to re-evaluate this part of the analysis as we discovered an error in how we described the analysis of the early endosome localisation. For these data, we did indeed resize cell regions based on percentage of cell area, as suggested by the reviewer, as this was a robust method for analysing these data in pilot analysis with cells treated with *crLIS1*, NTC or nocodazole. The relevant section of the Materials and Methods (line 705–715) has been modified accordingly.

It is also unclear how the size and intensity of each "spot" are accounted for in the analysis as this is an important aspect of the quantification given that the "spots" are not the same size/intensity.

The 'find spot' algorithm in the Columbus software package estimates background image intensity and then identifies puncta of substantially higher intensity. To ensure that the spot analysis was accurate, we iteratively modified the settings using our NTC, *crLIS1* and nocodazole reference image sets. This involved tuning parameters including 'detection sensitivity', which defines how intense a spot must be for detection, and 'splitting coefficient' which determines if adjacent spots are split or merged, and evaluating the proportion of real spots that were identified. In the case of RFP spot detection, a further background subtraction step was applied to ensure that the desired spots were captured. We have provided this information on lines 691–696 of the Materials and Methods, with details of settings included in the new 'Supplemental image analysis file'.

Redoing this analysis would not require the authors to collect any new data but could help in gene identification, especially given that the authors only identified ~50% of the known dynein-dynactin complex components to be disrupted in their assay. These genes might have more subtle phenotypes that could be amplified by doing more precise image analysis and quantification.

We did not just rely on one metric to shortlist genes from the genome-wide screen for further experimentation, as summarised in Tables S2 and S3. For example, we also included different measures of spot number, as well as morphology in the case of early endosomes, as we were aware that each metric could have limitations that might limit detection of subtle phenotypes (more information on metrics that were captured is now included in the 'Supplemental image analysis file'). To illustrate the value of this process, we have reproduced below scatter plots of some of the primary screen data from the manuscript with the position of all known dynein-dynactin components and regulators now marked (Fig. R1). The plots show that different dynein-dynactin components are recovered as hits with different metrics (Table S3 gives an overview of which dynein-dynactin subunits and co-factors were hits with each metric applied to the primary screen data). This was also the case for non-dynein-dynactin components (Table S2).

Fig. R1 also shows that genes encoding several of the known components of the transport machinery are far away from the threshold for hit calling, consistent with our previous assumption that these do not have subtle phenotypes that are missed by our pipeline. These genes may be aphenotypic for other reasons, including redundancy, protein perdurance or inefficient targeting (as previously stated in the Discussion (lines 491–494)). However, the reviewer is correct that we cannot rule out some important regulators of dynein-based trafficking being lost from our pipeline because they have subtle phenotypes, and we now mention this caveat on lines 494–497.

'We cannot, however, rule out that some bona fide regulators of dynein function have subtle phenotypes that might be picked up with novel image analysis tools...'

Nonetheless, we believe it is prudent to focus on the hits with the strongest phenotypes given the 'one shot' format of a primary screen and the significant costs that would be associated with ordering synthesis of additional crRNA pools for validation (we mention on lines 741–742 of the Materials and Methods the limited capacity to take hits forward as further justification for our hit selection criteria).

Figure R1: Examples of position of known dynein-dynactin components and regulators in primary screen metrics. Regions shaded in light blue correspond to region used for hit calling for each metric.

Minor comments:

1. It would be helpful if the authors could change gene names to a bold or brighter font in scatter plots in all figures. It is hard to read the names the way they are right now.

We have changed the font in scatter plots to bold, which we believe has improved legibility substantially. We thank the reviewer for this suggestion.

2. Line 185 the authors say: "We also analysed the induction of micronuclei (Supplementary figure 6B), which to our knowledge has not been assessed in earlier screens." What screens are the authors referring to? Could you add references here?

We have now cited the screens that assess nuclear morphology but did not assess micronuclei formation (line 180–181).

3. Line 256: "Each cargo was assayed in two independent screens, in which there was good agreement in general between the effects of the crRNA pools (Supplementary figure 8)." The authors also indicate in the legend for Supplementary figure 8 that "The only metric with a poor R2 score (proportion of cells with two γ -Tubulin puncta) was not used for hit calling." However, the EEA1 localization ratio also shows poor R2 score, shouldn't this screen also be excluded? In general, what was the cutoff for R2 score? This information should be included.

The R² score for EEA1 localisation ratio (0.2296) was substantially higher than the score for 'proportion of cells with two γ -Tubulin puncta' (0.07254). We now include the cut-off of 0.2 for the R² value in the legend to this supplemental figure (now Fig. S5), as requested. The hits for EEA1 localisation ratio included several known components of the transport machinery, as well as the FHF complex and PAFAH1B2, suggesting that this metric has significant value. We were also able to combine the data for EEA1 localisation ratio with features related to EEA1 morphology, which had a higher R² score (0.4902) when performing phenotypic fingerprinting.

Reviewer #1 (Significance (Required)):

This work is the first genome-wide loss-of-function CRISPR study (to my knowledge) aimed at identifying dynein-driven trafficking disruption phenotypes. In general, the data generated in this study will enrich the field's understanding of how dynein is regulated and how it achieves its broad cargo and functional specificity. This manuscript will also provide a resource and experimental setup for the design of other genome-wide loss-of-function CRISPR studies.

I have broad expertise in the cytoskeleton field with a detailed understanding of dynein's function from a mechanistic and functional perspective. I have minimal experience with high throughput screening, but I am experienced with CRISPR-based assays and cell imaging techniques.

Reviewer #2 (Evidence, reproducibility and clarity (Required)):

In this manuscript the authors have conducted a genome-wide CRISPR loss-of-function screen in human cells to find regulators of cytoplasmic dynein, a microtubule-based motor that plays a major role in the transport of cargo towards microtubule minus ends. The screen was carried out to address how dynein is synthesised and assembled, and how its activity is controlled to enable the motor to selectively transport a wide variety of cargoes. Cells were fixed cells 72 hours after transfection and fluorescently stained for intracellular markers. Several read-outs were used in the screen, of which the major ones were the distribution of dynein-tethered peroxisomes and early endosomes. The authors used a 384 well format (61 unique 384-well plates) and a fluorescence microscopy-based imaging readout to gauge dynein activity. From a guide RNA library targeting 18,253 genes, the authors recovered 195 validated hits. For one gene (SUGP1) follow-up studies demonstrate that the protein encoded by this gene controls the levels of the dynein activator LIS1 and thereby promotes cargo trafficking by dynein. The dataset reported here represents a source for investigating proteins that might be involved in minus-end microtubule-based transport, as well as in other aspects of cellular organisation that were captured in the high-content imaging approach.

I find this an interesting and well written resource manuscript, both from the perspective of how to conduct and analyse a high-content imaging screen, as well as from the dynein biology view. Results presented in this manuscript deserve follow-up studies. I do have a number of comments.

Major comments

1) On page 6 the authors state they used 61 x 384 wells. This equals 23,424 wells, but the authors state they analysed (8,150,065 cells from) 24,576 wells. What causes this difference in number?

We inadvertently included some additional controls in the calculation and therefore misreported these numbers. The correct numbers (23,424 wells and 8,150,247 viable cells)

have been included in the revision (line 156–157). We apologise for our mistake and thank the reviewer for spotting it.

More importantly, the authors target 18,253 genes with four guides per gene. If I understand correctly these four guides per gene are present in a single well and the high-content imaging experiment was only done once. Although many cells were analysed per well (four fields of view per well; median of 345 cells analysed per well) and results are interesting and appear solid, I do think a replicate experiment is necessary.

If the reviewer is suggesting that we run the whole-genome screen again, we find this comment very surprising. Of course, running a duplicate of any genome-wide screen would improve the quality of the output by removing some of the false positives and reducing the number of false negatives. However, it is simply not practical to perform such a time consuming and expensive experiment twice. For this reason, the strategy of analysing data from only one run is commonplace for high-throughput imaging-based screens. In our case, a single screen with our pipeline takes three weeks of full-time use to the AstraZeneca functional genomics platform (to which we no longer have access). Moreover, the use of the genome-wide crRNA library, transfection reagents and antibody staining at this scale means that consumable costs for a single screen are extremely high. In any case, the problem of false positives in a single genome-wide screen can be removed by rigorous downstream efforts to validate hits, which is perhaps what the reviewer means when they refer to replication. We ran two independent biological replicates – each with four technical replicates – with the shortlisted hits from the primary screen. This point has now been made explicit in the manuscript (line 255–256). This process provided ~200 validated hits. False negatives are of course a risk with any genome-wide screen. Nonetheless, the large number of validated hits from the screen already represents a rich source of new hypotheses for the field. Following the reviewer's comment, we now give an example of how our hits can be used to bootstrap the identification of players in dynein-based trafficking that may have been missed in the primary screen (lines 498–500).

'Identifying biochemical interactors of these proteins may reveal additional players in dynein-based trafficking that were refractory to our CRISPR screening approach.'

We also now mention the hypothetical advantage of replicating the genome-wide screen (line 494–497):

'We cannot, however, rule out that some bona fide regulators of dynein function have subtle phenotypes that might be picked up with....replicates of the genome-wide screen.'

2) The screen was developed based on the U-2 OS PEX line, in which tethering of dynein to peroxisomes is achieved by addition of rapamycin acting via a split BICD2 protein. Thus, the assay depends on the BICD2 adapter. Is this limiting when one is looking for dynein regulators?

We chose the BICD2 tethering assay as it gives a strong and tuneable relocalisation phenotype, which is well suited to genome-wide screening. Screening for this phenotype does give access to regulators of dynein and dynactin, which we and others have shown drive peroxisome relocalisation in this system. To report on dynein-based transport by a second adaptor type, we simultaneously assayed endogenous early endosome localisation in the genome-wide screen. Although screening for two cargoes was a large undertaking, we believe it was a key strength of our approach (as pointed out by Reviewer 1).

3) Related to the question above, the authors do not recover BICD1 in their screen. Is this because BICD1 is not expressed in the cell systems used or is there another reason?

BICD1 is moderately expressed in U2OS cells according to proteomic data (Beck et al., 2011; <https://www.embopress.org/doi/full/10.1038/msb.2011.82>). It is not surprising that *BICD1* was not returned in the peroxisome screen as these organelles were transported by induced tethering with BICD2. We now cover these points on lines 204–207:

'The gene encoding the BICD2 paralogue, BICD1, was not recovered in these analyses, despite being expressed in U-2 OS cells (Beck et al., 2011). This result is expected as peroxisome coupling in our assay is mediated by BICD2.'

BICD1 was presumably not recovered in the screen for mislocalisation of early endosomes due to the stated role of Hook proteins as activating adaptors for this cargo.

4) It has very recently been shown (doi.org/10.1038/s41467-023-38116-1) that BICD2 phosphorylation by CDK1 in the G2 phase of the cell cycle promotes its interaction with PLK1. This is followed by PLK1 phosphorylation in the N-terminus of BICD2, which in turn facilitates interaction with dynein and dynactin, allowing the formation of active motor complexes. Thus, adaptor activation through phosphorylation regulates dynein activity. In the present manuscript the authors use PLK1 as a read-out of cell viability. However, PLK1 also appears to regulate dynein via BICD2 phosphorylation. Given the latest results would the authors interpret their PLK1 data differently? Would it be preferable to screen for regulators of dynein activity in non-dividing cells?

We thank the reviewer for pointing out this interesting study. *PLK1* is a strong hit in many genome-wide screens that are unrelated to dynein-based trafficking because of the proliferation defects associated with its disruption (<https://orcs.thebiogrid.org/Gene/5347>). For this reason, it is an effective positive control for editing efficiency in screening projects (e.g. Strezoska et al. 2017; Ross-Thriepland et al. 2020). The proliferation phenotype of *crPLK1* is almost certainly unrelated to its interaction with BICD2, as *crBICD2* did not elicit a mitotic phenotype even though it had a strong effect in the peroxisome relocalisation assay. We now show in Fig. S4 A that *crBICD2* does not affect proliferation and mention this in the context of the study referred to by the reviewer (lines 165–168).

'Although PLK1 was recently implicated in regulation of BICD2 function (Gallisa-Sune et al., 2023), the library copy of crBICD2 did not affect viability (Fig. S4 A) even though it was active in other assays (see below). This observation suggests that PLK1's role in cell proliferation does not involve BICD2.'

5) Using the 377 genes listed in Supplementary Table 4 I performed a Metascape analysis. The results suggest that many of the hits are proteins involved in RNA metabolism or the cell cycle and that many of the encoded proteins form complexes. Based on this I wonder whether the screen yielded many proteins that are involved in controlling the steady state levels of dynein, microtubules, or of the dynein regulators. SUGP1 is an example of this. I suggest that the authors include an extensive Metascape analysis in a new version of the manuscript.

6) On page 11 a UMAP plot is described, which is shown in Figure 4B. How were the "members of the same protein complexes, such as histones, ribosomal proteins, RNA polymerase II, the RUVBL and TRiC/CCT chaperonins, FAM160A2- AKTIP-HOOK3 and dynein-dynactin" identified?

7) How do the complexes identified in Figure 4B relate to the MCODE-based complexes identified in Metascape?

We thank the reviewer for these questions, which we have addressed together as they all relate to analysis of known functions of hits from the screen. In our previous submission, genes encoding proteins that are members of the same complex were identified manually by

inspection of the UNIPROT database, which pools together many datasets, and inspection of associated primary literature. We performed manual curation in order to mitigate against any errors introduced by automated curation of large datasets or false positive interactors from high throughput (and potentially low stringency) protein-protein interaction studies (which usually include no validation).

In response to the reviewer's comments, we have performed the proposed full analysis of the hits using Metascape. The Metascape pathway and process enrichment analysis referred to above by the reviewer is now included on the second tab of Table S4. This analysis does indeed show particularly strong enrichment of terms related to RNA metabolism and cell cycle, as well as enrichment of terms related to several other processes, including trafficking and microtubule cytoskeleton organisation (stated on lines 242–246). We thank the reviewer for leading us to include this informative analysis. It is possible that genes associated with RNA metabolism affect steady state levels of proteins important for transport, as we indeed show is the case for *SUGP1* and *LIS1*. However, testing the effect of targeting so many genes on the expression of multiple dynein and dynactin components and tubulin isoforms would require a separate study. Nonetheless, we believe that the clustered phenotypic fingerprints (Fig. S6; explorable in the 'Supplementary phenotypic heatmap' file and Table S8) already provide a useful tool for generating hypotheses for how specific genes affect trafficking.

Regarding the MCODE-based complexes identified by analysing the hits in Metascape, there are several aspects of these that appear unreliable. This is presumably due to the aforementioned problems with high throughput interactome studies, which can result in artefacts. To give just two examples: (i) some histones cluster with ribosomal components whereas others cluster with dynein-dynactin components, and some components of RNA polymerase II cluster with the proteasome whereas others cluster with dynein-dynactin; we are not aware of any compelling evidence that these interactions exist. We therefore do not believe it is appropriate to replace our manually curated protein complexes in Fig. 4B (now Fig. 5B) with one that incorporates MCODE-based complexes. We have, however, included information on how the complexes were manually defined in the legend to Fig. 5 and in the Materials and Methods (lines 741–742). We have also now labelled the components of the protein complexes in the figure panel, which should help convince readers that bona fide complexes are depicted, and provided the source data with gene names in Table S7 so that readers can further interrogate the data. We thank the reviewer for leading us to clarify this part of the work.

Reviewer #2 (Significance (Required)):

I think the present manuscript is an interesting resource paper for the dynein community. The advance is technical rather than conceptual.

I am a cell biologist with an interest in microtubules and how this cytoskeletal network controls cell shape and function. I analyse this using fluorescence microscopy and -omics approaches. I am not an expert in high-content imaging screens and analyses but the data presented here seem solid and novel to me.

Reviewer #3 (Evidence, reproducibility and clarity (Required)):

In this manuscript, the authors performed an arrayed CRISPR loss-of-function screen targeting 18,253 genes with the goal of uncovering gene products that regulate cytoplasmic dynein-1 motor function. In order to assess the impact of gene knockout, the authors optimized a protocol for transfecting pools of cells with mRNA encoding Cas9 and scalably delivering arrayed pools of synthetic guides targeting a single gene to knock-out. In order to link gene

knockouts to dynein-1 function the authors employed (1) a previously developed cell model U-2 OS PEX and (2) anti-EEA1 and anti- α -Tubulin antibodies and (3) hoechst as high-content fluorescent readouts for their genome-wide screen.

The authors then picked a subset of genes to move forwards with that were deemed as hits. A secondary round of screening was performed on these hit genes and unsupervised phenotypic clustering was performed on the feature vectors derived from the high-content images. These analyses revealed several distinct phenotypic clusters that can be categorized by the dynein cargoes or other functional categories including proteostasis related functions. The authors identified the gene SUGP1, which has never previously been linked to dynein-dynactin functionality.

The authors then show that targeting SUGP1 reduces the mRNA of both LIS1 and DYNC112 and the subsequent protein abundance of only LIS1.

In summary, the authors provide an optimized method for performing what they have termed 'one-shot' genome wide arrayed screening with pools of synthetic guides. They additionally have generated a data resource for others interested in understanding early endosome pathways and dynein-dynactin functionality.

The technical feat of generating such a large dataset and optimizing the protocol for arrayed synthetic guide pools will undoubtedly be useful for the community. However, this work has several limitations including (1) lack of adequate documentation for reproducing the analyses and (2) minimal mechanistic insight into the function of SUGP1.

Major Comments:

- The authors do not provide code or even pseudocode for the algorithms used to generate the features from the high-content images. If the authors are claiming that this would be a resource for the community to use then the authors need to provide an easy way for others to recreate their analysis.

The algorithms we used are from the commercial Columbus software, which we used as it is fully integrated in the Functional Genomics pipeline at AstraZeneca. Because the software is proprietary we do not have access to the codes. Nonetheless, we agree with the reviewer that we should make it clear how we generated data from the high-content images so that others can recreate the analysis in Columbus or take equivalent steps with other platforms. We now include details for each of the features in a new, 29-page supplemental methods file ('Supplemental image analysis file'). This document additionally includes other information requested by the reviewers. We have also extended substantially the section on the analysis procedures in the Materials and Methods (lines 675–727) in order to summarise the steps described in the new supplemental file. We thank the reviewer for leading us to improve the manuscript in this way.

- The authors mention that they will make the images from their screen publicly available, which is an essential part of making their work a useful resource for the community. However, more details need to be provided about how they will share the results. While a "data dump" of images will be useful to a narrow group of computationally savvy scientists, the broader community will require an interactive interface to enable browsing of the data. The authors should establish such a platform and make it available to reviewers of the revised manuscript to evaluate its usefulness.

The images from the project will be hosted on the AstraZeneca Open Innovation platform after publication of the manuscript; readers will then be able to batch download the data. Whilst, just like with transcriptomic datasets, reprocessing of the data to mine new information will

require computational skills, other users will be able to visualise images for genes that hold a particular interest, should they wish. We do not believe that constructing a website to allow browsing of individual images will add significant value to the resource. Browsing individual images in isolation will have limited use without being able to analyse the numerous controls simultaneously. For the vast majority of users, much of the value of the resource will come from the quantitative data that we have already extracted from the images and provided in an accessible spreadsheet format. These data can be easily plotted with users' favourite tools, even by those that are not computationally savvy.

Moreover, building and maintaining a website to browse individual images will be a large amount of work and we do not have the resources to deliver this. We note that several other large-scale imaging-based CRISPR screens (e.g. Yan et al. 2021, *J Cell Biol*, <https://doi.org/10.1083/jcb.202008158>; Kanfer et al., *J Cell Biol*, doi.org/10.1083/jcb.202006180; Ross-Thriepland et al., *SLAS Discovery* 2020, [https://slas-discovery.org/article/S2472-5552\(22\)06576-5/fulltext](https://slas-discovery.org/article/S2472-5552(22)06576-5/fulltext)) have not produced a web interface, presumably due to the issues described above.

- The authors highlight SUGP1 as an example for "novel mechanistic insights" - but the insights they provide are really minimal. If they authors want to claim mechanistic insights, they should experimentally address questions such as: Does SUGP1 physically interact with LIS1 mRNA? Which region of LIS1 mRNA confers regulation by SUGP1? Can the authors generate a version of LIS1 resistant to SUGP1 regulation to show that the effect of SUGP1 loss is mediated by LIS1 (and not additional factors?).

Whilst we fully agree with the reviewer's point about not defining a detailed molecular mechanism, we respectfully disagree that the insights on SUGP1 are 'really minimal'. In order to provide a proof-of-principle of how our screen can provide new knowledge about dynein biology, we performed extensive transcriptomic and splicing analysis of SUGP1-edited cells, NTC-treated cells and *crXCR1*-edited cells (a stringent (and atypical) control in which we factored in a potential contribution of double strand breaks to gene expression changes), as well as functional experiments to validate a hypothesis generated from the approach. Our data indicate that *LIS1* mRNA expression is a key target of SUGP1 in the context of dynein-based trafficking and that this effect does not involve changes in *LIS1* splicing or 3'-end usage. As Reviewer 1 points out, this work provides an 'interesting new way of regulating dynein function'. Discovering the underlying molecular mechanisms would constitute a new, long-term study and therefore goes well beyond the scope of what can be included in the paper, which already covers a lot of ground. Therefore, in the revised manuscript we have taken the steer of the reviewer to remove the term 'mechanistic insight' when referring to the SUGP1 work. We now also explicitly mention the need for further mechanistic studies on line 566–568.

'Further investigation of the molecular mechanism by which SUGP1 promotes LIS1 expression could shed light on post-transcriptional regulation of dynein-based trafficking, which is a largely unexplored topic.'

Minor Comments:

- Primary and Secondary antibody pairs are described nowhere in this paper. This would be impossible for anyone to recreate with just the list of primary and secondary antibodies used here.

This has been added (Table S13). We thank the reviewer for the suggestion.

- The authors provide no description of how the segmentation was performed or any reference to the code that they used for segmentation regarding the definition of perinuclear region.

Considering so many of the results are based on these values it is important that others are able to recreate these values.

We have now added information on segmentation procedures in Columbus in the 'Supplemental image analysis file'; this document also included examples of the outputs of segmentation (as requested below).

- Line 132: The authors do not explain what a min-max analysis is anywhere in the paper. This should be explained.

We now see that min-max was not a helpful term. We now refer to these experiments based on their purpose of testing scalability of the editing procedures (line 121).

- There is no discussion of how the authors quantify micronuclei formation. If they state that they are the first to do this and that this is a novel technique they at the minimum need to explain the methods for quantifying micronuclei.

There seems to be a slight misunderstanding here, which results from us not being sufficiently clear in the first submission. We did not mean to convey that our quantification method was novel but rather that other genome-scale screens had not looked specifically at micronuclei formation. We have now modified the language in line 179–181 to address this point. We also provide details in the 'Supplemental image analysis file' of how the quantification was performed for the micronuclei analysis, including example images. The process is also summarised in the Materials and Methods (line 715–720).

- Supplemental Fig 4C if a per cell intensity quantification is done I would like to see a metric for the segmentation accuracy on these cells overlaid with a cytoskeletal stain.

This experiment did not use a cytoskeletal stain for segmentation, but rather the diffuse background signal of Hoechst staining. The Columbus package does not, as far as we are aware, provide a metric for segmentation accuracy for cell boundaries. However, the segmentation of cells worked well with the Hoechst-based method (with a few minor errors that should be equivalent across wells) as exemplified in Fig. R2.

Figure R2. Example of cell and nuclear segmentation output based on Hoescht staining overlaid on DYNC1H1 IF signals (NTC cells).

- It would be nice to have examples of nuclei or morphology that were excluded from downstream analysis, perhaps in a supplemental figure.

Examples are now included in the Supplemental image analysis file.

- Nowhere in the manuscript is it explained how the SUGP1 intensity measurement in Figure 6D is calculated, is this one a per well basis or a per cell basis?

Fig. 6 D (now Fig. 7 D) shows intensity measurements for DYNC1I2 and LIS1, not SUGP1. Nonetheless, the reviewer makes another important point. We have now clarified in this and other relevant legends that the values were derived from aggregating per cell intensity values at the per well level (at least 4 wells from a 384-well plate per experiment), i.e. plotting each 'mean per well' value as an individual circle.

Reviewer #3 (Significance (Required)):

The generation of the dataset described in this manuscript is impressive. However, to reach its full significance and usefulness for the scientific community, the authors should provide relevant technical details, in particular of their analysis pipeline, and share the screen results in an accessible, interactive interface. If they want to claim mechanistic insights into SUGP1, more mechanistic work is required.

As described above, we have provided much more information on the image analysis pipeline and removed the phrase 'mechanistic insight' from the SUGP1 sections. We are grateful to the reviewer for leading us to improve these aspects of the study. Furthermore, the images from the screen will be accessible on the AstraZeneca Open Innovation platform, as described above, with extensive quantitative data from image analysis accessible in the manuscript supplement.

February 12, 2024

RE: JCB Manuscript #202306048R

Dr. Simon L Bullock
MRC Laboratory of Molecular Biology
Cell Biology
Francis Crick Avenue
Cambridge CB2 0QH
United Kingdom

Dear Dr. Bullock:

Thank you for submitting your revised manuscript entitled "Genome-scale requirements for dynein-based transport revealed by a high-content arrayed CRISPR screen". We would be happy to publish your paper in JCB pending final revisions necessary to meet our formatting guidelines (see details below).

A. MANUSCRIPT ORGANIZATION AND FORMATTING:

- 1) Text limits: Character count for Tools is < 40,000, not including spaces. Count includes abstract, introduction, results, discussion, and acknowledgments. Count does not include title page, figure legends, materials and methods, references, tables, or supplemental legends.
- 2) Figures limits: Tools may have up to 10 main text figures.
- 3) Figure formatting: Scale bars must be present on all microscopy images, including inset magnifications. Molecular weight or nucleic acid size markers must be included on all gel electrophoresis.
- 4) Statistical analysis: Error bars on graphic representations of numerical data must be clearly described in the figure legend. The number of independent data points (n) represented in a graph must be indicated in the legend. Statistical methods should be explained in full in the materials and methods. For figures presenting pooled data the statistical measure should be defined in the figure legends. Please also be sure to indicate the statistical tests used in each of your experiments (either in the figure legend itself or in a separate methods section) as well as the parameters of the test (for example, if you ran a t-test, please indicate if it was one- or two-sided, etc.). Also, if you used parametric tests, please indicate if the data distribution was tested for normality (and if so, how). If not, you must state something to the effect that "Data distribution was assumed to be normal but this was not formally tested."
- 5) Abstract and title: The abstract should be no longer than 160 words and should communicate the significance of the paper for a general audience. The title should be less than 100 characters including spaces. Make the title concise but accessible to a general readership.
- 6) Materials and methods: Should be comprehensive and not simply reference a previous publication for details on how an experiment was performed. Please provide full descriptions in the text for readers who may not have access to referenced manuscripts.
- 7) * All antibodies, cell lines, animals, and tools used in the manuscript should be described in full, including accession numbers for materials available in a public repository such as the Resource Identification Portal. Please be sure to provide the sequences for all of your primers/oligos and RNAi constructs in the materials and methods. You must also indicate in the methods the source, species, and catalog numbers (where appropriate) for all of your antibodies. Please also indicate the acquisition and quantification methods for immunoblotting/western blots. *
- 8) Microscope image acquisition: The following information must be provided about the acquisition and processing of images:
 - a. Make and model of microscope
 - b. Type, magnification, and numerical aperture of the objective lenses
 - c. Temperature
 - d. Imaging medium

- e. Fluorochromes
- f. Camera make and model
- g. Acquisition software
- h. Any software used for image processing subsequent to data acquisition. Please include details and types of operations involved (e.g., type of deconvolution, 3D reconstitutions, surface or volume rendering, gamma adjustments, etc.).

10) * Supplemental materials: There are strict limits on the allowable amount of supplemental data. Tools may have up to 5 supplemental figures. While we will be able to give you a bit more space, please try to reduce the count by moving data to the main figures and/or combining SI figures if possible. Please be sure to correct the callouts in the text to reflect any changes. Please also note that tables, like figures, should be provided as individual, editable files. A summary of all supplemental material should appear at the end of the Materials and methods section.

13) ORCID IDs: ORCID IDs are unique identifiers allowing researchers to create a record of their various scholarly contributions in a single place. Please note that ORCID IDs are now *required* for all authors. At resubmission of your final files, please be sure to provide your ORCID ID and those of all co-authors.

Please note that JCB now requires authors to submit Source Data used to generate figures containing gels and Western blots with all revised manuscripts. This Source Data consists of fully uncropped and unprocessed images for each gel/blot displayed in the main and supplemental figures. Since your paper includes cropped gel and/or blot images, please be sure to provide one Source Data file for each figure that contains gels and/or blots along with your revised manuscript files. File names for Source Data figures should be alphanumeric without any spaces or special characters (i.e., SourceDataF#, where F# refers to the associated main figure number or SourceDataFS# for those associated with Supplementary figures). The lanes of the gels/blots should be labeled as they are in the associated figure, the place where cropping was applied should be marked (with a box), and molecular weight/size standards should be labeled wherever possible.

Journal of Cell Biology now requires a data availability statement for all research article submissions. These statements will be published in the article directly above the Acknowledgments. The statement should address all data underlying the research presented in the manuscript. Please visit the JCB instructions for authors for guidelines and examples of statements at (<https://rupress.org/jcb/pages/editorial-policies#data-availability-statement>).

B. FINAL FILES:

****It is JCB policy that if requested, original data images must be made available to the editors. Failure to provide original images upon request will result in unavoidable delays in publication. Please ensure that you have access to all original data images prior to final submission.****

****The license to publish form must be signed before your manuscript can be sent to production. A link to the electronic license to publish form will be sent to the corresponding author only. Please take a moment to check your funder requirements before choosing the appropriate license.****

Thank you for this interesting contribution, we look forward to publishing your paper in Journal of Cell Biology.

Sincerely,

Melissa Gardner
Monitoring Editor

Andrea L. Marat
Senior Scientific Editor

Journal of Cell Biology

Reviewer #1 (Comments to the Authors (Required)):

Wong et al. performed a genome-wide CRISPR loss-of-function screen using peroxisome-tethered and endosome localization assays as readouts to identify genes linked to dynein function. Detailed data analysis and supervised and unsupervised phenotypic clustering of targets from an extensive RNA gene library (~20,000 genes) revealed ~200 genes disrupted in different cargo trafficking processes. The authors also showed that a novel gene SUGP1, identified in their screen, promotes the expression of a critical dynein activator, LIS1. The generated datasets provide a rich library of genes that can be further mined by other researchers in the field.

The data provided supports the conclusions and is clearly presented. The controls used are sufficiently tested and validated. In the revised version, the authors provided additional explanations (main text and methods) on off-targeting due to multiple crRNAs used, which was my major concern in the original version. The additional explanation of the methods used for image analysis also strengthens the manuscript, allowing other researchers to replicate the results. The follow-up experiments used to validate the novel regulator of LIS1 mRNA levels, SUGP1, are sufficient and within the scope of this manuscript.

The authors have adequately addressed my previous comments. The revised version of the manuscript is publication-quality.